

# Looking at supersymmetric black holes for a very long time

Henry W. Lin[1,2], Juan Maldacena[2], Liza Rozenberg[1] and Jieru Shan[1]

**1** Jadwin Hall, Princeton University, Princeton, NJ 08540, USA
**2** Institute for Advanced Study, Princeton, NJ 08540, USA

## Abstract

We study correlation functions for extremal supersymmetric black holes. It is necessary to take into account the strongly coupled nature of the boundary supergraviton mode. We consider the case with $\mathcal{N} = 2$ supercharges which is the minimal amount of supersymmetry needed to give a large ground state degeneracy, separated from the continuum. Using the exact solution for this theory we derive formulas for the two point function and we also give integral expressions for any $n$-point correlator. These correlators are time independent at large times and approach constant values that depend on the masses and couplings of the bulk theory. We also explain that in the non-supersymmetric case, the correlators develop a universal time dependence at long times. This paper is the longer companion paper of [1].



## 1   Introduction

We study the extremal limit of supersymmetric black holes with an AdS$_2$ near horizon geometry and develop methods to compute correlation functions of local operators. These correlators give us statistical information about physical properties of the supersymmetric ground states of the system. In other words, if $O$ represents a simple operator measuring the value of some field around the black hole, we are interested in the correlation functions of such operator once it is projected on to the ground states

$$\text{Tr}\big[\hat{O}_1 \cdots \hat{O}_n\big], \quad \hat{O} \equiv POP, \quad P \equiv \lim_{\beta \to \infty} e^{-\beta H}, \tag{1}$$

where $P$ is the projector on to zero energy states (zero energy above extremality). These correlators give us some information about how the ground states look to an observer outside that measures the values of various fields. This information is statistical, since it is interpreted as an average over all the ground states, due to the trace in (1). Of course, the formula (1) is an *interpretation* of a gravity computation that we will do on the Euclidean black hole geometry in the limit that $\beta \to \infty$ with all operators at fixed angles in the Euclidean circle.

In this extremal limit, it is necessary to take into account the quantum mechanics of a certain gravitational mode that becomes strongly coupled at low energies [2–6]. Fortunately, this dynamics can be exactly solved [7–11]. In our case, we are interested in a supersymmetric version of this boundary theory [9, 12, 13] so that we have a large ground state degeneracy. The amount of supersymmetry depends on the amount of supersymmetry that the black hole preserves. For simplicity, we will consider the minimal case with a large degeneracy [9, 12], which is a total of $\mathcal{N} = 2$ supersymmetries. As an example, this is the case that describes the supersymmetric black hole in $AdS_5 \times S^5$ [14], in the fixed charge sector [15]. Supersymmetric extremal black holes in asymptotically flat space have four supercharges, $\mathcal{N} = 4$, and the corresponding quantum dynamics was studied in [16].

We quantize the supersymmetric Schwarzian theory using two approaches. In the first approach, we focus on the two sided system in a thermofield double-like state. See figure 1a. We use its gauge invariant description in terms of a supersymmetric Liouville theory [17]. The two point function then becomes a one point function in the super-Liouville quantum mechanics. This lets us interpolate between a short distance conformal regime to the long distance regime where only zero energy states contribute and the two point function becomes constant. With a suitable normalization of the operator, this constant value depends only on the anomalous dimension of the operator.

As a second approach, we consider the propagator in the supersymmetric Schwarzian theory, extending the discussion in [10, 11, 18]. We focus on the zero energy propagator, which is appropriate for computing properties of the zero energy limit of the supersymmetric black

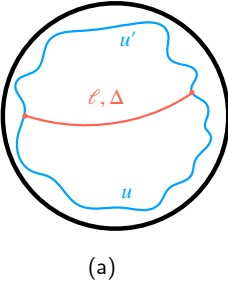
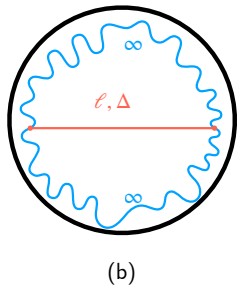
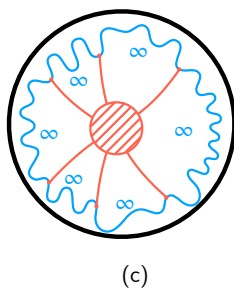

(a)  (b)  (c)

Figure 1: Various correlators. (a) Two point function at finite boundary euclidean times, $u, u'$. (b) Two point function in the zero energy limit, with $u = u' = \infty$. (c) A general long time correlator.

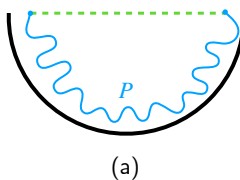
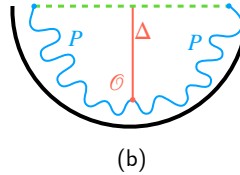
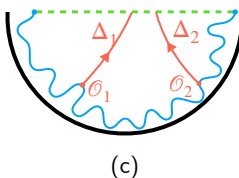

(a)  (b)  (c)

Figure 2: (a) The construction of an empty supersymmetric wormhole arising after the evolution over a long Euclidean time. (b) By adding an operator during the Euclidean time evolution we produce extra matter. (c) We could add several single particle operators to produce a multiparticle state in the wormhole. The length of the green line represents the size of the wormhole.

holes. By zero energy, we mean zero energy above the extremal value. These zero energy states preserve supersymmetry. Using these propagators we can compute correlation functions of the simple operators that have definite conformal dimensions in the higher energy theory. These are given by correlation functions in $\text{AdS}_2$ dressed by this zero energy boundary propagator, see figure 1c.

Of course, the zero energy states give rise to the extremal entropy, a very well studied subject since the initial match of the area formula [19], and including the very interesting non-perturbative contributions as in [20]. The correlators we are studying give us further statistical information about what these states "look like" to an observer that measures fields around the black hole. We can view these as the correlators of a "topological quantum mechanics", which is simply a theory with no Hamiltonian, $H = 0$, where the correlators do not depend on time, but could depend on the order of the operators. It is a theory where the asymptotic symmetries of $\text{AdS}_2$, time reparametrizations, become actual symmetries of the correlators.

This discussion lets us construct and explore supersymmetric wormhole configurations. These arise by taking the low energy limit of the standard wormhole that arises at finite temperature. One interesting feature is that the length of the wormhole stays constant in the extremal limit of zero energies. By inserting operators, we can add matter to this wormhole, see figure 2. The matter need not be BPS, but nevertheless we get a supersymmetric wormhole after taking this low energy limit. We argue that adding matter in this way increases the distance between the two sides and it also decreases the entanglement entropy between the two sides. See figure 2.

Perhaps the most interesting aspect of this description is that the theory in the bulk contains a time direction, while the theory in the boundary contains no time in this limit, the Hamiltonian is zero acting on the ground states. So, this could be viewed a system with an emergent time. In this description, this time coordinate appears to arise from the projector $P$

in (1). This projector is realized on the gravity side by undergoing a long period of Euclidean evolution on both sides of the operator. The fact that there can be matter in the bulk at zero energy is related to the fact that the bulk energy, or bulk time, is not the same as the boundary energy, or boundary time. In fact, this observation is useful to circumvent arguments against the existence of an exact description for zero energy AdS$_2$ backgrounds [21,22]. It is also interesting to note that the entanglement entropy of the empty wormhole in figure 2 is maximal, $S = S_0$, the extremal entropy, while the entanglement entropy of the wormholes with additional matter can only be smaller. This is a structure which, in the infinite $S_0$ limit, becomes that of a type II$_1$ algebra, as in the de Sitter discussion of [23]. In fact, our system has some vague similarity to de Sitter, in the sense that $H = 0$ and that there is an emergent bulk time.

The formalism that we develop here can also be used to study the low energy limit of the $\mathcal{N} = 2$ supersymmetric SYK model introduced in [12]. In fact, our results make specific predictions about the SYK correlators which we check by performing a numerical exact diagonalization of the model for the case of $N = 16$ complex fermions. We find agreement, within a few percent, with the formulas following from the quantization of the super-Liouville theory discussed above.

We have focused on supersymmetric black holes because in this case there is a sharp sector at zero energy separated from the higher energies by a gap. However many qualitative questions are rather similar in the non-supersymmetric case. In the non supersymmetric case the low energy correlators become universal functions of the boundary time. The bulk correlators determine an overall numerical coefficient. So, in this case too, there is a certain disconnect between the bulk time and the boundary time.

This paper is organized as follows. In section 2, we describe the computation of the two point function using the gauge invariant variables of the empty wormholes, which reduces to a super-Liouville quantum mechanics. We compute the two point functions on the disk and the cylinder. In section 3, we compare these results against numerical SYK computations. In section 4, we introduce a superspace formalism, which is an extension of the one in [13] and we use it to compute the zero energy propagators. We discuss a general integral expression for an $n$ point function. As a check, we reproduce the previous results for the two point functions. In section 5, we use this propagator to study wormholes filled with matter. We discuss how the matter changes the length of the wormhole. In section 6, we discuss aspects of the low energy limit for the non-supersymmetric JT gravity case and point out some common features with the supersymmetric situation.

In various appendices we give extra information. We discuss the computation of the propagator for the case with $\mathcal{N} = 1$ supersymmetry, which is a case initially studied in [13].

## 2 Two point functions from the Liouville quantum mechanics of the empty wormhole.

In this section, we discuss the computation of the two point function using the Liouville method, see figure 3. The dimension $\Delta$ of the operator refers to the behavior of the operator in the finite-temperature conformal regime of the NAdS$_2$ geometry; see also 5.5.[1] This is a method that is based on studying the quantum mechanics of the empty (super) wormhole geometry.

---

[1]We sometimes refer to the finite temperature conformal regime as the "UV" since it involves energies bigger than the gap scale. The dimension $\Delta$ is a property of the "UV" in this sense. It is a property of the fields in the NAdS$_2$ region. It is defined in the throat of the black hole geometry. Please do *not* confuse this with how the throat is embedded in a higher dimensional geometry, which is a further "UV" limit that is irrelevant in this discussion.

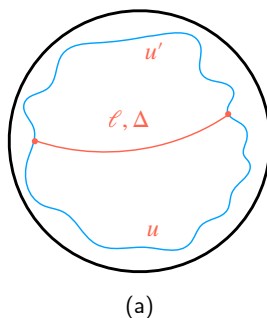
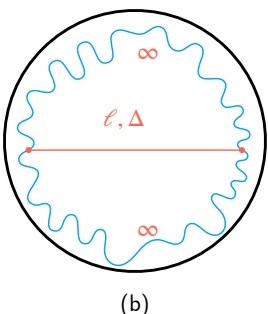

Figure 3: (a) Computation of the two point function of two operators separated by boundary time $u$ on one side and boundary time $u'$ on the other. (b) As $u, u' \to \infty$ we get the zero energy correlator which is a constant. In both cases $\ell$ denotes a renormalized distance between the two boundary points.

The two point function we obtain below can in principle be extracted from a limit of the $\mathcal{N} = 2$ supersymmetric SYK model by using the chord method discussed in [24],[2] where a large $N$ and large $\hat{q}$ version of the model was considered. In other words, [24] performed a more general computation than the one we perform below. Here we obtain directly the JT gravity result by more elementary methods.

## 2.1 Basic set up and Lagrangian

We will first compute the two point function using a method which centers on using the dynamics of the thermofield double state. In the usual non-supersymmetric case, this state is described by a two dimensional phase space consisting of energy and a relative time shift between the two sides [25]. Alternatively we can think in terms of the distance between the two boundaries and its momentum conjugate. In terms of the latter, for JT gravity in the Schwarzian limit, the dynamics is given in terms of a Liouville like action [17, 26]

$$I = \frac{\phi_r}{2\pi} \int \mathrm{d}t \left[ \frac{1}{2} \dot{\ell}^2 + 2 e^{-\ell} \right], \tag{2}$$

where $\ell$ is related to the residual distance after we extract an infinite additive constant related to the Schwarzian limit. In appendix A we give more details and explain how this action follows from the Schwarzian action

$$I = -\frac{\phi_r}{2\pi} \int \mathrm{d}t \{f, t\}, \quad \text{with} \quad \{f, t\} = \frac{f'''}{f'} - \frac{3}{2} \frac{f''^2}{f'^2}, \tag{3}$$

and also explain the black hole interpretation of the parameter $\phi_r$.

If we now consider the supersymmetric case, then we find that the wormhole or thermofield double state is parametrized by further fermionic and possibly bosonic variables. For the $\mathcal{N} = 1$ case, we have a pair of Majorana fermions which are the partners of the length variable under the supersymmetry on the left side or the right side. Notice that, even though there is only one Hamiltonian, there are two separate supersymmetries, each squaring to the same Hamiltonian [17]. As shown in [17], the resulting Lagrangian is the same as the one that we obtain by dimensionally reducing a (1,1) supersymmetric Liouville theory in 1+1 dimensions. Of course, the same answer is obtained by supersymmetrizing (2). When we go to the $\mathcal{N} = 2$ case, we now have two supersymmetries acting on each side, the left and the right. So we have four

---

[2]We thank Vladimir Narovlansky for performing a significant part of this limit and for communicating it to us.

supersymmetries in total. This means that we have two complex fermions. In addition, we have a scalar that is related to the relative phase of the wormhole under the $U(1)_R$ symmetry. In the bulk, this is a Wilson line of a U(1) gauge field across the wormhole. The Lagrangian can be obtained by supersymmetrizing (2) or by dimensionally reducing a $(2,2)$ supersymmetric Liouville theory in 1+1 dimensions [17]. Either way, we get the Lagrangian

$$I = \frac{\phi_r}{16\pi}\left[\int \mathrm{d}t\,\mathrm{d}\kappa_l\,\mathrm{d}\bar{\kappa}_l\,\mathrm{d}\kappa_r\,\mathrm{d}\bar{\kappa}_r L\bar{L} + 8\int \mathrm{d}t\,\mathrm{d}\bar{\kappa}_l\,\mathrm{d}\kappa_r e^{-L/2} + 8\int \mathrm{d}t\,\mathrm{d}\kappa_l\,\mathrm{d}\bar{\kappa}_r e^{-\bar{L}/2}\right], \qquad (4)$$

where $\kappa_{l,r}$ and $\bar{\kappa}_{l,r}$ are superspace variables and $D = \partial_\kappa + \bar{\kappa}\partial_t$, $\bar{D} = \partial_{\bar{\kappa}} + \kappa\partial_t$ are the corresponding covariant derivatives. $L$ is a (twisted) chiral superfield obeying $\bar{D}_r L = 0 = D_l L$, which starts as $L = \ell + 2ia + \kappa$-terms, where $a$ is the phase associated to a relative $U(1)_R$ symmetry. We can expand the Lagrangian to obtain the Lagrangian in components

$$I = \int \mathrm{d}u \left[\frac{1}{4}\dot{\ell}^2 + \dot{a}^2 + i\bar{\psi}_r\dot{\psi}_r + i\bar{\psi}_l\dot{\psi}_l + i\bar{\psi}_l\psi_r e^{-\ell/2-ia} + i\psi_l\bar{\psi}_r e^{-\ell/2+ia} + e^{-\ell}\right], \qquad (5)$$

where we have set $\phi_r = \pi$ as a choice of units of time and normalized the fermions appropriately. This choice corresponds to setting the coupling in front of the Schwarzian (3) to a half. Or equivalently, the time $t$ in (3) and the time $u$ in (5) are related by

$$t = \frac{\phi_r}{\pi}u. \qquad (6)$$

From now on we will work in terms of the time $u$. There is an $R$ symmetry that acts on the right side, $J_r$ and one on the left side, $J_l$

$$J_r = -i\partial_a - \frac{1}{2}[\bar{\psi}_r, \psi_r], \quad J_l = i\partial_a - \frac{1}{2}[\bar{\psi}_l, \psi_l], \qquad (7)$$

which are defined so that $\psi_r$, $\psi_l$, have charge one, $[J_r, \psi_r] = \psi_r$ and $[J_l, \psi_l] = \psi_l$. In the systems we consider there is some quantization condition on the $R$ charge which is part of the definition of the superSchwarzian theory. In principle, this can be derived from the UV model which gives rise to the Schwarzian theory. This amounts to saying that the $R$ charges are quantized in units of $1/\hat{q}$, which means that the field $a$ is periodic with period $a \sim a + 2\pi\hat{q}$, where $\hat{q}$ is some integer. In the case of the black hole in $AdS_5 \times S^5$ we have $\hat{q} = 1$ [15]. In the SYK section 3 we will get models with odd values of $\hat{q} = 3,\ 5, \ldots$[3]

We can write down the supercharges

$$Q_r = \psi_r(i\partial_\ell + \frac{1}{2}\partial_a) + e^{-\ell/2+ia}\psi_l, \qquad\qquad \bar{Q}_r = \bar{\psi}_r(i\partial_\ell - \frac{1}{2}\partial_a) + e^{-\ell/2-ia}\bar{\psi}_l,$$

$$Q_l = \psi_l(i\partial_\ell - \frac{1}{2}\partial_a) - e^{-\ell/2-ia}\psi_r, \qquad\qquad \bar{Q}_l = \bar{\psi}_l(i\partial_\ell + \frac{1}{2}\partial_a) - e^{-\ell/2+ia}\bar{\psi}_r, \qquad (8)$$

obeying the supersymmetry algebra

$$\{Q_r, Q_l\} = 0 = \{\bar{Q}_r, \bar{Q}_l\}, \quad \{Q_r, \bar{Q}_r\} = \{Q_l, \bar{Q}_l\} = H, \qquad (9)$$

with

$$H = -\partial_\ell^2 - \frac{1}{4}\partial_a^2 + i[\bar{\psi}_l\psi_r e^{-\ell/2-ia} + \psi_l\bar{\psi}_r e^{-\ell/2+ia}] + e^{-\ell}. \qquad (10)$$

---

[3]In principle, we can also have a quantization condition which is saying that $j = (n + \frac{1}{2})\frac{1}{\hat{q}}$, with integer $n$. This arises in the $\mathcal{N} = 2$ SYK for odd $N$. We will not describe explicitly this case, but our formulas below are also valid in that case.

Notice that $[Q_r, \ell] = i\psi_r, [\bar{Q}_r, \ell] = i\bar{\psi}_r$ and similarly for the left supercharges. So these fermions are the superpartners of $\ell$, which is geometrically interpreted as the length between the two sides. In section 4 we will discuss the geometric interpretation of $\psi$ (see (132) for the identification). We also see that we have separate partners for the right and left supercharges.

Although the supercharges are not invariant under $J_l, J_r$, they are invariant under the $Z_{\hat{q}} \times Z_{\hat{q}}$ subgroup generated by $g_l = e^{2\pi i J_l}, g_r = e^{2\pi i J_r}$. In addition, we can define $(-1)^{F_l} = \exp(\hat{q}\pi i J_l), (-1)^{F_r} = \exp(\hat{q}\pi i J_r)$ which anti-commute with $Q_l, Q_r$ respectively. The total fermion number $(-1)^F = (-1)^{F_l}(-1)^{F_r}$.

## 2.2 Eigenfunctions of the super-Liouville theory

We now study the wavefunctions which are energy eigenstates of (10). We can choose a fermion vacuum given by

$$\psi_r \left| \frac{1}{2}, \frac{1}{2} \right\rangle = 0 = \psi_l \left| \frac{1}{2}, \frac{1}{2} \right\rangle, \tag{11}$$

where, according to (7), $|\frac{1}{2}, \frac{1}{2}\rangle$ has $J_l = J_r = \frac{1}{2}$, which explains the notation. We also see that there is a connection between the fermion number and the total $R$ charge $J_l + J_r$. In particular, (11) has odd fermion number. Since the thermofield double state has zero total charge, $(J_l + J_r)|\text{TFD}\rangle = 0$, we are interested in constructing states with opposite values of the $R$ charge $J_r = -J_l = j$. Such states automatically have zero fermion number. For states with positive energy, we find four states in the multiplet that we get by acting with the supercharges. We could pick two of them to have $J_r = -J_l$ and even fermion number, but the other two will not have either of these properties. In fact, it is simplest to start from a state with odd fermion number of the form

$$|F^+\rangle = e^{i(j-\frac{1}{2})a} h(\ell) \left| \frac{1}{2}, \frac{1}{2} \right\rangle, \quad Q_l|F^+\rangle = Q_r|F^+\rangle = 0, \quad J_r = j, \quad J_l = -j+1, \tag{12}$$

where $h(\ell)$ is a function we will soon fix. Note that the last two conditions are automatic due to (11) and the expressions of the supercharges (8). The Hamiltonian (10) acts in a simple way on this state since the terms with fermions annihilate $\left| \frac{1}{2}, \frac{1}{2} \right\rangle$ thanks to (11), leading to a simple equation for $h$

$$-h'' + e^{-\ell} h = \left[ E - \frac{1}{4}\left(j - \frac{1}{2}\right)^2 \right] h, \tag{13}$$

whose solution is

$$h = \frac{2\sqrt{E}}{\sqrt{\pi}} K_{2is}(2e^{-\ell/2}), \quad E = s^2 + \frac{(j-\frac{1}{2})^2}{4}. \tag{14}$$

We have chosen the solution that decays as $\ell \to -\infty$. The factor of $2\sqrt{E/\pi}$ just sets a convenient normalization. The solution is continuum normalizable if $s$ is real. We will denote this state as $|F_{s,j}^+\rangle$.

Having found this state we can now act with $\frac{\bar{Q}_l}{\sqrt{E}}$ to produce

$$|H_{s,j}\rangle = e^{ija} \left[ -g_1 e^{ia/2} \bar{\psi}_r + i g_2 e^{-ia/2} \bar{\psi}_l \right] \left| \frac{1}{2}, \frac{1}{2} \right\rangle, \quad J_r = -J_l = j, \tag{15}$$

$$g_1 = \frac{2}{\sqrt{\pi}} e^{-\ell/2} K_{2is}(2e^{-\ell/2}), \tag{16}$$

$$g_2 = \frac{1}{\sqrt{\pi}} \left[ 2e^{-\ell/2} K_{1-2is}(2e^{-\ell/2}) + \left(j - \frac{1}{2} + 2is\right) K_{2is}(2e^{-\ell/2}) \right]. \tag{17}$$

This state has zero fermion number and $J_r = -J_l = j$. Despite appearances, both functions are invariant under $s \to -s$. Similarly, acting with $\frac{i\bar{Q}_r}{\sqrt{E}}$ on (12) we get

$$|L_{s,j}\rangle = e^{i(j-1)a}\left[-\tilde{g}_1 e^{ia/2}\bar{\psi}_r + i\tilde{g}_2 e^{-ia/2}\bar{\psi}_l\right]\left|\frac{1}{2},\frac{1}{2}\right\rangle, \quad J_r = -J_l = j-1, \qquad (18)$$

$$\tilde{g}_1 = \frac{1}{\sqrt{\pi}}\left[2e^{-\ell/2}K_{1-2is}(2e^{-\ell/2}) + \left(\frac{1}{2}-j+2is\right)K_{2is}(2e^{-\ell/2})\right], \qquad (19)$$

$$\tilde{g}_2 = \frac{2}{\sqrt{\pi}}e^{-\ell/2}K_{2is}(2e^{-\ell/2}). \qquad (20)$$

This state also has zero fermion number and $J_r = -J_l = j-1$. Note that it is in the same multiplet as (15) and it has a different $R$ charge, but its energy is still given by equation (14). Finally, there is a fourth state in the multiplet, obtained by acting with both supercharges, $\frac{1}{E}\bar{Q}_r\bar{Q}_l|F_+\rangle$, which gives

$$|F^-_{s,j}\rangle = e^{i(j-\frac{1}{2})a}\frac{2\sqrt{E}}{\sqrt{\pi}}K_{2is}(2e^{-\ell/2})\bar{\psi}_r\bar{\psi}_l\left|\frac{1}{2},\frac{1}{2}\right\rangle, \qquad J_r = j-1, \quad J_l = -j. \qquad (21)$$

We have now obtained the four states in the multiplet.

We had mentioned that $|F^+_{s,j}\rangle$ is normalizable only when $s$ is real. However, the solution $|H_{s,j}\rangle$ is also normalizable when we take $s$ to be imaginary and equal to $s = \pm i\frac{1}{2}(j-\frac{1}{2})$. (it does not matter if the first sign if plus or minus since the wavefunction is invariant under $s \to -s$.). These states have zero energy $E = 0$. They have the form

$$|Z_j\rangle = |H_{s,j}\rangle\big|_{s=\frac{i}{2}(j-\frac{1}{2})} = \frac{2}{\sqrt{\pi}}e^{ija}\left[-e^{ia/2}e^{-\ell/2}K_{\frac{1}{2}-j}(2e^{-\ell/2})\psi_r + ie^{-ia/2}e^{-\ell/2}K_{\frac{1}{2}+j}(2e^{-\ell/2})\psi_l\right]\left|\frac{1}{2},\frac{1}{2}\right\rangle. \qquad (22)$$

Due to the overall factor of $e^{-\ell/2}$, these wavefunctions are normalizable as long as $|j| < \frac{1}{2}$. These zero energy wavefunctions are annihilated by all four supercharges. They represent normalizable ground states in the Liouville potential. The purely bosonic part of the potential is positive and goes as $e^{-\ell}$. The term involving fermions can be negative if the fermions act on the right state, and it goes like $e^{-\ell/2}$ which can dominate over the purely bosonic one for large $\ell$. These two terms can in principle lead to bound states. This analysis shows that we get just a single zero energy bound state for each $R$ charge $j$ in the range $|j| < \frac{1}{2}$.

We can also work with $|L_{s,j}\rangle$ to get the same zero energy states. We need to set $s = \pm\frac{i}{2}(j-\frac{1}{2})$, which gives $|Z_{j-1}\rangle$.

The theory (5) has a charge conjugation symmetry where $a \to -a - \pi$ and the barred and unbarred fermions are exchanged. Under this symmetry we have that $(H_j, L_j) \to (L_{1-j}, H_{1-j})$. This is also consistent with figure 4.

It will be useful for us to record the norm of these states. For example, in the case of the states $|H_{s,j}\rangle$ the inner product is given by

$$\langle H_{s',j'}|H_{s,j}\rangle = \int_{-\infty}^{\infty}d\ell\int_0^{2\pi\hat{q}}\frac{da}{2\pi\hat{q}}(g_1'^*g_1 + g_2'^*g_2), \qquad (23)$$

where we picked a measure factor that includes a factor of $1/(2\pi\hat{q})$. This gives

$$\langle L_{s',j'}|L_{s,j}\rangle = \langle H_{s',j'}|H_{s,j}\rangle = \delta_{j,j'}\delta(s-s')\frac{E\pi}{s\sinh 2\pi s}, \qquad (24)$$

where we assumed that the period of $a$ is $a \sim a + 2\pi\hat{q}$, so that we can allow fractional charges. A simple trick to compute these norms will be mentioned after (60). The $|F^+\rangle$ states have the norm

$$\langle F^-_{s',j'}|F^-_{s,j}\rangle = \langle F^+_{s',j'}|F^+_{s,j}\rangle = \delta_{j,j'}\delta(s-s')\frac{E\pi}{s\sinh 2\pi s}. \qquad (25)$$



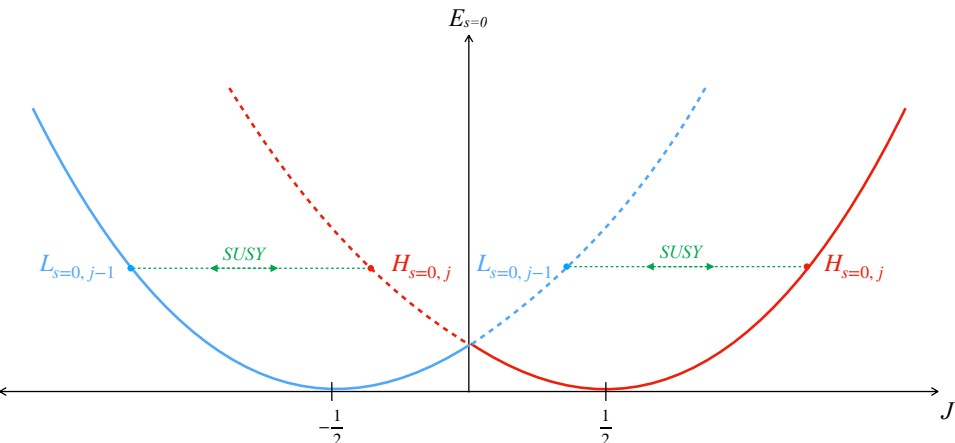

Figure 4: Lowest energy fermion even continuum states as a function of the $R$ charge $J$. Note that for $J > 0$ the lowest energy continuum state is $H_{s=0,j}$, with $J = j$. There is another state in the same multiplet, $L_{s=0,j}$ which has $J = j-1$. On the other hand, for negative $J$, the lowest energy state is $L_{s=0,j+1}$ with $J = j < 0$.

The zero energy states have norm

$$\langle Z_{j'}|Z_j\rangle = \delta_{j,j'}\frac{1}{\cos\pi j}, \quad |j| < \frac{1}{2}, \quad E = 0. \tag{26}$$

These are normalizable for $|j| < \frac{1}{2}$.

## 2.3 The two point function for BPS operators

### 2.3.1 Matrix elements of the charged operator

We will be interested in computing the matrix elements of the charged operator

$$e^{-\Delta(\ell+2ia)}, \tag{27}$$

which has $R$ charge $R_r = -R_l = -2\Delta$. This operator is BPS, in the sense that it commutes with some some of the supercharges, $[\bar{Q}_r, C] = [Q_l, C] = 0$. We can compute its matrix elements with all the members of the multiplet, see appendix E for details.

We will denote $E = s^2 + \frac{1}{4}(\frac{1}{2} - j)^2$ and $E' = s'^2 + \frac{1}{4}(\frac{1}{2} - j')^2$. Charge conservation gives $j'$ in terms of $j$, but note that the R charge of the $|L_{s,j}\rangle$ wavefunctions is $j-1$, not $j$, (18).

$$\langle H_{s',j'}|e^{-\Delta(\ell+2ia)}|H_{s,j}\rangle = \frac{E}{\pi}\frac{\Gamma(\Delta \pm is \pm is')}{\Gamma(2\Delta)}, \qquad j' = j - 2\Delta, \tag{28}$$

$$\langle L_{s',j'}|e^{-\Delta(\ell+2ia)}|L_{s,j}\rangle = \frac{E'}{\pi}\frac{\Gamma(\Delta \pm is \pm is')}{\Gamma(2\Delta)}, \qquad j' = j - 2\Delta, \tag{29}$$

$$\langle L_{s',j'}|e^{-\Delta(\ell+2ia)}|H_{s,j}\rangle = \frac{1}{\pi}\frac{\Gamma\left(\Delta + \frac{1}{2} \pm is \pm is'\right)}{\Gamma(2\Delta)}, \qquad j' = j + 1 - 2\Delta, \tag{30}$$

$$\langle H_{s',j'}|e^{-\Delta(\ell+2ia)}|L_{s,j}\rangle = 0, \qquad j' = j - 1 - 2\Delta, \tag{31}$$

$$\langle H_{s',j'}|e^{-\Delta(\ell+2ia)}|Z_j\rangle = 0, \qquad j' = j - 2\Delta, \tag{32}$$

$$\langle Z_{j'}|e^{-\Delta(\ell+2ia)}|H_{s,j}\rangle = \frac{1}{\pi}\frac{\Gamma\left(\Delta+\frac{1}{2}\pm\frac{2j'+1}{4}\pm is\right)}{\Gamma(2\Delta)}, \qquad j' = j-2\Delta, \qquad (33)$$

$$\langle Z_{j'}|e^{-\Delta(\ell+2ia)}|L_{s,j}\rangle = 0, \qquad j' = j-1-2\Delta, \qquad (34)$$

$$\langle L_{s',j'}|e^{-\Delta(\ell+2ia)}|Z_j\rangle = \frac{1}{\pi}\frac{\Gamma\left(\Delta+\frac{1}{2}\pm\frac{2j-1}{4}\pm is'\right)}{\Gamma(2\Delta)}, \qquad j' = j+1-2\Delta, \qquad (35)$$

$$\langle Z_{j'}|e^{-\Delta(\ell+2ia)}|Z_j\rangle = \frac{1}{\pi}\Gamma\left(\frac{1}{2}+j\right)\Gamma\left(2\Delta+\frac{1}{2}-j\right), \qquad j' = j-2\Delta. \qquad (36)$$

### 2.3.2 Assembling the two point function for BPS operators

In order to compute the two point function we first need to calculate the proper expression for the wormhole or thermofield double state. This is a combination of the various states discussed above. More specifically, we need to consider the zero fermion number states

$$|\text{TFD}\rangle_\beta = \sum_j\left[\mathcal{A}_{z,j}|Z_j\rangle + \int_0^\infty ds\,\mathcal{A}_{s,j}e^{-E_{s,j}\beta/2}\left(|H_{s,j}\rangle + |L_{s,j}\rangle\right)\right], \qquad (37)$$

where we anticipated that the two states in the multiplet will appear with the same coefficient in the thermofield double state. We determine the coefficient of each of these states by comparing to the computation of the partition function in [9, 16, 17] which is

$$Z(\beta) = e^{S_0}\left[\sum_{|j|<\frac{1}{2}}\cos\pi j + 2\sum_j\int_0^\infty ds\,\frac{s\sinh 2\pi s}{\pi E_{s,j}}e^{-\beta E_{s,j}}\right], \qquad E_{s,j} = s^2 + \frac{1}{4}\left(j-\frac{1}{2}\right)^2, \qquad (38)$$

where $\hat{q}j$ is an integer. The factor of two in front of the integral accounts for the two states, $|H_{s,j}\rangle$ and $|L_{s,j}\rangle$, in the multiplet. We then conclude that

$$\mathcal{A}_{s,j} = e^{\frac{S_0}{2}}\frac{s\sinh 2\pi s}{\pi E_{s,j}}, \qquad \mathcal{A}_{z,j} = e^{\frac{S_0}{2}}\cos\pi j. \qquad (39)$$

In principle, we could worry about a possible phase in the relative contribution of $|H_{s,j}\rangle$ and $|L_{s,j}\rangle$ in (37). The TFD obeys the condition

$$(Q_r - iQ_l)|\text{TFD}\rangle = 0. \qquad (40)$$

Up to signs, this equation follows from the supersymmetry of the propagator $e^{-\beta H}$; see Appendix I. We have not properly derived the minus sign, but we have checked that if we were to impose it with the opposite sign then the correlators we obtain would not have the expected positivity properties.[4]

The full two point function contains contributions where the continuum states are propagating and also where the zero energy states are propagating. We then have

$$\langle 2\,\text{pt}\rangle = \text{Tr}\left[e^{-u'H}O^\dagger e^{-uH}O\right] = {}_{u'}\langle\text{TFD}|e^{-\Delta(\ell+2ia)}|\text{TFD}\rangle_u = \langle 2\,\text{pt}\rangle_{c'c} + \langle 2\,\text{pt}\rangle_{z'c} + \langle 2\,\text{pt}\rangle_{c'z} + \langle 2\,\text{pt}\rangle_{z'z}. \qquad (41)$$

Each of these contributions can be computed using the explicit expression of the thermofield double state (37) (39) and the expressions of the matrix elements in (28)-(29). It has the explicit form

---

[4]By opposite sign, we mean the equation $(Q_r + iQ_l)|\text{TFD}\rangle = 0$. In this case, the sign of the second line in the $c'c$ contribution (42) would flip and the two point function would not have the right behavior at small $u$ through changes in the discussion leading to (63). The $i$ comes from the switch between the propagator interpretation and the TFD interpretation, a rotation by $\pi$ for a fermionic field.

$$\langle 2\,\mathrm{pt}\rangle_{c'c} = \frac{e^{S_0}}{\pi} \sum_{j,j'} \int \mathrm{d}s \int \mathrm{d}s'\, e^{-uE_{s,j}-u'E_{s',j'}} \frac{s\,\sinh(2\pi s)}{\pi E_{s,j}} \frac{s'\,\sinh(2\pi s')}{\pi E_{s',j'}} \left(E_{s,j}+E_{s',j'}\right)$$
$$\times \frac{\Gamma(\Delta \pm is \pm is')}{\Gamma(2\Delta)}\,\delta_{j',j-2\Delta}$$
$$+ \frac{e^{S_0}}{\pi} \sum_{r} \int \mathrm{d}s \int \mathrm{d}s'\, e^{-uE_{s,j}-u'E_{s',j'}} \frac{s\,\sinh(2\pi s)}{\pi E_{s,j}} \frac{s'\,\sinh(2\pi s')}{\pi E_{s',j'}}$$
$$\times \frac{\Gamma(\Delta+\frac{1}{2} \pm is \pm is')}{\Gamma(2\Delta)}\,\delta_{j',j-2\Delta+1}\,, \quad (42)$$

$$\langle 2\,\mathrm{pt}\rangle_{z'c} = \frac{e^{S_0}}{\pi} \sum_{|j'|<\frac{1}{2}} \cos(\pi j') \int \mathrm{d}s\, e^{-uE_{s,j}} \frac{s\,\sinh(2\pi s)}{\pi E_{s,j}} \frac{\Gamma(\Delta+\frac{1}{2} \pm \frac{2j'+1}{4} \pm is)}{\Gamma(2\Delta)}\,\delta_{j',j-2\Delta}\,, \quad (43)$$

$$\langle 2\,\mathrm{pt}\rangle_{c'z} = \frac{e^{S_0}}{\pi} \sum_{|j|<\frac{1}{2}} \cos(\pi j) \int \mathrm{d}s'\, e^{-u'E_{s',j'}} \frac{s'\,\sinh(2\pi s')}{\pi E_{s',j'}} \frac{\Gamma(\Delta+\frac{1}{2} \pm \frac{2j-1}{4} \pm is')}{\Gamma(2\Delta)}\,\delta_{j',j-2\Delta+1}\,, \quad (44)$$

$$\langle 2\,\mathrm{pt}\rangle_{z'z} = \frac{e^{S_0}}{\pi} \sum_{|j|,|j'|<\frac{1}{2}} \cos(\pi j)\cos(\pi j')\Gamma(\tfrac{1}{2}+j)\Gamma(2\Delta+\tfrac{1}{2}-j)\,\delta_{j',j-2\Delta}\,, \quad (45)$$

where the integrals over $s, s'$ are from zero to infinity. Here $\Gamma(\Delta \pm a \pm b) = \Gamma(\Delta + a + b) \times \Gamma(\Delta + a - b)\Gamma(\Delta - a + b)\Gamma(\Delta - a - b)$ indicates the product of the four possible terms.

## 2.4 The two point function for neutral operators

Here we repeat the above computation but now for the two point function of a non-BPS operator with zero $R$ symmetry charge. This involves computing the expectation value of $e^{-\Delta \ell}$. The steps are the same as above. We first compute the matrix elements and we then assemble the two point function. We now describe them in detail.

### 2.4.1 Matrix elements for the neutral operator

The matrix elements are given below, for details see appendix E,

$$\langle H_{s',j'}|e^{-\Delta \ell}|H_{s,j}\rangle = \frac{1}{2\pi}\left(E+E'+\Delta(\Delta+j-\tfrac{1}{2})\right)\frac{\Gamma(\Delta \pm is \pm is')}{\Gamma(2\Delta)}\,, \quad j'=j\,, \quad (46)$$

$$\langle L_{s',j'}|e^{-\Delta \ell}|L_{s,j}\rangle = \frac{1}{2\pi}\left(E+E'+\Delta(\Delta-j+\tfrac{1}{2})\right)\frac{\Gamma(\Delta \pm is \pm is')}{\Gamma(2\Delta)}\,, \quad j'=j\,, \quad (47)$$

$$\langle L_{s',j'}|e^{-\Delta \ell}|H_{s,j}\rangle = \frac{1}{2\pi}\frac{\Gamma(\Delta+\frac{1}{2} \pm is \pm is')}{\Gamma(2\Delta)}\,, \quad j'=j+1\,, \quad (48)$$

$$\langle H_{s',j'}|e^{-\Delta \ell}|L_{s,j}\rangle = \frac{1}{2\pi}\frac{\Gamma(\Delta+\frac{1}{2} \pm is \pm is')}{\Gamma(2\Delta)}\,, \quad j'=j-1\,, \quad (49)$$

$$\langle H_{s',j'}|e^{-\Delta \ell}|Z_{j}\rangle = \frac{1}{2\pi}\left(E'+\Delta(\Delta+j-\tfrac{1}{2})\right)\frac{\Gamma(\Delta \pm \frac{2j-1}{4} \pm is')}{\Gamma(2\Delta)}\,, \quad j'=j\,, \quad (50)$$

$$\langle Z_{j'}|e^{-\Delta\ell}|H_{s,j}\rangle = \frac{1}{2\pi}\left(E+\Delta(\Delta+j-\frac{1}{2})\right)\frac{\Gamma(\Delta\pm\frac{2j-1}{4}\pm is)}{\Gamma(2\Delta)}, \quad j'=j, \tag{51}$$

$$\langle Z_{j'}|e^{-\Delta\ell}|L_{s,j}\rangle = \frac{1}{2\pi}\frac{\Gamma(\Delta+\frac{1}{2}\pm is\pm\frac{2j'-1}{4})}{\Gamma(2\Delta)}, \quad j'=j-1, \tag{52}$$

$$\langle L_{s',j'}|e^{-\Delta\ell}|Z_{j}\rangle = \frac{1}{2\pi}\frac{\Gamma(\Delta+\frac{1}{2}\pm is'\pm\frac{2j-1}{4})}{\Gamma(2\Delta)}, \quad j'=j+1, \tag{53}$$

$$\langle Z_{j'}|e^{-\Delta\ell}|Z_{j}\rangle = \frac{1}{2\pi}\Delta\frac{\Gamma(\Delta)^2\Gamma(\Delta+\frac{1}{2}\pm j)}{\Gamma(2\Delta)}, \quad j'=j. \tag{54}$$

### 2.4.2 Assembling the two point function for neutral operators

Writing the two point function as in (41) we get

$$\langle 2\,\mathrm{pt}\rangle_{cc'} = \frac{e^{S_0}}{\pi}\left[\sum_j\int \mathrm{d}s\int \mathrm{d}s'e^{-E_{s,j}u-E_{s',j'}u'}\frac{s\sinh(2\pi s)}{\pi E_{s,j}}\frac{s'\sinh(2\pi s')}{\pi E_{s',j'}}\right. \tag{55}$$

$$\left.\times\left((E_{s,j}+E_{s',j'}+\Delta^2)\frac{\Gamma(\Delta\pm is\pm is')}{\Gamma(2\Delta)}\delta_{j',j}+\frac{\Gamma(\Delta+\frac{1}{2}\pm is\pm is')}{\Gamma(2\Delta)}\delta_{j',j+1}\right)\right], \tag{56}$$

$$\langle 2\,\mathrm{pt}\rangle_{zc'} = \frac{e^{S_0}}{\pi}\sum_{|j|<\frac{1}{2}}\cos\pi j\int \mathrm{d}s e^{-E_{s,j}u}\frac{s\sinh(2\pi s)}{\pi E_{s,j}}\frac{\Gamma(\Delta+\frac{1}{2}\pm\frac{2j+1}{4}\pm is)}{\Gamma(2\Delta)}, \tag{57}$$

$$\langle 2\,\mathrm{pt}\rangle_{cz'} = \frac{e^{S_0}}{\pi}\sum_{|j|<\frac{1}{2}}\cos\pi j\int \mathrm{d}s e^{-E_{s,j}u'}\frac{s\sinh(2\pi s)}{\pi E_{s,j}}\frac{\Gamma(\Delta+\frac{1}{2}\pm\frac{2j+1}{4}\pm is)}{\Gamma(2\Delta)}, \tag{58}$$

$$\langle 2\,\mathrm{pt}\rangle_{zz'} = \frac{e^{S_0}}{2\pi}\sum_{|j|<\frac{1}{2}}(\cos\pi j)^2\Delta\frac{\Gamma(\Delta)^2\Gamma(\Delta+\frac{1}{2}\pm j)}{\Gamma(2\Delta)}, \tag{59}$$

where the integrals over $s, s'$ are from zero to infinity.

## 2.5 Some limits and consistency checks

As a simple consistency check, we can see that if we set $\Delta = 0$, we recover the result for the partition function (38), with $\beta = u + u'$. This is expected since in that case we are simply inserting the identity operator. In order to check this, it is useful to note the identity

$$\lim_{\Delta\to 0}\frac{\Gamma(\Delta\pm is\pm is')}{\Gamma(2\Delta)} = \frac{\pi^2}{s\sinh 2\pi s}\delta(s-s'). \tag{60}$$

This identity is also useful for computing the norms of the continuum states in (24) (25) from the matrix elements.

We can now consider the short distance limit of the correlator. In this limit we keep $u'$ fixed and take $u \to 0$. This means that we will get contributions from large values of $s$. So we will need the large $s$ limits of the matrix elements, which can be easily obtained using Stirling's formula for the gamma function. For large $s$ we find

$$\Gamma(\Delta\pm is\pm is')\sim(2\pi)^2 s^{4\Delta-2}e^{-2\pi s}, \quad s\gg 1, \tag{61}$$

where $s'$ is kept fixed and can be real or imaginary. When we combine all factors we see that the $s$ dependence is of the form

$$\int\frac{\mathrm{d}s}{s}s^{4\Delta}e^{-s^2 u}\propto\frac{1}{u^{2\Delta}}, \quad \text{for} \quad u\ll 1. \tag{62}$$

Note that only the continuum contribution in the $u$ channel gives rise to this large result in the small $u$ limit. However, we should sum over both the continuum and zero energy contributions in the $u'$ channel. Assembling all constants we find that the two point function in this limit behaves as

$$\langle 2\,\mathrm{pt}\rangle = Z(u')\frac{1}{u^{2\Delta}}, \quad u \ll 1, \tag{63}$$

where $Z(u')$ is the partition function at temperature $\beta = u'$, (38). Indeed, this is the expected normalization from the correlator if we start from (3) and set $f \propto u$, see (A.7). Indeed, (63) is true both for the BPS and the charge neutral operator. These results are saying that we are normalizing the two point function of the operator to the standard $1/u^{2\Delta}$ behavior. However, we should remember that here we are defining $u$ in units of the Schwarzian coupling (6) so that, restoring the Schwarzian coupling, the result (63) becomes

$$\frac{\langle 2\,\mathrm{pt}\rangle}{Z} = \left(\frac{\phi_r}{\pi t}\right)^{2\Delta} = \left(\frac{2C}{t}\right)^{2\Delta}, \quad C = \frac{\phi_r}{2\pi}, \tag{64}$$

where $C$ is the coefficient of the Schwarzian term in the action, $I = -C \int \mathrm{d}t\{f,t\}$.

With this normalization, the long time, very low temperature limit of the correlator is simply given by the last line in (45) (59). We can calculate the correlator for each value of the $R$ charge $j$ on which the operator $\hat{O}$ is acting

$$\langle \hat{O}^\dagger \hat{O}\rangle_j = \frac{e^{S_0}}{\pi}\cos\pi(j-2\Delta)\cos\pi j\,\Gamma\left(\frac{1}{2}+j\right)\Gamma\left(\frac{1}{2}+2\Delta-j\right), \quad \text{charged, BPS,} \tag{65}$$

$$\langle \hat{O}\hat{O}\rangle_j = \frac{e^{S_0}}{\pi}\frac{1}{2}(\cos\pi j)^2\frac{\Delta\Gamma(\Delta)^2\Gamma(\Delta+\frac{1}{2}\pm j)}{\Gamma(2\Delta)}, \quad \text{neutral}. \tag{66}$$

In the special case that $\hat{q} = 1$, then there is only one BPS state in the Schwarzian description. Namely, only $j = 0$ is allowed. In that case a BPS operator, which has non-zero charge, vanishes after projecting onto the zero energy states. The uncharged neutral operators at low energies behave as

$$\langle \hat{O}\hat{O}\rangle = e^{S_0}2^{-4\Delta}\Gamma(1+2\Delta), \quad \text{for}: \quad \text{neutral}, \quad \hat{q}=1. \tag{67}$$

In the limit $u' \to \infty$, the partition function $Z(u')$ only has contribution from the zero energy states

$$\lim_{u'\to\infty} Z(u') = e^{S_0}\sum_{|j|<\frac{1}{2}}\cos(\pi j). \tag{68}$$

Note that with our definitions, the number of zero energy states is not exactly $e^{S_0}$ unless only $j = 0$ contributes. When $\hat{q} = 1$ the $R$ charges are integer and only $j = 0$ contributes.

We now present a plot of the two point function in figure 5. We consider the uncharged case, setting $j = j' = 0$ and dividing by the $j = 0$ contribution to the zero energy partition function. We take the $u' \to \infty$ and plot the answer as a function of $u$.

We can also consider the setting where $u = u' = \beta/2$. This is the thermal left right correlator (the two points at opposite points on the euclidean circle). The behavior is similar to the previous case. We show the uncharged case in figure 6 for $j = 0$. The small $u$ behavior can be obtained from the thermal partition function $\left[\frac{\beta}{\pi}\sin\frac{\tilde{u}\pi}{\beta}\right]^{-2\Delta}$ which becomes $\left(\frac{\pi}{\beta}\right)^{2\Delta}$ for $\tilde{u} = \beta/2$. The charged operator exhibits similar behavior.

As another comment, notice that from the expressions of the matrix elements we can compute the expectation value of the length $\ell$ in the various ground states. We obtain this by taking

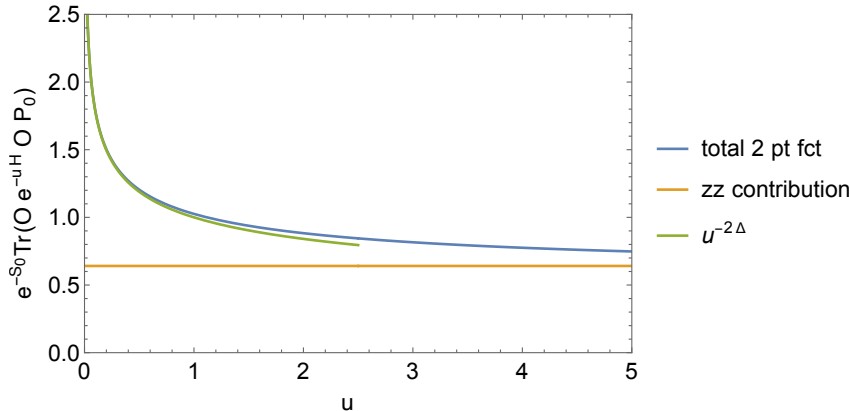

Figure 5: Neutral two point function for finite $u$ and $u' \to \infty$ in the charge $j = 0$ sector. We chose $\Delta = 1/8$. Note that the correlator behaves as $u^{-2\Delta}$ for $u \ll 1$ and it becomes constant for large $u$.

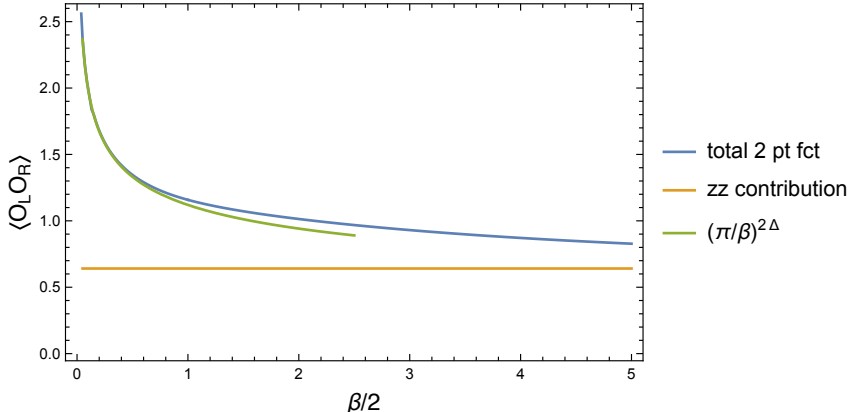

Figure 6: The neutral two-sided correlator evaluated in the zero-charge thermofield double state $\langle \text{TFD} | O_L O_R | \text{TFD} \rangle_{j=0} = \text{Tr}_{j=0}[O e^{-\beta H/2} O e^{-\beta H/2}]/Z_{j=0}(\beta)$ with $\Delta = \frac{1}{8}$. We also show the long time and short time approximations in orange and green.

(minus) the $\Delta$-derivative of the uncharged operator matrix elements (54) and then divide by the matrix elements. In other words, we compute

$$\langle \ell \rangle_j = - \partial_\Delta \log\left[ \langle Z_j | e^{-\Delta \ell} | Z_j \rangle \right]\big|_{\Delta=0} = -\psi\left(\frac{1}{2} + j\right) - \psi\left(\frac{1}{2} - j\right), \tag{69}$$

where $\psi(x) = \Gamma'(x)/\Gamma(x)$. This diverges as $|j| \to \frac{1}{2}$ which is when the state becomes non-normalizable.

## 2.6 The Lorentzian case

It is a straightforward matter to do the analytic continuation to Lorentzian time for the two point functions discussed above. To get the real time thermal two point function we set $u = it$, $u' = \beta - it$. This Lorentzian correlator also goes to a constant for very large Lorentzian times $t$. This is simply because all the finite energy contributions are oscillatory and average out while the zero energy contribution remains. One might wonder whether this means that if we perturb a black hole, then the perturbation will remain forever. This is a case where we should remember the famous adage "correlation is not causation". In fact, this constant value of the two point function is real and is the same as the Euclidean answer. On the other hand,

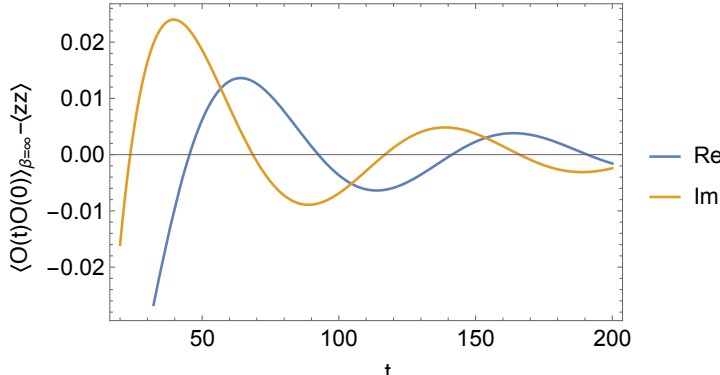

Figure 7: The Lorentzian two point function for a neutral correlator in the $\beta = \infty$ limit in the sector with $j = 0$. (We have divided by $e^{S_0}$). We have suppressed the constant $zz$ contribution, which is real and equals $\frac{\Gamma(5/4)}{\sqrt{2}} \sim 0.64$. The period of oscillation $\tau$ is related to the energy gap: $\tau = 2\pi/E_g \sim 100$, with $E_g = 1/16$ in this case.

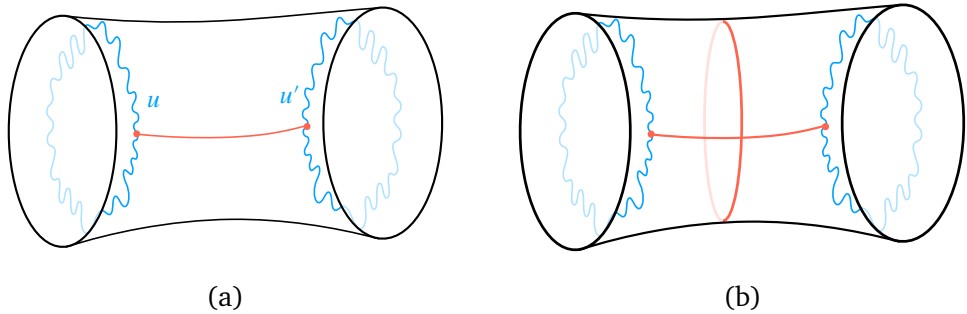

(a)          (b)

Figure 8: Contributions to the cylinder diagram. (a) Contribution due to a single particle path joining the two operator insertions. (b) Contributions with an additional disconnected path that wraps the cylinder. $u$ and $u'$ are the total euclidean times along the boundary of the left and right boundaries of the cylinder.

the response to a perturbation involves a commutator, given by the imaginary part of the correlator. In other words, imagine that we modify the unitary evolution of the system by adding a small perturbation of the form $e^{i \int \varepsilon(t)O(t)}$, with small $\varepsilon$. Then the change in expectation value of the operator $O(t')$ at a later time involves $\langle O(t') \rangle_\varepsilon = i \int \mathrm{d}t [O(t'), O(t)]\varepsilon(t)$. This commutator can be computed as the difference between two possible analytic continuations $u = \pm i(t - t')$, $u' = \beta \mp i(t - t')$, and this in turn picks out the imaginary part of the two point function. Since the imaginary part goes to zero, it means that physical perturbations are indeed forgotten by the black hole at long times. They are simply acting as a unitary operation on the ground states. Since we are summing over these states, we do not notice the action of the unitary by looking only at a one sided correlator. Of course, this discussion is fairly standard, and we are simply applying it in the context of the ground states.

## 2.7 Two point function on the cylinder

By a similar method it is possible to compute one contribution to the cylinder two point function. This is the contribution where we sum over paths that connect the two operator insertions, as in [27], see figure 8a. This neglects other possible contributions which include any number of additional particles that wrap the cylinder, see figure 8b. These additional contributions can be neglected in situations where the minimal cross section of the cylinder has a

relatively large length. We expect this to be the case when $\Delta \gg 1$, or for large Lorentzian times, as in [28]. In section (5.4) we will discuss another application of these wormholes which seems reliable for similar reasons. These neglected contributions could lead to divergences and will be discussed in more detail in [29].

For the contribution of the diagram in figure 8a, we simply trace over all states of the Liouville theory so that we have an expression of the form

$$\langle 2\,\text{pt}\rangle_{cyl} = \text{Tr}_{\text{Liou}}[e^{-\Delta\ell}e^{-(u+u')E}] = \overline{\text{Tr}[Oe^{-uH}]\,\text{Tr}[Oe^{-u'H}]}, \tag{70}$$

where the trace in the first expression is over the states of the Liouville theory that we explicitly described above. The last expression in (70) is giving the interpretation of this computation in a hypothetical quantum mechanical dual where the trace is over microstates and the overline denotes an average over couplings. In this case we get contributions also from the wavefunctions with odd fermion number, namely $|F_{s,j}^{\pm}\rangle$. We only get a non-zero contribution when we consider a neutral operator. For a charged operator the conservation of the $U(1)_R$ charge implies that the cylinder two point function is zero.

### 2.7.1 Matrix elements

Since we are computing a trace over the Liouville Hilbert space, we only need diagonal matrix elements of the operator. In the continuum case, there are four of them, for the four states of each energy multiplet. The first two are $\langle H_{s',j'}|e^{-\Delta\ell}|H_{s,j}\rangle$ and $\langle L_{s',j'}|e^{-\Delta\ell}|L_{s,j}\rangle$. We now list the other two, derived in appendix E,

$$\langle F_{s',j'}^+|e^{-\Delta\ell}|F_{s,j}^+\rangle = \langle F_{s',j'}^-|e^{-\Delta\ell}|F_{s,j}^-\rangle = \frac{\sqrt{E\,E'}}{\pi}\frac{\Gamma(\Delta\pm is\pm is')}{\Gamma(2\Delta)}. \tag{71}$$

We will also use $\langle Z_{j'}|e^{-\Delta\ell}|Z_j\rangle$, which was computed above in (54).

### 2.7.2 Assembling the two point function

The two-point function for the cylinder is a sum of the continuous contributions and the discrete term. For each of these terms we should compute the diagonal matrix elements from (46), (47), (71) and (54). Since the states are not unit normalized, we should remember to divide by their norms (24-26). For the zero energy contribution we get

$$\langle 2\,\text{pt}\rangle_{cyl,z} = \frac{1}{2\pi}\sum_{|j|<\frac{1}{2}}\cos\pi j\,\frac{\Delta\Gamma(\Delta)^2\Gamma(\Delta+\frac{1}{2}\pm j)}{\Gamma(2\Delta)}, \tag{72}$$

where the factor of $\cos\pi j$ comes from the inverse of the norm in (26) and the rest from the matrix element in (54). We can similarly get the continuum contribution which involves the sum over the four states in the multiplet. We then write the answer as the sum of the two terms

$$\begin{aligned}
\langle 2\,\text{pt}\rangle_{cyl} &= \langle 2\,\text{pt}\rangle_{cyl,c} + \langle 2\,\text{pt}\rangle_{cyl,z} \\
&= \sum_j \int_0^\infty ds\,\frac{s\sinh 2\pi s}{\pi^2 E_{s,j}}e^{-E_{s,j}(u+u')}(4E+\Delta^2)\frac{\Gamma(\Delta)^2\Gamma(\Delta\pm 2is)}{\Gamma(2\Delta)} \\
&\quad + \sum_{|j|<\frac{1}{2}}\frac{\cos\pi j}{2\pi}\frac{\Delta\Gamma(\Delta)^2\Gamma(\Delta+\frac{1}{2}\pm j)}{\Gamma(2\Delta)}.
\end{aligned} \tag{73}$$

As we have already stressed, we expect this answer to be reliable only when $\Delta \gg 1$.

# 3 Correlation functions in $\mathcal{N}=2$ supersymmetric SYK

In this section we consider the $\mathcal{N}=2$ supersymmetric SYK model introduced in [12] whose conventions we use. The model involves $N$ complex fermions and a supercharge

$$Q = i \sum_{1 \leq i < j < k \leq N} C_{ijk} \psi^i \psi^j \psi^k, \quad \{\psi^i, \bar{\psi}_j\} = \delta^i_j, \tag{74}$$

and a supercharge $\bar{Q} = Q^\dagger$. These anticommutation relations imply $Q^2 = \bar{Q}^2 = 0$. The complex numbers $C_{ijk}$ are gaussian with a second moment given by

$$\langle C_{ijk} \bar{C}^{ijk} \rangle = \frac{2J}{N}, \quad \text{(no sum)}, \tag{75}$$

and the Hamiltonian is defined to be $H = \{Q, \bar{Q}\}$. The model also has a generalization where the supercharge is given in terms of $\hat{q}$ fermion terms. In (74) we have $\hat{q} = 3$. The model has an $R$ charge with $\hat{q} = 3$, meaning that the supercharge has charge one, but the elementary fields have charge 1/3. At large $N$ and for low, but not too low, energies the model develops a nearly conformal regime. In that regime the fermion correlation function has a behavior

$$\langle \bar{\psi}^i(0) \psi^i(t) \rangle = b_{\hat{q}} \frac{1}{(Jt)^{2\Delta}}, \quad b_{\hat{q}} = \left[ \frac{\tan \frac{\pi}{2\hat{q}}}{2\pi} \right]^{\frac{1}{\hat{q}}}, \quad \Delta = \frac{1}{2\hat{q}}. \tag{76}$$

As we get to even lower energies, it is necessary to take into account the quantum mechanics of a super-Schwarzian mode with action

$$S = -C \int dt \{f, t\} + \text{susy partners}, \quad C = \frac{\alpha_S N}{J}. \tag{77}$$

In the case of $\hat{q} = 3$, it was numerically found that [30][5]

$$\alpha_S = 0.00842\ldots. \tag{78}$$

One of the non-trivial features that we obtain from this analysis is the presence of an energy gap of the form [9]

$$8C E_{\text{gap}} = \left( j - \frac{1}{2} \right)^2, \tag{79}$$

with $C$ as in (77). Here we have multiplied (14) by a factor of $\frac{1}{2C}$ to restore the units. In [1] we have checked this prediction numerically for $N = 16$ using the large $N$ prediction for $C$ given by (77) (78).

We can use these numbers together with the results of section 2 to derive the expected form for the two point functions. It is convenient to separate the ground states according to their $R$ charges. In this case, these have $R$ charges $j = 0, \pm 1/3$, for $N$ even. We will only treat the $N$ even case.[6] The total number of BPS states is

$$\text{Tr}[P] = N_{BPS} = e^{S_0} \hat{L}, \quad \hat{L} \equiv \sum_{|j| \leq \frac{1}{2}} \cos \pi j, \quad P = \sum_j P_j, \tag{80}$$

where $P_j$ is the projector to zero energy states of R charge $j$. We will not need the value of $S_0$ for the comparisons we will do here. These degeneracies were already discussed in [12]. Note

---

[5]Curiously this is somewhat close to the large $\hat{q}$ answer $\alpha_S = \frac{J}{4\hat{q}^2 \mathcal{J}}$ extrapolated to $\hat{q} = 3$, after using $\mathcal{J} = 3J/2$, which gives $\alpha_s = 0.0092\ldots$

[6]When $N$ is odd the $R$ charges are all shifted by a $1/(2\hat{q})$ additive constant.

Table 1: Comparison of exact diagonalization results for the 2-pt function with the Schwarzian predictions. Here we use the large $N$ value for the Schwarzian, $C = 0.00842N = 0.135$. The $R$-charge corresponds to the $R$-charge of the state, see equations (81)-(83). The error bar we display is the statistical error, computed by changing the value of $i$. This is related to the error we should get if we vary the coupling constants.

| Operator | $R$-charge | Schwarzian prediction | Numerical answer ($N$=16) |
|---|---|---|---|
| $\psi_i$ | 0 | 0.111 | $0.110 \pm 0.005$ |
| | +1/3 | 0.111 | $0.110 \pm 0.005$ |
| $\psi_i \psi_j$ | +1/3 | 0.0247 | $0.024 \pm 0.003$ |
| $\bar{\psi}_i \psi_j$ | −1/3 | 0.0282 | $0.027 \pm 0.001$ |
| | 0 | 0.0874 | $0.079 \pm 0.001$ |
| | +1/3 | 0.0282 | $0.027 \pm 0.001$ |

that the operator $\psi$ anticommutes with $Q$, $[Q, \psi] = 0$. This implies that it is a BPS operator in the conformal regime. Then we are interested in computing the correlators

$$\frac{\langle \psi^i \bar{\psi}_i \rangle_j}{N_{BPS}} = \frac{\text{Tr}\left[P_j \psi^i P_{j-\frac{1}{3}} \bar{\psi}_i P_j\right]}{\text{Tr}[P]} = \left[\frac{\tan \frac{\pi}{2\hat{q}}}{2\pi}\right]^{\frac{1}{\hat{q}}} \frac{1}{(2\alpha_S N)^{2\Delta}} \frac{\cos \pi j \cos \pi(j-2\Delta)}{\pi \hat{L}} \Gamma(\frac{1}{2} + j)\Gamma(2\Delta + \frac{1}{2} - j), \quad (81)$$

where $\hat{q} = 3$, $\Delta = 1/6$. The second expression is an explicit expression for the operator in the SYK model that uses the projectors $P_j$ onto the ground states of specific $R$ charge. In this formula, we have taken the zero energy matrix element in (45) and we have multiplied by two factors to take into account the proper normalization of the operators, the first from (76) and the second from (64) using (77). In a similar way we can write the expressions for an operator with twice the charge, obtained by taking $\psi^i \psi^k$.

$$\frac{\langle (\psi^i \psi^k)(\bar{\psi}^k \bar{\psi}^i) \rangle_j}{N_{BPS}} = \frac{Tr[P_j(\psi^i \psi^k)P_{j-\frac{2}{3}}(\bar{\psi}^k \bar{\psi}^i)P_j]}{Tr[P]}$$

$$= \left[\frac{\tan \frac{\pi}{2\hat{q}}}{2\pi}\right]^{\frac{2}{\hat{q}}} \frac{1}{(2\alpha_S N)^{2\Delta}} \frac{\cos(\pi j)\cos(\pi(j-2\Delta))}{\pi \hat{L}} \Gamma\left(\frac{1}{2} + j\right)\Gamma\left(2\Delta + \frac{1}{2} - j\right), \quad \Delta = 2/6. \tag{82}$$

Finally, we can consider an uncharged operator $\psi^i \bar{\psi}^k$ to obtain

$$\langle (\psi^k \bar{\psi}^i)(\psi^i \bar{\psi}^k) \rangle_j = \left[\frac{\tan \frac{\pi}{2\hat{q}}}{2\pi}\right]^{\frac{2}{\hat{q}}} \frac{1}{(2\alpha_S N)^{2\Delta}} \frac{\cos^2(\pi j)}{\pi \hat{L}} \frac{1}{2} \frac{\Delta \Gamma(\Delta)^2}{\Gamma(2\Delta)} \Gamma(\Delta + \frac{1}{2} \pm j), \quad \Delta = 2/6. \tag{83}$$

Inserting the value of $\alpha_S$ in (78) and setting $N = 16$ we obtain the numbers in the "Schwarzian prediction" column in table 1. In that table, this is compared to the numerical answers for $N = 16$.

As an aside, let us mention one point that can cause confusions in interpreting some of the numerical computations. This is the fact that some operators have extra zero modes which are not expected for larger values of $N$, see appendix G.

There are also some curious features we initially found numerically but which can be explained in terms of the symmetries. The first observation is that

$$P\psi^i e^{-uH} \psi^j P, \tag{84}$$

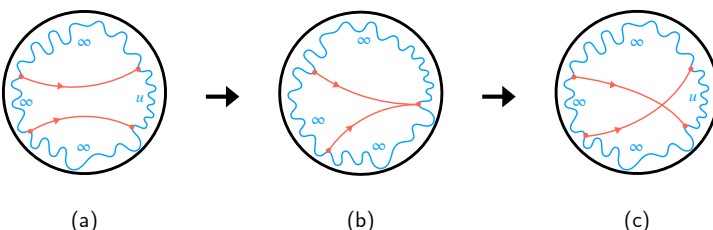

(a)  (b)  (c)

Figure 9: Here we see the operation where we move the insertion of the two chiral operators on the right until they cross each other. Since their anticommutator vanishes in the UV this relates the values of ($a$) and ($c$) up to a sign, both of which are $u$ independent thanks to the $u$ independence of (84).

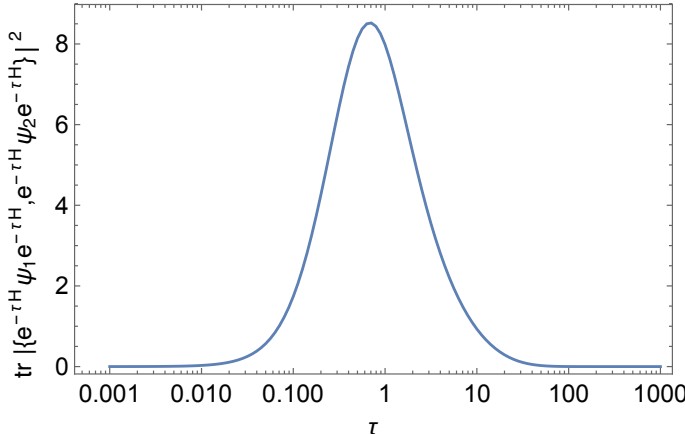

Figure 10: Anti-commutator of the chiral operators $e^{-\tau H}\psi e^{-\tau H}$. At short times, the anti-commutator is just given by the UV commutator $\{\psi_i, \psi_j\} = 0$. At intermediate times, the commutator is something complicated, but at long times the commutator again vanishes since we are projecting to BPS states. Here the results are for $N = 10$ complex fermions.

is independent of $u$. Here $P$ is the projector onto zero energy states. The reason is that the $u$ derivative brings down the Hamiltonian $H$. Writing the Hamiltonian as $H = \{Q, \bar{Q}\}$ and anticommuting $Q$ either with the left $\psi$ or the right $\psi$ we get a $Q$ acting as $QP = 0$ or $PQ = 0$.

This implies that the long time TOC and OTOC are equal (up to a sign). This is because the property (84) implies that we can bring $\psi^i$ and $\psi^j$ close to each other as long as other times are infinite, exchange their order, and then bring them far from each other, all without changing the value of the correlator. See Figure 9.

This means that the infrared operators $\hat{\psi}^i$ and $\hat{\psi}^j$ anticommute. This is nontrivial. Of course the UV operators anticommute $\{\psi^i, \psi^j\} = 0$, but if we start projecting them onto lower energy states, constructing $\psi_\tau^i = e^{-\tau H}\psi^i e^{-\tau H}$, then they do not anticommute for intermediate $\tau$, but they anticommute again as $\tau \to \infty$. We have computed this anticommutator numerically and the results are in Figure 10.

This also has another interesting implication. We can consider a state produced by euclidean evolution by inserting two $\bar{\psi}$ operators, and then evolving by an infinite amount of Euclidean time. This produces some kind of wormhole with two particles. One question we can ask is whether these two particles are on top of each other or whether they are separated from each other. We can address this question by computing the overlap created by that state and a state with a single conformal operator insertion which is a two particle operator of the schematic form $\partial^n \psi^i \psi^j$ where the derivatives are distributed between the two factors so as

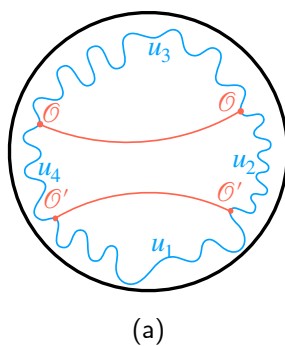 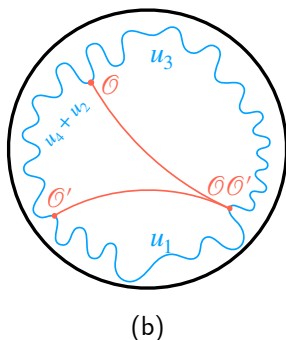

|  (a) | (b) |

Figure 11: We consider the four point correlator in a time ordered configuration. It turns out that the answer depends only on $u_4 + u_2$ and not on $u_2$ and $u_4$ separately. This means that the answer is equal to the three point correlator in $(b)$ where the two operators are coincident.

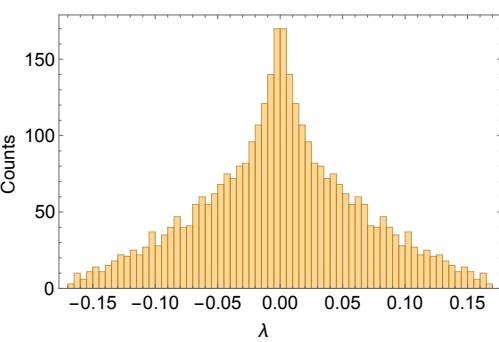 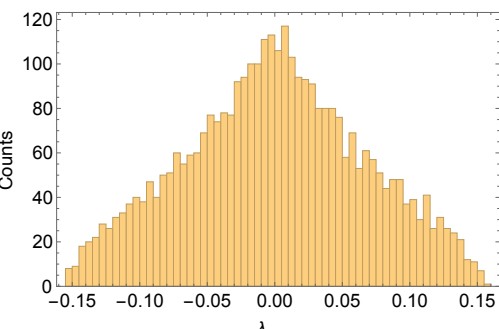

Figure 12: *Left*: Eigenvalue distribution of the operator $\frac{1}{2}P(\psi_1\bar{\psi}_2 + \psi_2\bar{\psi}_1)P$, where $P = e^{-\infty H}$ projects onto the ground states. We removed the zero eigenvalues from this distribution. Results were obtained using $N = 16$ SYK. *Right*: a histogram of eigenvalues obtained by a toy model where $\psi_1, \psi_2$ are 3432×4374 complex Gaussian matrices. In the limit where we take $n \times m$ matrices with $m \gg n$ we would expect a rounder distribution.

to make a conformal primary. We can produce these operators by taking them to finite $u$ as in Figure 9 and then taking derivatives. This shows that all the overlaps are zero. This means that the two particles are on top of each other. We will also give an independent argument in Section 5.2.

There is another relation that should hold for the TOC. This is based on the point that the time ordered operator depends only on $u_1, u_2 + u_4, u_3$, see figure 11. This holds at any time, not just long times. This then implies an equality between the TOC four point function and a three point function that involves a two particle operator $\tilde{O} = OO'$. See figure 11. Setting $u_2 = 0$ and taking all the other times to infinity we get an identity that should hold among zero energy correlators. Numerically we find that this identity holds within a factor of 4%.

The UV operators $\psi^i$ are very simple and have simple eigenvalues. Actually, to talk about eigenvalues it is more convenient to consider an operator of the form $A = \bar{\psi}^i\psi^j + \bar{\psi}^j\psi^i$ which is hermitian. We go to the IR and we define the projected operator

$$\hat{A} = PAP, \quad \text{with} \quad A = \bar{\psi}^i\psi^j + \bar{\psi}^j\psi^i. \tag{85}$$

We can think of $\psi^i P$ as a rectangular random matrix with the short side of the rectangle determined by the IR dimension of the Hilbert space, $e^{S_0}$, and the long side equal to the UV dimension. Then we expect that the eigenvalues of $\hat{A}$ will be those of a random matrix. We

have computed this explicitly for the $N = 16$ case and we find indeed a dense spectrum for the operator, but the overall distribution is not quite a semi-circle law as we expected. The reason is that because $N$ is small the dimensions of the Hilbert spaces involved in the computation are actually such that the relevant UV dimensions are actually smaller than the relevant IR dimensions, due to the fact that $N = 16$ is relatively small and that the operator $\psi$ changes the R charge, see appendix G. So for comparison, in figure 12, we have made the approximation that the $\psi$'s are random matrices between the spaces with the dimensions discussed in appendix G.

It is also possible to consider the states that result from adding an operator $P_j A P_j$, see figure 13, and numerically compute the entropy. We find a reduction of the entropy, relative to the case with no operator of the form, $S_j - S_{\hat{A}} = \log(4374) - 7.64... = 0.74$. (The value quoted is for $j = 0$ and $A \propto \psi_1 \bar{\psi}_2 + \psi_2 \bar{\psi}_1$). This value is close to the theoretical prediction for an operator with small dimension $\Delta \approx 0$, which is $S - S_0 = 0.7296...$ as we will discuss around (167). However, this agreement should not be taken too seriously since we also found a substantial number of zero modes which contributes significantly to the entropy reduction. Such zero modes are not present at $N \geq 20$, see appendix G. (There are 942 additional zero modes, which would account for a reduction in entropy of about 0.24.)

# 4  Calculation of the propagator

The goal of this section is to calculate the propagator for the boundary particle, described by the $\mathcal{N} = 2$ super-Schwarzian theory. We want to find it to be able to write integral expressions for general correlators. We will focus on finding the propagator for the zero energy states. We expect that the same method should work for the propagator at arbitrary energies.

## 4.1  Review of the group theory approach for $\mathcal{N} = 0$

In this section, we will review the derivation of the propagator in the $\mathcal{N} = 0$ Schwarzian theory. We will do so using a parameterization that will be convenient for generalizing to the $\mathcal{N} = 2$ case. In the Appendix C, we also give some formulas for the $\mathcal{N} = 1$ case.

AdS$_2$ can be viewed as the coset $SL(2, \mathbb{R})/U(1)$. We can think of this space as the set of $SL(2)$ two by two matrices with the further identification $g \sim gh$, where $h$ is an element of the $U(1)$ subgroup. If we had a non-relativistic quantum particle moving on the $SL(2)$ group manifold, we would have generic functions of $g$. These functions can be acted upon by a set of two $SL(2)$ symmetries $g \rightarrow s_L g s_R$. The quotient by $U(1)$ breaks the right $SL(2)$ subgroup. If we require that the functions are invariant under this $U(1)$ group, we get the space of functions on AdS$_2$. For our problem we need two modifications. First we will require that the functions have a definite charge $\tilde{q}$ under the $U(1)$. If we choose this $U(1)$ to generate a rotation, then we would get functions on AdS$_2$ that carry an "electric" charge $\tilde{q}$. The second modification is that we need to take the Schwarzian limit which involves taking the $\tilde{q} \rightarrow \infty$ limit and a similar limit of the interior coordinates, see [10, 11]. It is possible to perform both of these steps at once by picking the quotient subgroup to be generated by a null generator $L_+$ with eigenvalue q, which is now finite. In this case, we do not need to take any limits and we land on the Schwarzian theory directly [31].

In the rest of this subsection we review these steps more explicitly as a prelude to the supersymmetric case.

The algebra of $SL(2)$ is

$$[\mathcal{L}_m, \mathcal{L}_n] = (m-n)\mathcal{L}_{m+n}. \tag{86}$$

We can realize this algebra in the following way. We start by considering a parameterization

of the group $SL(2)$

$$g = e^{-xL_-}e^{\rho L_0}e^{\gamma L_+}\,, \tag{87}$$

in terms of abstract[7] generators $L_\pm$, $L_0$ which we define to obey the "opposite" $SL(2)$ algebra:[8]

$$[L_m, L_n] = -(m-n)L_{m+n}\,. \tag{88}$$

Note that there is a minus sign on the RHS of (86), in contrast to the more usual definition of the algebra.

We can define now the left acting symmetry by multiplying (87) on the left by the generators $L_a$ and then rewriting the answer in terms of an infinitesimal change of the coordinates $x$, $\rho$, $\gamma$. In this way we find concrete expressions for the generators:

$$
\begin{aligned}
\mathcal{L}_- &= -\partial_x\,, \\
\mathcal{L}_0 &= -x\partial_x + \partial_\rho\,, \\
\mathcal{L}_+ &= -x^2\partial_x + 2x\partial_\rho + e^{-\rho}\partial_\gamma\,,
\end{aligned}
\tag{89}
$$

which also obey the algebra (86) when acting on functions of $x$, $\rho$, $\gamma$. The reason for the switch in sign between (88) and (86) is that we defined things so that $\mathcal{L}_a g = L_a g$. This then means that $\mathcal{L}_b\mathcal{L}_a g = \mathcal{L}_b(L_a g) = L_a\mathcal{L}_b g = L_a(L_b g)$ where in the third equality we used that the $L_a$ and $\mathcal{L}_b$ commute with each other. So the difference between the abstract and concrete generators is that they act in the opposite order. Therefore, a commutator $[L_a, L_b]$ becomes minus a commutator of $[\mathcal{L}_a, \mathcal{L}_b]$. We will only consider the generators $\mathcal{L}_a$ from now on.

In addition we can define another set of generators that come from acting on (87) with $L_a$ on the right.

$$
\begin{aligned}
\mathcal{L}_-^R &= \gamma^2\partial_\gamma - 2\gamma\partial_\rho - e^{-\rho}\partial_x\,, \\
\mathcal{L}_0^R &= \partial_\rho - \gamma\partial_\gamma\,, \\
\mathcal{L}_+^R &= \partial_\gamma\,, \\
[\mathcal{L}_m^R, \mathcal{L}_n^R] &= -(m-n)\mathcal{L}_{m+n}^R\,.
\end{aligned}
\tag{90}
$$

The minus sign in the algebra, relative to (86) is due to the fact that we are defining the generators as acting on the right. These right generators commute with the left ones (89).

The Casimir, which commutes with both left and right generators, is

$$\mathcal{C} = \mathcal{L}_0^2 - \frac{1}{2}(\mathcal{L}_+\mathcal{L}_- + \mathcal{L}_-\mathcal{L}_+) = \partial_\rho^2 + \partial_\rho + e^{-\rho}\partial_x\partial_\gamma\,, \tag{91}$$

and the Hamiltonian of our system is $H = -\mathcal{C} - \frac{1}{4}$, where the 1/4 is introduced just for convenience as an overall energy shift. When we perform the quotient, we select the generator $\mathcal{L}_+^R$ and we demand that the states obey the condition that $\mathcal{L}_+^R = -\mathsf{q}$. The minus sign is just a convention.

This is equivalent to demanding that the only dependence of the wavefunctions on $\gamma$ is of the form $\psi \sim e^{-\mathsf{q}\gamma}$.

With this quotient, we can identify $(x, \rho)$ with a rescaled version of the AdS coordinates

$$ds^2 = \frac{dx^2 + d\tilde{z}^2}{\tilde{z}^2} = d\rho^2 + \frac{1}{\epsilon^2}e^{2\rho}dx^2\,, \quad \tilde{z} = \epsilon e^{-\rho}\,, \tag{92}$$

where the rescaling, with $\epsilon \to 0$ is related to the fact that we are taking the Schwarzian limit, where the boundary is far away.

---

[7]It is also possible to choose a two dimensional representation for the generators in terms of Pauli matrices and write $g$ as an explicit two by two matrix. We do not need to do that in what follows.

[8]We thank G. Penington for pointing out an error in the previous version of the draft.

We can then proceed to compute the propagator. We consider functions of two points, characterized by two sets of coordinates as above, with a wavefunction of the form

$$P \sim e^{-\mathfrak{q}(\gamma_1-\gamma_2)}F(x_1,\rho_1;x_2,\rho_2;u), \quad \partial_u P = -HP. \tag{93}$$

Notice that the second particle has the opposite eigenvalue under the right generator, $\mathcal{L}_{1+}^R = -\mathcal{L}_{2+}^R = -\mathfrak{q}$. In particular, we demand that it is invariant under the sum of the left generators acting on the two coordinates $(\mathcal{L}_{1a} + \mathcal{L}_{2a})P = 0$. This implies that $P$ is a function of two invariants only:

$$\mathfrak{p} = \gamma_1 - \gamma_2 + \varphi_{12}, \quad \varphi_{12} = \frac{e^{-\rho_1}+e^{-\rho_2}}{x_1-x_2}, \quad e^\ell = e^{\rho_1+\rho_2}(x_1-x_2)^2. \tag{94}$$

Of course, since (93) has a specific dependence on $\gamma_{12}$, then the dependence on the first invariant is fixed. The dependence on the second invariant can be decomposed in terms of eigenfunctions of the Hamiltonian. Then we end up with an expression of the form [10, 11]

$$P = e^{-(\gamma_1-\gamma_2+\varphi_{12})} \int_0^\infty \mathrm{d}E \rho(E) e^{-Eu} e^{-\ell/2} K_{2i\sqrt{E}}(2e^{-\ell/2}), \tag{95}$$

after we have set $\mathfrak{q} = 1$. The function $\rho(E)$ can be determined by demanding that the propagator obeys the composition law $\int \mathrm{d}x_2 d\rho_2 e^{\rho_2} P(1,2;u) P(2,3;u') = P(1,3;u+u')$. This argument gives $\rho(E) \propto \sinh 2\pi\sqrt{E}$ [10, 11], see also Appendix I.

### 4.1.1 Connection with Liouville quantum mechanics

We can connect this discussion with the wavefunctions of Liouville quantum mechanics.

To evaluate the expectation value of a function of the distance we would need to integrate that function against the product $P(1,2)P(2,1)$ over each of the points and divide by the volume of $SL(2)$. Note that in this product of two propagators, the exponential prefactors in (95) cancel out. Dividing by the volume of $SL(2)$ can be achieved by fixing a gauge, such as

$$x_1 = 1, \quad x_2 = 0, \quad \rho_2 = 0, \quad \Rightarrow \quad e^\ell = e^{\rho_1}. \tag{96}$$

The Fadeev Popov determinant is trivial in this gauge. Then we find that the factor of $e^{\rho_1} = e^\ell$ in the measure of integration over the first point cancels out against the two prefactors of $e^{-\ell/2}$ in the Bessel functions in (95). This leaves just the simple measure $\int \mathrm{d}\ell$, integrating the Bessel functions $\psi_E(\ell) = K_{2i\sqrt{E}}(2e^{-\ell/2})$ which are the ones appearing in the Liouville approach as eigenfunctions of the Liouville Hamiltonian

$$H = -\partial_\ell^2 + e^{-\ell}. \tag{97}$$

More explicitly, we can consider a wavefunction of two points that is only a function of the invariants $e^{-\hat{q}(\gamma_1-\gamma_2+\varphi_{12})}F(\ell)$. Then we may write the Casimir acting on the first point as

$$\mathcal{C}_1 = \partial_\ell^2 + \partial_\ell - \mathfrak{q}e^{-\ell}. \tag{98}$$

Now notice that if we define $P = e^{-(\gamma_1-\gamma_2+\varphi_{12})}e^{-\ell/2}\psi_{\text{Liouville}}(\ell)$, we get

$$(-\mathcal{C}_1 - \frac{1}{4})P = e^{-(\gamma_1-\gamma_2+\varphi_{12})}e^{-\ell/2}(-\partial_\ell^2 + e^{-\ell})\psi_{\text{Liouville}}(\ell) = HP, \tag{99}$$

where in the last line we substituted in a Liouville eigenfunction $\psi_{\text{Liouville}}(\ell) = \psi_E(\ell)$.

We now will repeat all these steps for the $\mathcal{N} = 2$ supersymmetric case. The $\mathcal{N} = 1$ case is discussed in appendix C.

## 4.2 The $\mathcal{N} = 2$ supergroup and its symmetry generators

The $\mathcal{N} = 2$ superconformal algebra is given by (86) and

$$
\begin{aligned}
&[\mathcal{L}_m, \mathcal{G}_r] = \left(\frac{m}{2} - r\right)\mathcal{G}_{m+r}\,, \quad [\mathcal{L}_m, \bar{\mathcal{G}}_r] = \left(\frac{m}{2} - r\right)\bar{\mathcal{G}}_{m+r}\,, \\
&[\mathcal{L}_m, \mathcal{J}] = 0\,, \\
&\{\mathcal{G}_r, \bar{\mathcal{G}}_s\} = 2\mathcal{L}_{r+s} + (r-s)\mathcal{J}\delta_{r+s}\,, \\
&\{\mathcal{G}_r, \mathcal{G}_s\} = \{\bar{\mathcal{G}}_r, \bar{\mathcal{G}}_s\} = 0\,, \\
&[\mathcal{J}, \mathcal{G}_r] = \mathcal{G}_r\,, \quad [\mathcal{J}, \bar{\mathcal{G}}_r] = -\bar{\mathcal{G}}_r\,,
\end{aligned} \tag{100}
$$

with $m = 1, 0, -1$ and $r, s = \pm\frac{1}{2}$. This is also known as the superalgebra $SU(1,1|1)$ or $OSp(2|2)$. We can write a supergroup element as

$$
g = e^{-xL_-}e^{\theta_- G_- + \bar{\theta}_- \bar{G}_-}e^{\rho L_0}e^{\theta_+ G_+ + \bar{\theta}_+ \bar{G}_+}e^{\gamma L_+}e^{iaJ}\,. \tag{101}
$$

As before, we can define the left generators by multiplying $g$ on the left and rewriting the result as a differential operator acting from the left side. As with the bosonic case (86), the abstract generators appearing in (101) obey the *opposite* superalgebra compared to the left generators. Here "opposite" means that we add and extra minus sign to the right hand side relative to (100), both for commutators and anticommutators.

The left generators of the group are

$$
\begin{aligned}
&\mathcal{J} = -i\partial_a + \bar{\theta}_- \partial_{\bar{\theta}_-} - \theta_- \partial_{\theta_-} + \bar{\theta}_+ \partial_{\bar{\theta}_+} - \theta_+ \partial_{\theta_+}\,, \\
&\mathcal{L}_- = -\partial_x\,, \\
&\mathcal{L}_0 = -x\partial_x + \partial_\rho - \frac{1}{2}\left(\bar{\theta}_- \partial_{\bar{\theta}_-} + \theta_- \partial_{\theta_-}\right)\,, \\
&\mathcal{L}_+ = e^{-\rho}\partial_\gamma - x^2\partial_x + x\left(2\partial_\rho - \bar{\theta}_- \partial_{\bar{\theta}_-} - \theta_- \partial_{\theta_-}\right) - \theta_- \bar{\theta}_- \mathcal{J} - e^{-\rho/2}\left(\bar{\theta}_- \bar{D}_+ + \theta_- D_+\right)\,, \\
&\mathcal{G}_+ = e^{-\rho/2}D_+ + x(\partial_{\theta_-} - \bar{\theta}_- \partial_x) + \bar{\theta}_- \mathcal{J} + 2\bar{\theta}_- \partial_\rho\,, \\
&\bar{\mathcal{G}}_+ = e^{-\rho/2}\bar{D}_+ + x(\partial_{\bar{\theta}_-} - \theta_- \partial_x) - \theta_- \mathcal{J} + 2\theta_- \partial_\rho\,, \\
&\mathcal{G}_- = \partial_{\theta_-} - \bar{\theta}_- \partial_x\,, \\
&\bar{\mathcal{G}}_- = \partial_{\bar{\theta}_-} - \theta_- \partial_x\,,
\end{aligned} \tag{102}
$$

where

$$
D_+ = \partial_{\theta_+} + \bar{\theta}_+ \partial_\gamma\,, \quad \bar{D}_+ = \partial_{\bar{\theta}_+} + \theta_+ \partial_\gamma\,. \tag{103}
$$

We can check that these generators obey the algebra (100). We can similarly define right generators by multiplying (101) on the right and then expressing the result as a *left* acting differential operator, e.g., $\mathcal{L}_a^R g = g L_a$, $\mathcal{G}_r^R g = g G_r$. These right generators obey the relations[9]

$$
\begin{aligned}
&[\mathcal{L}, \mathcal{L}^R] = [\mathcal{L}, \mathcal{G}^R] = \{\mathcal{G}, \mathcal{G}^R\} = 0\,, \\
&[\mathcal{L}_a^R, \mathcal{L}_b^R] = -\mathcal{L}_{[a,b]}^R\,, \quad [\mathcal{L}_a^R, \mathcal{G}_b^R] = -\mathcal{G}_{[a,b]}^R\,, \quad \{\mathcal{G}_a^R, \mathcal{G}_b^R\} = -\mathcal{L}_{\{a,b\}}^R\,.
\end{aligned} \tag{104}
$$

---

[9]The last minus sign in (104) is explained by considering $[\epsilon_\alpha \mathcal{L}_{H_\alpha}^R, \epsilon_\beta \mathcal{L}_{H_\beta}^R] = -\mathcal{L}_{[\epsilon_\beta H_\beta, \epsilon_\alpha H_\alpha]}^R$, where $H$ denotes any generator and $\mathcal{L}_H$ its differential operator. For fermionic generators, $\epsilon$ should be a Grassmann-odd number. The right generators formally satisfy the same algebra as the abstract generators in (101).

Explicitly,

$$
\begin{aligned}
\mathcal{J}^R &= -i\partial_a\,, \\
\mathcal{L}^R_- &= -e^{-\rho}\partial_x + \gamma^2\partial_\gamma - \gamma\left(2\partial_\rho - \bar{\theta}_+\partial_{\bar{\theta}_+} - \theta_+\partial_{\theta_+}\right) - \theta_+\bar{\theta}_+\mathcal{J}^R - e^{-\rho/2}\left(\bar{\theta}_+\bar{D}_- + \theta_+ D_-\right)\,, \\
\mathcal{L}^R_0 &= -\gamma\partial_\gamma + \partial_\rho - \frac{1}{2}\left(\bar{\theta}_+\partial_{\bar{\theta}_+} + \theta_+\partial_{\theta_+}\right)\,, \\
\mathcal{L}^R_+ &= \partial_\gamma\,, \\
\mathcal{G}^R_+ &= e^{-ia}\left[\partial_{\theta_+} - \bar{\theta}_+\partial_\gamma\right]\,, \\
\bar{\mathcal{G}}^R_+ &= e^{ia}\left[\partial_{\bar{\theta}_+} - \theta_+\partial_\gamma\right]\,, \\
\mathcal{G}^R_- &= e^{-ia}\left[e^{-\rho/2}D_- - \gamma(\partial_{\theta_+} - \bar{\theta}_+\partial_\gamma) + \bar{\theta}_+\mathcal{J}^R - 2\bar{\theta}_+\partial_\rho\right]\,, \\
\bar{\mathcal{G}}^R_- &= e^{ia}\left[e^{-\rho/2}\bar{D}_- - \gamma(\partial_{\bar{\theta}_+} - \theta_+\partial_\gamma) - \theta_+\mathcal{J}^R - 2\theta_+\partial_\rho\right]\,,
\end{aligned}
\tag{105}
$$

with

$$
D_- = \partial_{\theta_-} + \bar{\theta}_-\partial_x\,, \quad \bar{D}_- = \partial_{\bar{\theta}_-} + \theta_-\partial_x\,.
\tag{106}
$$

The right generator $\mathcal{J}^R$ is interpreted as the global symmetry generator (the $R$-charge) of the boundary theory. We will discuss the interpretation of the other right generators shortly.

We need the expression for the Casimir, which is

$$
\mathcal{C} = -\frac{1}{4}J^2 + L_0^2 - \frac{1}{2}(L_+L_- + L_-L_+) - \frac{1}{4}[\bar{G}_-, G_+] - \frac{1}{4}[G_-, \bar{G}_+]\,.
\tag{107}
$$

If we insert the left generators into the Casimir we find the following differential operator

$$
\mathcal{C} = \partial_\rho^2 + e^{-\rho}\partial_x\partial_\gamma - \frac{1}{4}(i\partial_a + \theta_+\partial_{\theta_+} - \bar{\theta}_+\partial_{\bar{\theta}_+})^2 - \frac{1}{2}e^{-\rho/2}(D_-\bar{D}_+ + \bar{D}_-D_+)\,,
\tag{108}
$$

with $D_+$ and $D_-$ as in (103), (106).

If we want to describe the bulk superspace, then we would need to quotient by a combination of the $\mathcal{L}^R$ generators which acts as a rotation. In addition, we also quotient by $\mathcal{J}^R$. This leaves two bosonic variables and four $\theta$ variables, which is what is necessary to describe a bulk two dimensional theory with $(2,2)$ supercharges. We want a bulk quantum field theory with $(2,2)$ supersymmetry and an anomaly free R symmetry.

## 4.3 The boundary superparticle

Now we turn to the description of the boundary particle, which is described by the $\mathcal{N}=2$ superschwarzian theory. We want to recover this description by thinking about a particle moving in the supercoset. The advantage of this description is that it will make the target space symmetries of the model very explicit.

As in the bosonic case, we quotient by a null bosonic generator, which we can take to be $\mathcal{L}^R_+$. We also demand that the wavefunctions have $\mathcal{L}^R_+ = -\mathrm{q}$. This leaves us with three bosonic degrees of freedom: the two coordinates of AdS$_2$ and an angular coordinate associated to the $R$ symmetry, which is good. However, it also leaves us with four fermionic coordinates, and their derivatives, which would build up to four complex fermions along the boundary. However, the $\mathcal{N}=2$ super-Schwarzian has only three complex fermionic degrees of freedom (or one complex fermion with a three derivative kinetic term), so we have an excess of one complex fermionic degree of freedom.

In general, the ungauged right symmetry generators that commute with $\mathcal{L}^R_+$ have the interpretation as generators of a global symmetry. For example $\mathcal{J}^R$ is the physical charge operator.[10]

---

[10]If we had an AdS$_2$ problem with a non-Abelian global symmetry $G$, the low energy description would be the Schwarzian mode plus a particle on a group manifold $G$. The right generators would be the physical charge operators.

In addition, if we quotient by $\mathcal{L}_+^R$ we see that we have some extra global symmetries, such as the ones generated by $\mathcal{G}_+^R$, which commute with all the left generators and also with $\mathcal{L}_+^R$. These are not symmetries of the superschwarzian theory. Therefore we are not reproducing the superschwarzian theory. We will describe two equivalent ways to deal with this problem.

A word of clarification: when we are discussing "gauging" in this context, we are talking about the *right* symmetries. This should not be confused with the symmetries of the Schwarzian action, which are sometimes also called "gauge symmetries". Those symmetries are the *left* symmetries. At this stage, we are trying to go from the particle on a supergroup manifold to the Schwarzian; to do so, we need to remove some of the right symmetries that do not appear at all in the Schwarzian action. In later steps, we will also gauge the left symmetries but we are not there yet.

### 4.3.1 The enlarged gauging formalism

One way to deal with this problem is to imagine that we are additionally gauging the $\mathcal{G}_+^R$ and $\bar{\mathcal{G}}_+^R$ generators. We cannot just demand that the wavefunction is invariant under these generators because their anticommutator is $\mathcal{L}_+^R$ which is non-zero. A procedure that produces the desired answer is the following. We first introduce an extra degree of freedom consisting of a single complex fermion $\zeta$. This realizes a representation of the subalgebra generated by $L_+^R$, $G_+^R$, $\bar{G}_+^R$, $J^R$ as

$$\hat{G}_+^R = \sqrt{2\mathfrak{q}}\,\zeta\,, \quad \hat{\bar{G}}_+^R = \sqrt{2\mathfrak{q}}\,\bar{\zeta}\,, \quad \hat{L}_+^R = \mathfrak{q}\,, \quad J^R = \frac{1}{2}[\zeta,\bar{\zeta}]\,, \quad \{\zeta,\bar{\zeta}\} = 1\,. \tag{109}$$

We now consider the full system given by the original coordinates in (101) and we now add the coordinate $\chi$. We further demand that the wavefunction is invariant under the sum of the two generators,

$$(\mathcal{L}_+^R + \hat{L}_+^R)\Psi = 0\,, \quad (\mathcal{G}_+^R + \hat{G}_+^R)\Psi = 0\,, \quad (\bar{\mathcal{G}}_+^R + \hat{\bar{G}}_+^R)\Psi = 0\,. \tag{110}$$

This procedure effectively removes one combination of the $\theta_+$ and $\bar{\theta}_+$ variables.

As a comment, note that this procedure is analogous to the following. Imagine that we are gauging a non-abelian symmetry, say $SU(N)$. One option is to demand that the wavefunction is invariant. Another option is to introduce an extra "quark" degree of freedom, just a single fundamental representation and demand that the total system is invariant. Notice that the Hilbert space of the degree of freedom only needs to furnish a representation of the gauge algebra in (110), but not of the full supergroup.

### 4.3.2 A formalism with fewer Grassmann variables

There is a second equivalent procedure where the extra variables are removed from the beginning. This works as follows. We first note that that the left generators involve the $\theta_+$ and $\bar{\theta}_+$ variables only in the combination $D_+$ and $\bar{D}_+$, see (102). We can realize the algebra of these operators in terms of just one Grassmann coordinate, instead of two

$$D_+ \to 2e^{ia}\chi\,\partial_\gamma\,, \quad \bar{D}_+ = e^{-ia}\partial_\chi\,. \tag{111}$$

In addition, we remove the $\theta_+$ and $\bar{\theta}_+$ terms from $\mathcal{J}$ so that

$$\mathcal{J} = -i\partial_a + \bar{\theta}_-\partial_{\bar{\theta}_-} - \theta_-\partial_{\theta_-}\,. \tag{112}$$

Here the factors of $e^{ia}$ are included so that the commutation relations of $\mathcal{J}$ with $D_+$ and $\bar{D}_+$ are the same as before, and $\chi$ is neutral under $\mathcal{J}$. In addition, we have that now

$$\mathcal{J}^R = -i\partial_a - \chi\partial_\chi\,. \tag{113}$$

With this definition we see that (111) is invariant under $\mathcal{J}^R$, as are all the rest of the generators (102).

The net effect is that we have replaced $\theta_+$ and $\bar{\theta}_+$ by just one Grassmann variable $\chi$. We have done this at the cost of breaking the manifest charge conjugation symmetry. With this procedure we can now no longer define the $\mathcal{G}_+^R$ generators, so we lack the unwanted symmetries, which is a good thing. In this formalism we impose just the condition $\mathcal{L}_+^R = -\mathfrak{q}$.

We found this to be a good compromise, and we are going to use these variables to describe the propagators. In summary, the claim is that the Schwarzian theory is described by a Hilbert space generated by functions of the form $e^{-\mathfrak{q}\gamma}F(x, \rho, a, \theta_-, \bar{\theta}_-, \chi)$ with the supergroup symmetries acting as in (102) with $D_\pm$ defined in (111) and $\mathcal{J}$ in (112).

### 4.3.3 Equivalence of the two procedures

Let us be more explicit about the procedure in section 4.3.1. The Hilbert space for a particle on the supergroup manifold consists of functions of the form $F(x, \rho, \gamma, a, \theta_-, \bar{\theta}_-, \theta_+, \bar{\theta}_+)$. Let us ignore all the arguments except $(\gamma, \theta_+, \bar{\theta}_+)$. Now when we add the extra fermion $\zeta$ we should consider functions $e^{-\mathfrak{q}\gamma}F(\theta_+, \bar{\theta}_+, \chi)$ and we now represent

$$\hat{G}_+^R = -2\chi\,\partial_\gamma\,, \quad \hat{\bar{G}}_+^R = -\partial_\chi\,, \quad \hat{J}^R = -\chi\,\partial_\chi\,, \quad \hat{L}_+^R = -\partial_\gamma\,, \quad \zeta = \sqrt{2}\chi\,, \quad \bar{\zeta} = \frac{1}{\sqrt{2}}\partial_\chi\,, \quad (114)$$

where we introduced $\sqrt{2}$ in a asymmetric way since the formalism is breaking the manifest charge conjugation symmetry anyway. All the other right generators, left and right, (102) (105), leave $\chi$ invariant. The physical $R$-charge is $\mathcal{J}^R + \hat{J}^R$, which is indeed (113).

Imposing the conditions (110), we restrict to a 2-dimensional subspace of functions of the form

$$e^{-\mathfrak{q}\gamma}F(\theta_+, \bar{\theta}_+, \chi) = \exp\left\{-\mathfrak{q}\left(\gamma + \theta_+\bar{\theta}_+ + 2e^{ia}\theta_+\chi\right)\right\}\left[a + \left(\chi + e^{-ia}\bar{\theta}_+\right)b\right]. \quad (115)$$

This 2-dimensional subspace is the physical Hilbert space. If we had only imposed the bosonic constraint in (110), we would have had an 8-dimensional "auxiliary" Hilbert space consisting of arbitrary functions of $(\theta_+, \bar{\theta}_+, \chi)$.

Now notice that since the physical Hilbert space is 2-dimensional, we can gauge fix $\theta_+ = \bar{\theta}_+ = 0$ in (115) to get $e^{-\mathfrak{q}\gamma}(a + \chi b)$, where the second factor is simply an arbitrary function of $\chi$. But this is precisely what we have done in the formalism of section 4.3.2. Once we know $a, b$ we could always restore the $\theta_+, \bar{\theta}_+$ dependence using (115).

## 4.4 The worldline supercharge

The Schwarzian theory has an $\mathcal{N} = 2$ worldline supersymmetry. This means that, besides the Hamiltonian, we should be able to construct two Grassmann-odd operators, $Q$, $\bar{Q}$ with the algebra $Q^2 = \bar{Q}^2 = 0$ and $\{Q, \bar{Q}\} = H$. These should be invariant under all the left generators which are gauge symmetries.

We have not found a systematic way to derive them. We have simply guessed them and then checked the algebra. One efficient way to guess them is to use the right generators, which are already invariant under all the left ones. We pick Grassmann-odd combinations which are invariant under $\mathcal{L}_+^R$ and $\mathcal{G}_+^R$, $\bar{\mathcal{G}}_+^R$ so that they are invariant also under the gauging procedure described near (109) This gives

$$\begin{aligned}
Q &= \mathcal{L}_0^R\bar{\mathcal{G}}_+^R - \mathcal{L}_+^R\bar{\mathcal{G}}_-^R - (1 - \tfrac{1}{2}\mathcal{J}^R)\bar{\mathcal{G}}_+^R\,, \\
\bar{Q} &= \mathcal{L}_0^R\mathcal{G}_+^R - \mathcal{L}_+^R\mathcal{G}_-^R - (1 + \tfrac{1}{2}\mathcal{J}^R)\mathcal{G}_+^R\,.
\end{aligned} \quad (116)$$

Using these expressions it is possible to write the supercharges as differential operators

$$
Q = \frac{1}{\sqrt{2}} e^{ia} \left\{ [\partial_\rho - \frac{1}{2}(i\partial_a + \theta_+ \partial_{\theta_+} - \bar{\theta}_+ \partial_{\bar{\theta}_+} + 1)]\bar{D}_+ - e^{-\rho/2} \partial_\gamma \bar{D}_- \right\},
$$
$$
\bar{Q} = \frac{1}{\sqrt{2}} e^{-ia} \left\{ [\partial_\rho + \frac{1}{2}(i\partial_a + \theta_+ \partial_{\theta_+} - \bar{\theta}_+ \partial_{\bar{\theta}_+} + 1)]D_+ - e^{-\rho/2} \partial_\gamma D_- \right\},
$$
(117)

with $D_-$ and in (106) and $D_+$ is as in (103) if we use the $\theta_+$ and $\bar{\theta}_+$ variables, as in section 4.3.1. Note that $Q$ has charge one under $\mathcal{J}^R$ and $\bar{Q}$ has charge minus one.

Alternatively, in the formalism of section 4.3.2 we get

$$
Q = \frac{1}{\sqrt{2}} e^{ia} \left\{ [\partial_\rho - \frac{i}{2}\partial_a + 1]\bar{D}_+ - e^{-\rho/2} \partial_\gamma \bar{D}_- \right\},
$$
$$
\bar{Q} = \frac{1}{\sqrt{2}} e^{-ia} \left\{ [\partial_\rho + \frac{i}{2}\partial_a + 1]D_+ - e^{-\rho/2} \partial_\gamma D_- \right\},
$$
$$
\mathcal{C} = \partial_\rho^2 + e^{-\rho} \partial_x \partial_\gamma + \frac{1}{4}\partial_a^2 - \frac{1}{2} e^{-\rho/2}(D_- \bar{D}_+ + \bar{D}_- D_+),
$$
(118)

with $D_+$ and $\bar{D}_+$ as in (111). We have also given the expression of the Casimir in these variables.

These obey the algebra

$$
Q^2 = 0 = \bar{Q}^2, \quad \{Q, \bar{Q}\} = \partial_\gamma \mathcal{C} = \mathcal{L}_+^R \mathcal{C}.
$$
(119)

In the next subsections we will describe the construction of the propagator $P$ which is a function of two sets of coordinates as above with opposite eigenvalues under $\mathcal{L}_+^R$. Namely the $\gamma$ dependence of the propagator is

$$
P \propto e^{-q(\gamma_1 - \gamma_2)}.
$$
(120)

The rest of the dependence on the other bosonic and fermionic coordinates is constrained by demanding that $P$ is a function of $OSp$ invariants. These invariants are just the superanalog of the distance in the bosonic case.

## 4.5 Invariants in the formalism with fewer Grassmann variables

In this subsection we consider invariants under the sum of two generators of the form (102) with $D_\pm$ defined in (111). We have a total of $2 \times (4|3)$ variables and a set of $(4|4) + (1|0)$ constraints, where the last constraint comes from $\mathcal{L}_{1+}^R + \mathcal{L}_{2+}^R$ invariance. Therefore we expect a total of 3|2 invariants, three bosonic and two fermionic ones.

The three bosonic ones are completions of the two invariants we had in the bosonic case, (94), $\gamma_{12} + \varphi_{12}$, the distance $\ell$, together with $U(1)$ Wilson line $e^{i(a_1 - a_2)}$. We can work them out explicitly to find

$$
w = e^{(\rho_1 + \rho_2)/2} \left[ x_1 - x_2 - \theta_{1-} \bar{\theta}_{2-} - \bar{\theta}_{1-} \theta_{2-} \right],
$$
(121)

$$
e^{i\Sigma} = e^{i(a_1 - a_2)} \left( 1 - \frac{(\theta_{1-} - \theta_{2-})(\bar{\theta}_{1-} - \bar{\theta}_{2-})}{(x_1 - x_2)} \right),
$$
(122)

$$
\Phi = \gamma_1 - \gamma_2 + \frac{e^{(\rho_1 - \rho_2)/2} + e^{-(\rho_1 - \rho_2)/2}}{w} + 2(e^{ia_1} e^{\rho_2/2} \chi_1 - e^{ia_2} e^{\rho_1/2} \chi_2) \frac{(\theta_{1-} - \theta_{2-})}{w} +
$$
$$
- (e^{\rho_1} - e^{\rho_2}) \frac{(\theta_{1-} - \theta_{2-})(\bar{\theta}_{1-} - \bar{\theta}_{2-})}{w^2},
$$
(123)

$$
\eta_1 = \chi_1 - e^{-ia_1} e^{\rho_2/2} \frac{(\bar{\theta}_{1-} - \bar{\theta}_{2-})}{w},
$$

$$
\eta_2 = \chi_2 - e^{-ia_2} e^{\rho_1/2} \frac{(\bar{\theta}_{1-} - \bar{\theta}_{2-})}{w}.
$$
(124)

It is also possible to write the invariants using the extra gauging formalism of section 4.3.1 where we keep the $\theta_+$ and $\bar{\theta}_+$ variables. The explicit expressions are given in appendix D. We have also shown how those invariants together with the formalism of section (4.3.1) reduces them to the ones above.

## 4.6 Assembling the zero energy propagator

We will concentrate on the propagator for the zero energy states which should obey $Q_1 P = \bar{Q}_1 P = Q_2 P = \bar{Q}_2 P = 0$, where the subscript index indicates whether the supercharge acts on the first or second point. These also imply that $H_1 P = H_2 P = 0$. We can also fix the $R$ charge $j$ of the boundary particle.

As part of the right gauge symmetry, we fix the eigenvalue of $\mathcal{L}_+^R$ of the first particle to be $-\mathfrak{q}$.

Note that $\eta_1$ has charges $(-1, 0)$ under $\mathcal{J}_1^R$, $\mathcal{J}_2^R$. Similarly, $\eta_2$ has charge $(0, -1)$. This implies that there is no way to write a propagator containing these fermions which will be neutral under $\mathcal{J}_1^R + \mathcal{J}_2^R$. This is not a problem because the integral over the positions will also include an integral over $\chi$, $\int \mathrm{d}\chi$, which has $\mathcal{J}^R$ charge one. So the propagator should have $\mathcal{J}_1^R + \mathcal{J}_2^R = -1$. We then write the expression

$$P = e^{-\mathfrak{q}\Phi} e^{ij\Sigma} \left[ \eta_2 e^{-\frac{i}{2}\Sigma} A(w) + \eta_1 e^{\frac{i}{2}\Sigma} B(w) \right], \tag{125}$$

which has $R$ charges $(\mathcal{J}_1^R, \mathcal{J}_2^R) = (j - \frac{1}{2}, -j - \frac{1}{2})$. Since this is an expression in terms of invariants, it is convenient to express the supercharges (118) as differential operators acting on the invariants. It is clear that this is possible because the supercharges (anti)commute with the left generators, so they should map invariants to invariants. We find

$$
\begin{aligned}
Q_1 &= \frac{1}{\sqrt{2}} \left[ \frac{1}{2}(w\partial_\omega - i\partial_\Sigma)\partial_{\eta_1} + \frac{e^{i\Sigma}}{w}\partial_{\eta_2}\partial_\Phi \right], \\
\bar{Q}_1 &= \frac{1}{\sqrt{2}} \left[ (w\partial_\omega + i\partial_\Sigma)\eta_1\partial_\Phi - 2\frac{e^{-i\Sigma}}{w}\eta_2\partial_\Phi^2 \right].
\end{aligned}
\tag{126}
$$

Imposing that these annihilate (125) we get

$$\left( w\partial_w - j + \frac{1}{2} \right)A - 2\frac{\mathfrak{q}}{w}B = 0, \quad \left( w\partial_w + j + \frac{1}{2} \right)B - 2\frac{\mathfrak{q}}{w}A = 0, \tag{127}$$

which implies that

$$A = \frac{1}{w}K_{\frac{1}{2}+j}\left(\frac{2\mathfrak{q}}{w}\right), \quad B = \frac{1}{w}K_{\frac{1}{2}-j}\left(\frac{2\mathfrak{q}}{w}\right). \tag{128}$$

Hence we can write the final expression for the propagator as

$$P_j = e^{-\mathfrak{q}\Phi} e^{ij\Sigma} \frac{2\cos\pi j}{\pi} \frac{1}{w} \left[ \eta_2 e^{-\frac{i}{2}\Sigma} K_{\frac{1}{2}+j}\left(\frac{2\mathfrak{q}}{w}\right) + \eta_1 e^{\frac{i}{2}\Sigma} K_{\frac{1}{2}-j}\left(\frac{2\mathfrak{q}}{w}\right) \right]\Theta(x_1 - x_2). \tag{129}$$

The normalization factor will be explained in section 4.7.1. We have also included the step function $\Theta$ to ensure $x_1 > x_2$; this follows from the constraint that the particle only moves forwards in bulk time, see [11]. At this point we want to set $\mathfrak{q} = 1$ to match the conventions we had in section 2. We explain the overall normalization of (129) below.

Note that for $j = 0$ the propagator takes a particularly simple form

$$P_0 = e^{-\mathfrak{q}\Phi} \left[ \eta_2 e^{-\frac{i}{2}\Sigma} + \eta_1 e^{\frac{i}{2}\Sigma} \right] \frac{e^{-\frac{2}{w}}}{\sqrt{\pi}\sqrt{w}} \Theta(x_1 - x_2). \tag{130}$$

The measure of integration for each bulk point is

$$\int \mathrm{d}\mu = \frac{1}{2\pi\hat{q}} \int \mathrm{d}\rho\,\mathrm{d}x\,\mathrm{d}\theta_-\,\mathrm{d}\bar{\theta}_-\,\mathrm{d}a\,\mathrm{d}\chi\,. \tag{131}$$

This measure is invariant under the left generators, but it has right $R$-charge equal to minus one, which cancels against the $R$-charge of the product of the two boundary propagators coming in and out of this point.

## 4.7 The two point functions and comparison with the Liouville approach

Here we connect this propagator approach to the one we found in section 2 using the super-Liouville approach. First we note that we have only computed the zero energy propagator. So we can only compare to the zero energy wavefunctions in (22). We see that they have the same form when $\mathfrak{q} = 1$, which is why we set this value.

In the Liouville approach we were working directly with invariant quantities. These are related to the ones in (121-124). These are related as follows

$$e^{\ell/2} = w\,, \quad a = \Sigma\,, \quad \psi_r = -\frac{i}{\sqrt{2}}\partial_{\eta_1}\,, \quad \bar{\psi}_r = i\sqrt{2}\eta_1\,, \quad \psi_l = -\frac{1}{\sqrt{2}}\partial_{\eta_2}\,, \quad \bar{\psi}_l = -\sqrt{2}\eta_2\,. \tag{132}$$

Note that in the Liouville realization the fermions were operators whose anticommutation relations are realized in terms of the Grassmann variables of the formalism in this section. With this identification we see that the supercharges in (126) become $Q_r$ and $\bar{Q}_r$ in (8), after we set $\partial_\Phi = -\mathfrak{q} = -1$. This shows that the results in this section should match the Liouville results because we are solving the same equations. It is however instructive to compute the two point function in the current formalism to match it with the previous one.

In the formalism of this section, the zero energy two point function in a vacuum of $R$ charge $j$ is computed as

$$\langle 2\,\mathrm{pt}\rangle_{j,zz} = \pi e^{S_0} \int \frac{\mathrm{d}\mu_1\,\mathrm{d}\mu_2}{\mathrm{Vol}(OSp(2|2))} P_j(1,2) P_j(2,1) e^{-\Delta\ell}\,, \tag{133}$$

where the overall factor of $\pi$ was chosen so that the final answers, after the gauge fixing procedure, agree with the Liouville case. It is useful to note that the propagator in the other order $P(2,1)$ is given by

$$P_j(2,1) = e^{\mathfrak{q}\Phi} e^{-ij\Sigma} \frac{2\cos\pi j}{\pi} \frac{1}{w} \left[ \eta_1 e^{\frac{i}{2}\Sigma} K_{\frac{1}{2}+j}\left(\frac{2\mathfrak{q}}{w}\right) - \eta_2 e^{-\frac{i}{2}\Sigma} K_{\frac{1}{2}-j}\left(\frac{2\mathfrak{q}}{w}\right) \right] \Theta(x_1 - x_2)\,. \tag{134}$$

This was obtained by solving the $Q_1 P = \bar{Q}_1 P = 0$ again. In particular, this means that the term involving $\Phi$ cancels out in (133).

The gauge symmetry allows us to make the gauge choices

$$x_1 = 1\,, \quad x_2 = 0\,, \quad \rho_2 = 0\,, \quad a_2 = 0\,, \quad \theta_{1-} = \theta_{2-} = \bar{\theta}_{1-} = \bar{\theta}_{2-} = 0\,. \tag{135}$$

With these choices we find that (121-124) become

$$w = e^{-\ell/2}\,, \quad \text{with} \quad \ell = \rho_1\,, \quad \Sigma = a_1\,, \quad \eta_1 = \chi_1\,, \quad \eta_2 = \chi_2\,. \tag{136}$$

The Jacobian is just a constant so that the correlator (133) becomes

$$\langle 2\,\mathrm{pt}\rangle_{j,zz} = \pi e^{S_0} \frac{1}{2\pi\hat{q}} \int \mathrm{d}\rho_1\,\mathrm{d}a_1\,\mathrm{d}\chi_1\,\mathrm{d}\chi_2 P_j(1,2) P_j(2,1) e^{-\Delta\ell} \tag{137}$$

$$= e^{S_0} (\cos\pi j)^2 \int \mathrm{d}\ell \left\{ \left[ \frac{2}{\sqrt{\pi}} e^{-\ell/2} K_{\frac{1}{2}+j}(2e^{-\ell/2}) \right]^2 + \left[ \frac{2}{\sqrt{\pi}} e^{-\ell/2} K_{\frac{1}{2}-j}(2e^{-\ell/2}) \right]^2 \right\} e^{-\Delta\ell}\,.$$

The last expression is what we obtained for the zero energy sector using the Liouville approach. Here we have compared the neutral operator, but we can also obtain the answer for the charged operators. And we can also include the sum over $j$ if we wanted.

### 4.7.1 Check of the composition law for the boundary propagator

A non-trivial check of our formulas is that the propagator, (129), obeys the composition law

$$\int d\mu_2 P_j(1,2) P_j(2,3) = P_j(1,3). \tag{138}$$

This check is described in detail in appendix F. Demanding that there is a unit coefficient in the right hand side fixes the overall normalization in (129).

Let us comment on some aspects of this check that will be useful later. For simplicity, consider the case $j = 0$, where the propagator simplifies (130). To perform this check, it is convenient to use the symmetries to choose

$$x_1 = 1, \quad x_3 = a_3 = 0 = \rho_3, \quad \theta_{1-} = \bar{\theta}_{1-} = \theta_{3-} = \bar{\theta}_{3-} = 0. \tag{139}$$

This leaves an integral over some Grassmann variables in (138) which can be explicitly computed to obtain an expression of the form

$$I = \int_0^1 \frac{dx_2}{\sqrt{x_2(1-x_2)}} \int_0^\infty d\sqrt{z_2}\, e^{-\frac{(\sqrt{z_1}+\sqrt{z_2})^2}{(1-x_2)} - \frac{(1+\sqrt{z_2})^2}{x_2}} \left[ \frac{(1+\sqrt{z_2})^2}{x_2^2} + \frac{(\sqrt{z_1}+\sqrt{z_2})^2}{(1-x_2)^2} \right.$$
$$\left. + \frac{2(1+\sqrt{z_2})(\sqrt{z_1}+\sqrt{z_2})}{x_2(1-x_2)} - \frac{1}{2}\left(\frac{1}{x_2} + \frac{1}{1-x_2}\right) \right], \quad z_i = e^{-\rho_i}. \tag{140}$$

This integral can be done, as explained in appendix F. However, here we want to point out one feature of it, which is that the integrand is not positive definite. For example, it becomes negative for small values of $z_1$ and $z_2$, and $x_2 \sim 1$. This implies that after we integrate out the fermions, we do not get a positive measure on the space of paths. Therefore we cannot answer easily questions like: "what is the typical path that contributes". Below we discuss another application of (140).

## 4.8 Higher order correlators

This expression for the propagator enables us to compute the general expression for any correlator in the long boundary distance limit. The correlator involves two pieces. One is a correlator of a supersymmetric field theory in AdS$_2$ when we take the limit that the points are near the boundary. The other is the propagator of the boundary particle that we discussed above. We will discuss these two pieces in turn.

### 4.8.1 The boundary limit of a bulk correlator

Let us first recall how the boundary limit of bulk field correlators behave in the case with no supersymmetry

$$\langle \phi(x_1, \rho_1) \cdots \phi(x_n, \rho_n) \rangle = \left( \prod_i \epsilon^{\Delta_i} e^{-\Delta_i \rho_i} \right) \langle O(x_1) \cdots O(x_n) \rangle. \tag{141}$$

Note that our variables $\rho_i$ have been already defined so that they are of order one near the boundary. This is the origin of the factors of $\epsilon^{\Delta_i}$ in (141). That is also why we wrote an equality above. The fact that we have a simple behavior for the $\rho$ variable can be obtained by noticing that the bulk fields obey a massive wave equation given by the Casimir operator (91), but now with no $\gamma$ dependence, $H_i \phi_i = m_i^2 \phi_i$. Acting on each variable this constrains the $\rho$ dependence to the one in (141), with the usual relation between $m$ and $\Delta$. Here the notation

$\langle O(x_1)\cdots O(x_n)\rangle$ indicates that this term behaves as the correlation function of operators of dimensions $\Delta_i$ as a function of the bulk boundary times $x_i$.

As we explain below, in the case of correlators of bulk superfields, we expect that as we approach the conformal boundary we get

$$\langle \Phi(\mathbf{x}_1)\cdots\Phi(\mathbf{x}_n)\rangle \sim \prod_{i=1}^{n} \epsilon^{\Delta_i} e^{-\Delta_i \rho_i} \langle O(x_1,\theta_{1-},\bar{\theta}_{1-})\cdots O(x_n,\theta_{n-},\bar{\theta}_{n-})\rangle,$$
$$\text{with} \qquad \mathbf{x}_i = (x_i,\rho_i,\theta_{i-},\bar{\theta}_{i-},\theta_{i+},\bar{\theta}_{i+}). \tag{142}$$

In other words, the correlator becomes $\theta_+$ and $\bar{\theta}_+$ independent, and it depends on the $\rho_i$ variables in a simple way. The correlator on the right hand side can be viewed as a "boundary" correlator defined via the boundary limit of the bulk one; it depends on $x$ and the $\theta_-$, $\bar{\theta}_-$ variables.

The $\theta_+$ independence is expected for the following reason. The bulk fields live in AdS$_2$ which is a coset. This means that they are functions with zero eigenvalue under $\mathcal{L}_+^R$. However, since $\mathcal{G}_+^R$ and $\bar{\mathcal{G}}_+^R$ (see (105)) anticommute to this generator, and we expect a unitary representation, then this implies that they also annihilate the bulk field in this limit. Looking at (105) we conclude that we have no dependence on the $\theta_+$ variables. Note that this is an argument near the boundary only, not about the bulk of AdS$_2$, since in the bulk we are quotienting by a different bosonic generator. Once we demonstrate this $\theta_+$, $\bar{\theta}_+$ independence, we can impose that the casimir has an eigenvalue $(\Delta^2 - r^2/4)$, with $r$ the R-charge of the bulk field. This leads to the $\rho$ dependence in the right hand side of (142).

### 4.8.2 General expression

The correlation function for a general correlator has the form

$$\langle \hat{O}_1\cdots\hat{O}_n\rangle = \pi e^{S_0} \int \frac{\prod_i d\mu_i}{\text{vol}(SU(1,1|1))} P(1,2)P(2,3)\cdots P(n-1,n)P(n,1)$$
$$\times \left(\prod_i e^{-\Delta_i\rho_i + ir_i a_i}\right) \langle O_1(x_1,\theta_{1-},\bar{\theta}_{1-})\cdots O_1(x_1,\theta_{1-},\bar{\theta}_{1-})\rangle, \quad (143)$$

where the measure $d\mu_i$ is defined in (131). The factor of volume of the group in the denominator is fixed so that when we perform a gauge choice similar to the one we did for the two point function (135) we get no additional factor, as in (137). We have added a factor that ensures that the $R$ charges of the propagators are conserved at the vertices, taking into account the $R$ charges of the fields.

We could not evaluate these integrals but we expect them to be convergent so that they give us a finite constant for each correlation. Notice that the bulk piece, $\langle O_1(x_1,\theta_{1-},\bar{\theta}_{1-})\cdots O_1(x_1,\theta_{1-},\bar{\theta}_{1-})\rangle$ could have an intricate $x$ dependence. However, after we dress with the zero energy propagators and integrate, we simply get a constant.

Note that the correlator (143) is cyclically invariant but not fully permutation invariant.

We could also imagine integrating the propagators and the factor in parentheses in (143) over the variables $\rho_i, \chi_i, a_i$. This gives a function of $x_i, \theta_{i-}, \bar{\theta}_{i-}$ with conformal dimensions such that we can integrate it against $\langle O_1(x_1,\theta_{1-},\bar{\theta}_{1-})\cdots O_1(x_1,\theta_{1-},\bar{\theta}_{1-})\rangle$ in a conformal invariant fashion.

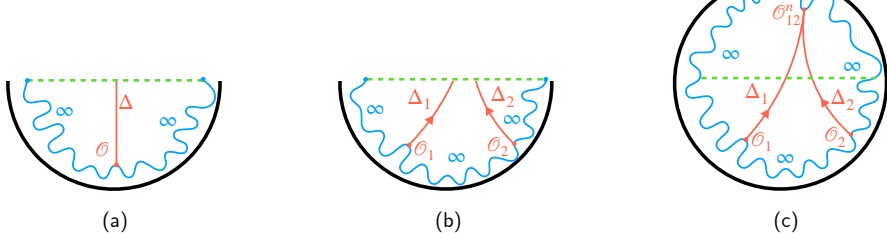

Figure 13: Diagrams that represent the creation of a zero energy wormhole, a supersymmetric wormhole. (a) A wormhole with one particle. (b) A wormhole with two particles. (c) An overlap between two different ways to construct a wormhole with two particles.

# 5 Applications of the propagator formalism

## 5.1 Supersymmetric wormholes

In holography, a wormhole is described by an entangled state in two copies of the boundary theory. A simple wormhole state is the thermofield double with inverse temperature $\beta$, which corresponds to the eternal black hole [32]. The $\beta \to \infty$ limit of the wormhole is a finite length supersymmetric wormhole. It is supersymmetric in the sense that it respects the boundary supercharges. Therefore, these configurations are an example of ER=EPR for supersymmetric states.

We can generate other states by inserting operators during the long time of euclidean evolution. These produce further wormholes that are filled with matter, which also have $E = 0$ and are therefore supersymmetric. So we have an interesting situation where supersymmetry seems to be broken in the bulk by the bulk matter, but nevertheless, the quantum mechanics of the boundary modes projects us again into a supersymmetric state. So this is a family of supersymmetric Einstein Rosen wormholes.

If we insert a single operator of dimension $\Delta$, then we get a single state whose properties will depend on the dimension of the operator we inserted, see figure 13a. One particular question we are interested in is the length of the wormhole in the presence of this additional matter.

We will explain how to compute this length in a saddle-point approximation when $\Delta$ is large. We consider the OTOC involving a pair of operators of dimension $\Delta$ and another pair of dimension $\Delta'$, see figure 14. If $\Delta$ is the dimension of the particle we insert, then we can think of $\Delta'$ as a small dimension which is a probe that will tell us how long is the wormhole. In particular, we could take the $\Delta' \to 0$ limit to extract the length. More precisely, the length is given by taking the derivative with respect to $\Delta'$ of the OTOC and then setting $\Delta' = 0$. This is the four point function in figure 14 for very small $\Delta'$.

Imagine first taking the limit $\Delta' = 0$. In this case, the OTOC reduces simply to two pairs of propagators. Each pair involves an integral of the form (138), which of course can be used to recover the two point function of the inserted operator. Let us now assume that the dimension $\Delta$ of this inserted operator is large. We can then estimate the distance in the bulk by looking at the integral for the two point function which, for $j = 0$ has the form

$$\langle e^{-\Delta \ell} \rangle_{j=0} \propto \int d\ell \, e^{-\Delta \ell} e^{-\ell} e^{-4e^{-\ell/2}} \,. \tag{144}$$

Using a saddle point approximation for large $\Delta$ we find that the saddle value is

$$\ell_* \sim -2 \log(\Delta + 1) \,. \tag{145}$$

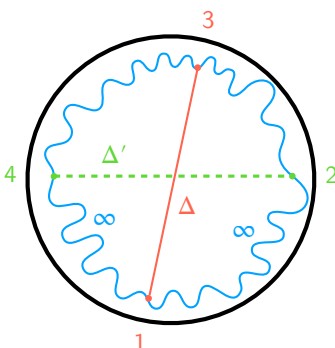

Figure 14: We imagine a 4pt function involving a pair of heavy operator of dimension $\Delta \gg 1$ and another operator of dimension $\Delta'$ between points 2 and 4. If we do not insert any operator at 2 and 4, we have a wormhole that contains matter in the representation with weight $\Delta$. The typical distance between 2 and 4 gives us an estimate for the size of the wormhole.

This is the distance $\ell_{13}$ in figure 14 when $\Delta' = 0$. The $+1$ is not to be trusted, but we left it because it gives the right order of magnitude when $\Delta = 0$ for the peak of the wavefunction of the empty wormhole. We see that the length becomes shorter when $\Delta$ is large. This is to be expected, the bulk geodesic pulls in the boundary particle. Of course, inserting this saddle point value in the integral (144) we get the large $\Delta$ approximation to the zero energy contribution to the two point function[11] (59). This gives:

$$\langle e^{-\Delta \ell} \rangle \propto \exp\left[2\Delta(\log \Delta - \log 2 - 1)\right]. \tag{146}$$

Now let us consider the expression for the OTOC using the results of the propagator formalism. It is convenient to make a gauge choice of the form

$$x_1 = 0, \quad x_3 - \rho_3 \to \infty, \quad \text{with} \quad x_3^2 e^{\rho_3} = 1, \quad a_3 = 0, \quad \theta_{1-} = \bar{\theta}_{1-} = \theta_{3-} = \bar{\theta}_{3-} = 0. \tag{147}$$

With this gauge choice, the distance $\ell_{13}$ is[12]

$$e^{\ell_{13}} \sim 1/z_1, \quad \text{with} \quad z_i = e^{-\rho_i}. \tag{148}$$

Then, in this limit the exponential piece of the propagators involving particle 3 goes as

$$P(2,3)P(3,4) \propto \exp\left(-x_2 + x_4 - 2\sqrt{z_2} - 2\sqrt{z_4}\right), \tag{149}$$

where we imagine that $x_2 > 0$ and $x_4 < 0$. Adding then the exponential pieces of the other two propagators we get

$$P(1,2)P(2,3)P(3,4)P(4,1) \sim \exp\left(-\frac{(\sqrt{z_1} + \sqrt{z_4})^2}{(-x_4)} - \frac{(\sqrt{z_1} + \sqrt{z_2})^2}{x_2} - x_2 + x_4 - 2\sqrt{z_2} - 2\sqrt{z_4}\right), \tag{150}$$

we focus on the exponential pieces because we will see that there are large values in the exponents that we want to minimize, so that we expect that possible prefactors of the exponentials are not as important as the exponent.

We assume that $\Delta$ is large, so that $\ell_{13}$ is approximated by (145). Then $z_1$ is large, $z_1 \propto \Delta^2$. We now want to find the typical values of $x_2$ and $x_4$ that contribute when $z_1$ is large. As a first

---

[11]Up to powers of $\Delta$ in the prefactor.

[12]The $z_i$ coordinates can be viewed as a rescaled version of the usual Poincare $z$ coordinate, see (92).

approximation we find that we minimize the exponent at $z_2 = z_4 = 0$. Then the problem for $x_2$ and $x_4$ factorizes and we find that the action is maximal at

$$x_2 \sim -x_4 \sim \sqrt{z_1}\,. \tag{151}$$

We now set these values in (150). Then we expand the resulting exponent in $z_2$ and $z_4$ to find that only values of order one in $z_2$ and $z_4$ contribute. Finally, we notice that the distance is of the form (see (A.7))

$$e^{\ell_{24}} \sim \frac{(x_2 - x_4)^4}{z_2 z_4} \quad \longrightarrow \quad \ell_{24} \varpropto 2 \log z_1 \sim 2 \log \Delta\,. \tag{152}$$

For large $\Delta$ we see that the distance between the two sides is growing when we add matter. Inserting the heavy particle has disrupted the simple correlations between the two sides.

### 5.1.1 The OTOC for large $\Delta$

We now consider the OTOC correlator in the large $\Delta = \Delta' \gg 1$ limit. We perform a saddle point approximation to the integral. We have a symmetric configuration and therefore we expect a saddle point that is symmetric, as in figure 15. We see that there are two distinct distances in the problem, one is the distance between consecutive points, called $\ell'$ in figure (15) and the other the distance between opposite points. These two distances can be computed by thinking about the expression for the distance in the radial coordinates $ds^2 = d\tilde{\rho}^2 + \sinh^2 \tilde{\rho}\, d\theta^2$

$$e^{-\ell} = \frac{1}{\epsilon^2} \frac{e^{-\tilde{\rho}_1 - \tilde{\rho}_2}}{(\sin \frac{\theta_1 - \theta_2}{2})^2} = \frac{e^{-\hat{\rho}_1 - \hat{\rho}_2}}{(\sin \frac{\theta_1 - \theta_2}{2})^2}\,, \quad \text{with} \quad e^{\hat{\rho}} = \epsilon e^{\tilde{\rho}}\,, \tag{153}$$

with $\epsilon \to 0$ and $\hat{\rho}$ kept finite. We now write the two distances in question

$$e^{-\ell} = e^{-\ell_{13}} = e^{-2\hat{\rho}}\,, \quad e^{-\ell'} = e^{-\ell_{12}} = 2e^{-2\hat{\rho}}\,, \tag{154}$$

where the two comes from $(\sin \frac{\pi}{4})^{-2}$. The correlator involves the factors

$$e^{-2\Delta \ell} \exp\left(-4 \times 2 e^{-\ell'/2} - 4 e^{-\ell'+\ell/2}\right) = e^{-4\Delta\hat{\rho}} \exp\left(-8\sqrt{2} e^{-\hat{\rho}} - 8 e^{-\hat{\rho}}\right)\,, \tag{155}$$

where the first factor comes from the two bulk propagators, the first term in the exponential from the Bessel function part of the propagator and the last term from the "phase" factors, which we simplified using (J.3) . Here we have ignored all Grassmann coordinates since they involve factors of the coordinates multiplying (155) and could contribute only powers of $\Delta$ but not exponentials in $\Delta$, which is what we will get from (155).

Extremizing (155) we get $2(\sqrt{2} + 1)e^{-\hat{\rho}} = \Delta$ and substituting in (155) we get

$$\exp\left\{4\Delta \log \Delta + 4\Delta\left[-1 + \log\left(\frac{(\sqrt{2} - 1)}{2}\right)\right]\right\}\,. \tag{156}$$

We when we compute the ratio of (156) to the square of the two point function (146) we get

$$\frac{\text{OTOC}}{\text{TOC}} \sim \exp\left(4\Delta \log(\sqrt{2} - 1)\right) \sim \exp(-(3.52...)\Delta)\,, \tag{157}$$

which is saying that the OTOC is suppressed relative to the TOC.

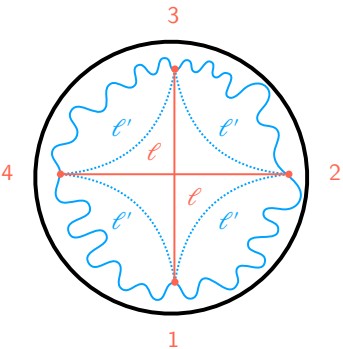

Figure 15: Computing the OTOC for large $\Delta = \Delta' \gg 1$. We expect the saddle point to be a symmetric configuration.

## 5.2 States with two particles

We can also create a wormhole state that contains two particles, such as the one displayed in figure 13b. These states arise by inserting two operators, each of which is separated by an infinite amount of Euclidean boundary time. It is interesting to ask how this wormhole looks from the inside. We expect that the state for given matter primaries is unique, once we attach the boundary particles in the zero energy state. However, the state constructed as in figure 13b will contain a superposition of primaries with dimensions

$$\Delta = \Delta_1 + \Delta_2 + n. \tag{158}$$

It is an interesting problem to figure out which linear combination of primary states this produces. One way to understand that is to first construct the two particle primary operator in the bulk and then dress that with the boundary particles, as in the top part of figure 13c. As in that figure, we can consider then the overlap between that state and the one we constructed in figure 13b. By taking the overlap with the various states we can figure out what state we get. This amounts to evaluating various 3-pt functions.

There are some special cases where the problem is easy to analyze. For example, we could consider the two operators to be BPS operators like the elementary fermions of a supersymmetric $\mathcal{N} = 2$ SYK model. In that case, we find that the action of such operators is independent on the distance between them, as discussed after eqn (84). This means that the two particles that are created are on top of each other in this case. This is closely related to the following identity that holds for the zero energy correlators:

$$\frac{\langle Z_{j-2\Delta_1-2\Delta_2}|e^{-\Delta_1(\ell+2ia)}|Z_{j-2\Delta_2}\rangle \langle Z_{j-2\Delta_2}|e^{-\Delta_2(\ell+2ia)}|Z_j\rangle}{\langle Z_{j-2\Delta_2}|Z_{j-2\Delta_2}\rangle} = \langle Z_{j-2\Delta_1-2\Delta_2}|e^{-(\Delta_1+\Delta_2)(\ell+2ia)}|Z_j\rangle, \tag{159}$$

which can be easily checked from (36). The left hand side is the expression we would obtain if two two operators are separated by a long time; the last expression is if they were on top of each other. Here we used the special properties of BPS boundary operators. So we expect that the insertion of two BPS particles as in figure (13)b produces two particles on top of each other, or equivalently that only the $n = 0$ primary in (158) is produced.

## 5.3 Entropy of a wormhole with particles

Here we consider the 2-sided state produced by the insertion of an operator $O$ and evolved by long Euclidean times so that we get $POP = \hat{O}$ which is the projection of $O$ to the low energy BPS sector. This produces an unnormalized density matrix $\tilde{\rho} = POPO^\dagger P$. For simplicity we

will take a neutral operator $O = O^\dagger$ and we will also specialize to the case where $P$ projects onto states with 0 charge. The unnormalized Renyi entropies

$$Z_n = \text{Tr}[\tilde{\rho}^n], \tag{160}$$

are given by an $2n$ point function of the operator. We will consider the limit where $\Delta$ is very large and all crossed diagrams are suppressed, which follows from section 5.1.1. With this assumption, the computation of $Z_n$ involves planar contractions. This implies that $O$ has an eigenvalue spectrum of a Gaussian $L \times L$ Hermitian matrix, e.g., a semi-circle distribution $\mu(\lambda) = 2L\sqrt{a^2 - \lambda^2}/(\pi a^2)$. Here $a$ is determined by the 2-pt function and $L = e^{S_0}$. To compute the entanglement spectrum of $\rho$, we need to consider the normalized eigenvalue distribution of $p = \lambda^2/Z$. So we obtain

$$\mu(p) = \mu(\lambda)d\lambda/dp = \frac{2LZ}{\pi a^2}\sqrt{\frac{a^2}{pZ} - 1}. \tag{161}$$

Enforcing the normalization condition $\int \mu(p)dp\, p = 1$ sets $Z = La^2/4$, giving an entanglement spectrum that is independent of $a$:

$$\mu(p) = \frac{L}{2\pi}\sqrt{\frac{4}{Lp} - 1}. \tag{162}$$

This is independent of the operator and its dimension, as long as $\Delta \gg 1$. We can use this distribution to calculate the von Neumann entropy:

$$S = -\text{Tr}[\rho \log \rho] = \int dp\, \mu(p)(-p \log p) = \log L - \frac{1}{2}. \tag{163}$$

So we see that the entropy is smaller than maximal, but only by an order one quantity. Curiously, the answer is independent of the conformal dimension of the operator (in this large $\Delta$ approximation).

Notice that in the large $L$ limit we have the structure of a type $\text{II}_1$ algebra, with an entropy that can only become smaller than maximal, when we add extra bulk particles.

In the classical approximation, one can computes the entropy using the RT formula, which instructs us to minimize the value of the dilaton. The minimum value is simply $S_0 = \log L$, independent of $\Delta$. When we include the quantum corrections, we get a shift by $-1/2$, (163).

As a side comment, we can also compute the Renyi entropies of $\rho$, either by using the distribution $\mu(p)$ or by simply counting planar Wick contractions [33]. We obtain

$$Z_n/(Z_1)^n = L^{1-n}C_n, \quad C_n = \frac{(2n)!}{(n+1)!n!}, \tag{164}$$

where $C_n$ are known as the Catalan numbers. Then by replacing the factorials in the formula with Gamma functions, we can use the replica trick to compute the entropy:

$$S = -\partial_n\left(\frac{\log Z_n}{n}\right)\Bigg|_{n=1} = \log L - \frac{1}{2}. \tag{165}$$

We can also consider a setup where we insert $q$ particles of the *same* species, all separated by long Euclidean times. Then we simply shift $C_n \to C_{qn}$. This gives

$$S - S_0 = \log\left(C_q\right) - q\left(-H_{q+1} + H_{q-\frac{1}{2}} + \log(4)\right), \tag{166}$$

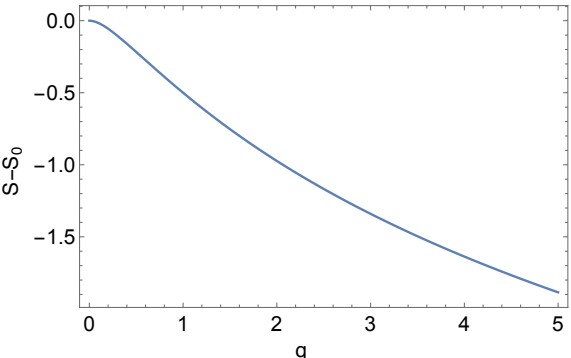

Figure 16: Entanglement entropy of a BPS wormhole with $q$ particles inserted (all particles are separated by infinite imaginary time). The entropy decreases as a function of $q$ away from its classical value $S_0$.

where in Mathematica, $H_q = \texttt{HarmonicNumber[q]}$. We plot $S$ in Figure (16) below.

Notice that in (166) the entropy decreases monotonically without bound, with $S - S_0 \sim -\frac{3}{2} \log q$ for large $q$. Presumably higher topologies are relevant in the regime where $S$ is negative.

We could also imagine that there are multiple non-interacting very heavy free fields in the bulk. Then we could imagine a wormhole with particles of different species. Then to compute the Renyi entropies, we would need to compute traces like $\text{tr}(M_1 M_2 M_3 \cdots)(M_1 M_2 M_3 \cdots)^\dagger$. Each of these $M_i$ is an independently drawn Gaussian matrix; since the matrices do not commute, a more general state would involve "words" of matrices in various different orders. Computing such correlators is an exercise in "free probability." We will leave a general discussion for the future. Similar issues have been discussed in [29].

We can also consider a very light operator $\Delta = 0$. For a light operator, the Witten diagram part of the computation becomes independent of the positions, since $e^{-\Delta \ell_{ij}} \to 1$ as $\Delta \to 0$. We just get a combinatoric factor due to the different possible Wick contractions.[13] Then

$$Z_n/(Z_1)^n = L^{1-n}(2n-1)!! = L^{1-n} 2^n \frac{\Gamma(n + \frac{1}{2})}{\sqrt{\pi}}, \tag{167}$$
$$S = \log L - 2 + \gamma_E + \log(2).$$

Curiously, the entropy deficit $S - S_0 = -0.7296...$ is actually *larger* for a light dimension operator than it is for a heavy operator $S - S_0 = -1/2$. Similarly, we can generalize this computation to inserting $q$ particles of the same species, all separated by long Euclidean evolution. We get

$$S - S_0 = \log((2q-1)!!) - q\left(\psi^{(0)}\left(q + \frac{1}{2}\right) + \log(2)\right). \tag{168}$$

Notice that at large $q$, we get approximately $S - S_0 \sim -q$. Hence there is a unitarity problem when $q \sim S_0$.

In Appendix K we study the relative entropy using similar techniques. This quantity is relevant for bulk reconstruction.

---

[13]This might be modified at large $n$ but we are interested in the small $n$ behavior. We expect that the Gaussian spectrum is valid to linear order in $\Delta$, since $e^{-\Delta \ell} = 1 - \Delta \langle \ell_{ij} \rangle = 1 - \Delta \langle \ell \rangle$, where $\langle \ell \rangle$ is the length of the empty wormhole. This linear piece in $\Delta$ leads to a linear in $n$ contribution to $Z_n$ so that it does not contribute to the entropy.

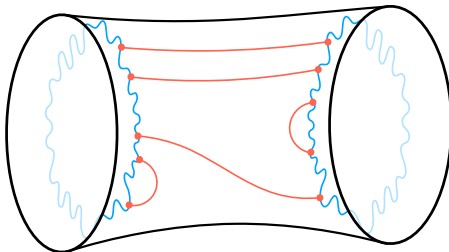

Figure 17: A sample contribution to the "matter" spectral form factor $|Z(x + iy)|^2$. All the Euclidean times between the matter insertions (red points) are infinite.

## 5.4 Eigenvalue repulsion in the projected operators

As discussed above, the projected operator $\hat{O}_\Delta$ for $\Delta \gg 1$ has a semi-circle spectrum which suggests that the projected operator possesses random matrix behavior. Although we did not compute the spectrum for general $\Delta$, we believe that for $\Delta > 0$ the spectrum will have a square root edge. This motivates us to ask whether there is eigenvalue repulsion in the spectrum of $\hat{O}$.[14] To do so, we can compute the spectral form factor of $\hat{O}$, e.g.,

$$|Z(x + iy)|^2 = \text{tr}\left[\exp\left((x + iy)\hat{O}\right)\right]\text{tr}\left[\exp\left((x - iy)\hat{O}\right)\right].\tag{169}$$

Note that we have used $x, y$ to emphasize that the "times" and "temperatures" in this spectral form factor have nothing to do with the boundary times, which are infinite. We can expand the connected part of the form factor

$$\left\langle |Z(x + iy)|^2 \right\rangle_c = \sum_{j,k} \frac{(x + iy)^j (x - iy)^k}{j!k!} \left\langle \text{tr}\,\hat{O}^j\,\text{tr}\,\hat{O}^k \right\rangle_c.\tag{170}$$

The leading contribution to $\left\langle \text{tr}\,\hat{O}^j\,\text{tr}\,\hat{O}^k \right\rangle_c$ is given by cylinder diagrams with propagators (including propagators which go between the same and different boundaries), see Figure 17 for an example with $j = k = 5$. In the $\Delta \gg 1$ limit, only the uncrossed diagrams contribute. Furthermore, each possible uncrossed Wick contraction gives simply a constant $a^{j+k}$ in the infinite Euclidean time limit, where $a$ is the 2-pt function. We see that the gravity answer has the same form as that of a Gaussian random matrix. This implies that the spectral form factor has a linear ramp $\left\langle |Z|^2 \right\rangle_c \propto y$, see e.g. [34].

Although our derivation of the ramp is only valid for large $\Delta$, we expect a ramp-like behavior for $\Delta > 0$ due to random matrix universality. It would be interesting to understand what happens to the divergences that come from particles wrapping the cylinder.

## 5.5 The matter Casimir

As we have already emphasized, the boundary theory has zero Hamiltonian in the extremal limit. One might wonder therefore whether the conformal dimension of an operator can even be defined without referencing a higher dimensional theory. In this subsection we point out the existence of an interesting operator that lets us classify the matter inside the wormhole, which allows us to read off $\Delta$ purely within the topological theory.

In the disk approximation, the gravity theory Hilbert space for the two sided wormholes can be viewed as $\mathcal{H} = (\mathcal{H}_L \times \mathcal{H}_R \times \mathcal{H}_m)/OSp(2|2)$, where $\mathcal{H}_L$ is the Hilbert space of the left

---

[14]Note that the operator $O_\Delta$ might be simple, but projecting $O_\Delta \to \hat{O}_\Delta$ to this 0 energy subspace might make it very complicated. For example, if we start with the SYK fermions, there is obviously no eigenvalue repulsion in the UV, but there can be once we project the fermions into the zero energy sector.



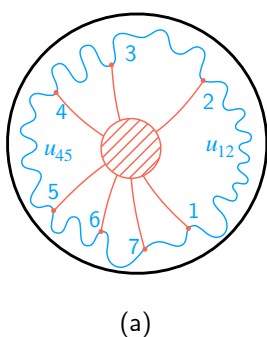
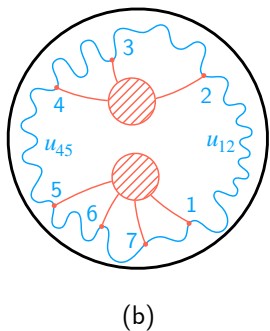

(a)                                        (b)

Figure 18: (a) A connected diagram. (b) A disconnected diagram. The former is suppressed relative to the latter.

side boundary gravity mode, $\mathcal{H}_R$ the right one and $\mathcal{H}_m$ the Hilbert space of (supersymmetric) matter in AdS$_2$. The quotient means that we keep just the states that are invariant under the overall group, which we are viewing as a gauge constraint. The matter Casimir commutes with the gauge constraints and is a physical operator of this theory. Using the constraints we can also express it as the Casimir for the sum of the generators acting on the left plus the right boundary graviton modes (e.g. $-\mathcal{L}_m = \mathcal{L}_1 + \mathcal{L}_2$). Of course, this gives an explicit expression in terms of the sum of generators given in (102) acting on the left plus the right particles. (Both are left generators in the sense of (102)). See [35,36] for a discussion in the $\mathcal{N} = 0$ case.

It would be nice to identify the matter Casimir in the boundary theory (or its $S_0 \to \infty$ limit) since it is directly related to the bulk time generator. Such an operator would necessarily involve two-sided operators. One expects that it can be written in terms of the length mode and its superpartners, and their conjugate variables. So *a priori*, it seems that finding such an operator would be closely related to finding a boundary expression for the length of the wormhole. Notice that the spectrum of this generator is telling us about the spectrum of the quantum field theory in AdS$_2$. Said slightly differently, the fact that the matter casimir can be non-zero while the boundary energies are zero seems to be related to the emergence of a bulk time.

## 6 The low energy limit in ordinary, non-supersymmetric JT gravity

As we remarked the correlation functions for ordinary JT gravity have the structure of some propagators dressing a correlator in rigid AdS$_2$. Here we point out that at long euclidean times these correlators have a simple and universal time dependence

$$\langle O_1(u_1) \cdots O_n(u_n) \rangle_{\text{con}} = \left[ \frac{1}{\prod_{i=1}^n (u_{i+1} - u_i)^{3/2}} \right] \tilde{C}, \qquad (171)$$

where $\tilde{C}$ is independent of time. Here $u_{i+1} - u_i$ is the euclidean time distance between the various correlators. Notice that we have $0 < u_i < u_{i+1} < \beta$ and $u_{n+1} - u_n = \beta - (u_n - u_1)$. This formula generalizes similar results for two and four point functions in [8].

This formula is true for a connected correlator, where the bulk diagram cannot be cut into two separate diagrams. See figure 18 for examples of connected and disconnected correlators. The reason for this simple time dependence is the following. The propagator involves a term

of the form [11]

$$P(1,2) \propto e^{-\varphi_{12}} \int ds\, s \sinh(2\pi s) e^{-s^2 u_{12}} \nu K_{2is}(\nu), \quad \nu = 2e^{-\frac{\ell_{12}}{2}}. \tag{172}$$

When $u \gg 1$ we can restrict to $s \sim 0$. If we set $s = 0$ in the Bessel function term, and approximate the other two terms as

$$\int ds\, s^2 e^{-s^2 u} \propto \frac{1}{u^{3/2}} \quad \Rightarrow \quad P \propto \frac{1}{u_{12}^{3/2}} P_0, \quad P_0 = e^{-\varphi_{12}} \nu K_0(\nu). \tag{173}$$

We then obtain (171) by making this approximation for each correlator. In this approximation, then the constant $\tilde{C}$ involves a $u$ independent equation where the propagator replaced by $P_0$ in (173). Therefore the constant $\tilde{C}$ is computed using these "zero energy" propagators, and it depends on bulk couplings, the masses of the fields, etc.

This procedure fails when we have a disconnected diagram as we see in figure 18b. In this case the propagators going between 12 and 45 will be unsuppressed at long distances. In other words, $K_0$ is not normalizable at long distances. In writing (173), we assumed that there is some matter propagating from 1 to 2, which would exponentially suppress the integrand at large distances, giving a finite answer. But for a disconnected diagram, there is no suppression factor, we cannot justify the approximation leading to (173) for the two propagators connecting the two disconnected (in the bulk) blobs. However, we can still approximate the long distance form of the propagators in terms of the small argument behavior of the Bessel function and the spacetime integrals lead to a delta function relating the $s$ parameters of both propagators. After this, we can then use the approximation similar to (172) but we now get a factor of

$$\frac{1}{(u_{12} + u_{45})^{3/2}}, \tag{174}$$

which is larger than the naive factor of $1/(u_{12} u_{45})^{3/2}$ we would have naively obtained from two separate propagators. So, in the end for disconnected diagrams we still get a simple time dependence, but with this new rule (174).[15]

The final conclusion of this discussion is that the low energy limit of the pure gravity case is conceptually similar to the low energy limit in the $\mathcal{N} = 2$ case. Namely, in both cases the boundary time dependence simplifies, and the constant factor encodes an AdS$_2$ Witten diagram dressed with a particular $u$ independent "zero energy" propagator. A very minor complication is that we have a slightly different formula for the disconnected diagrams.[16] This residual $u$ dependence in the non-supersymmetric case is also related to the fact that the entropy is decreasing. In both cases the structure of the AdS$_2$ theory seems to be related to the projection from the UV Hilbert space to the IR Hilbert space.

In this case, we can also make the discussion about the random matrices, and we also get a gaussian random matrix for a scalar operator. In this case, the non-planar contractions are suppressed because of the disconnected vs connected time dependence we discussed above so that this is valid for any conformal dimension. This a special case of the general connection between JT gravity plus bulk fields and random matrices originally proposed in [29].

## 7 Discussion

In this paper, we explored aspects of the very low energy behavior of black holes. This regime is interesting because the quantum gravity corrections are very large but calculable.

---

[15]For the particular example of figure 18b, we get a time dependence $\left[(u_{12} + u_{45})u_{23}u_{34}u_{56}u_{67}u_{71}\right]^{-3/2}$.

[16]This difference implies that the OTOC is smaller than the TOC by a factor of $1/u^{3/2}$ (in Euclidean time) [8].

We focused on supersymmetric black holes because they are conceptually simpler, since there is a clean low energy limit when there is a gap between the zero energy states and the continuum.[17] The problem is technically harder, because one needs to keep track of the fermionic variables. However, the final formulas, specially for the zero energy propagator, are relatively simple and hopefully could be used for further analysis.

We have computed the two point function as a function of Euclidean or Lorentzian time. This two point function has a constant limit as the time between the two operators goes to infinity $u, u' \to \infty$. This constant value reflects the two point function of the operator after it has been projected on to the zero energy subspace. This is giving us information about the microstates and how well they can be distinguished by the operators, as we discussed in more detail in [1].

In addition, we have argued that all $n$ point functions go to constants at long times, and we have given an integral formula (143) to compute it. These correlators are sensitive to interactions among the bulk fields. It would be interesting to know whether we can reconstruct the bulk theory from such correlators. This expression involves the zero energy propagator of the boundary (super) gravity mode. It would be nice to find the propagator for arbitrary boundary energies (or boundary times). For the $\mathcal{N} = 1$ case we have given such a propagator in appendix C.

We have compared some of these results to numerical simulations in $\mathcal{N} = 2$ SYK for $N = 16$ (sixteen complex fermions). We have found close agreement for the energy gap and the two point functions. We have also found that the ratios of OTOC to TOC are also of order one, as indicated by the general expressions of $n$ point functions. We have also plotted the eigenvalues of the IR operators, after the projection to the ground state.

The bulk theory, at the disk level, contains an interesting operator which is is the matter Casimir operator, see section 5.5. This commutes with the boundary Hamiltonian and seems related to the bulk time direction, since its spectrum contains the spectrum of bulk matter particles. We have not given an expression for this interesting operator from the point of view of the boundary quantum mechanical theory. Of course, the even simpler distance operator does not have a clear interpretation in the boundary theory [37, 38].

The entropies of certain extremal black holes can be computed using certain supersymmetric indices. Here we have discussed a new class of extremal black holes with matter inside the wormhole. In certain cases we were able to compute the entropy using the bulk picture. It would be interesting if supersymmetric techniques could be employed to compute (or at least bound) such entropies from the boundary point of view.

We have remarked that many of the conceptual issues are similar in the non-supersymmetric case. In that case, the boundary correlators of elementary fields develop a universal boundary time dependence. They are multiplied by constants that depend on the masses and couplings of the bulk theory. Here the derivation of the propagator is simpler [10, 11] but we have to be more careful in keeping track of some of this time dependence, as discussed in section 6. We also have to distinguish between connected and disconnected bulk diagrams.

The authors of [24] developed a technique to perform computations in a double scaled $\mathcal{N} = 2$ SYK limit. It would be interesting to understand the properties of zero energy correlators for any value of the non-trivial $\hat{q}^2/N$ parameter of that limit.

---

[17]Even with $\mathcal{N} = 2$ supersymmetry, whether this gap is present or not depends in detail on the precise spectrum of charges and the possible presence of a certain $\theta$-like term, see [9]. We have focused on cases where the gap is present.

## Acknowledgments

We would like to thank Daniel Jafferis, Henry Maxfield, Baurzhan Mukhametzhanov, Geoffrey Penington, Douglas Stanford, Phil Saad and Stephen Shenker for discussions. We also thank specially Gustavo Turiaci for his explanations of the super-Liouville methods in [17] and Vladimir Narovlansky for his explanation of the results in [24] as well as for providing us with the JT gravity limits of formulas in that work that were very useful for us.

J.M. is supported in part by U.S. Department of Energy grant DE-SC0009988 and by the Simons Foundation grant 385600.

## A   From the Schwarzian action to the Liouville action

The JT gravity action can be written as [3–5]

$$I = -\frac{1}{4\pi}\left[\int \phi(R+2) + 2\phi_b \int K\right] \longrightarrow -\frac{\phi_b}{2\pi}\int \mathrm{d}\tau\{f,\tau\}, \tag{A.1}$$

where $\phi$ is normalized so that the entropy is given just by $S = \phi_h$ where $\phi_h$ is the value at the horizon. The arrow in the right hand side means that if we take $\phi_b$ large and we go near the boundary, then the action reduces to the Schwarzian expression where $\tau$ is the proper time near the boundary. It is sometimes customary to defined a rescaled version of these $\phi_b = \frac{\phi_r}{\epsilon}$, $t = \epsilon\tau$ that remains constant as we take the boundary limit with $\epsilon \to 0$. We could also do the further rescaling $u = \frac{\pi}{\phi_r}t$ which sets the coefficient of the Schwarzian to a half

$$I = -\frac{1}{2}\int \mathrm{d}u\{f,u\} = \int \frac{1}{4}\dot{\ell}^2 + \pi_f(e^{-\ell} - f'). \tag{A.2}$$

A simple way to connect the parameter $\phi_r$ to the parameters of black holes is to compute their near extremal entropy as a function of the inverse temperature and compare with

$$S - S_0 = \frac{2\pi\phi_r}{\beta}. \tag{A.3}$$

For example, for the four dimensional Reissner Nordstrom black hole we have

$$S - S_e = \frac{4\pi}{4G_N}(r_+^2 - r_+ r_-) = S_e\frac{4\pi r_e}{\beta}, \quad \beta = 4\pi\frac{r_+^2}{(r_+ - r_-)}, \quad r_+ - r_- \ll r_+, \tag{A.4}$$

so that $\phi_r = 2S_e r_e$, where $r_e$ is the extremal radius, $r_e^2 = r_+ r_-$.

In these variables, we can think of the three SL(2) generators as

$$L_- = -\partial_f, \quad L_0 = -f\partial_f + \partial_\ell, \quad L_+ = -f^2\partial_f + 2f\partial_\ell - e^{-\ell}, \tag{A.5}$$

where $\pi_f$ is the momentum conjugate to $f$ and $\pi_\ell$ the momentum conjugate to $\ell$.

When we consider the TFD, it is convenient to choose a new variable $\tilde{f} = 1/f$, $\tilde{\ell} = \ell + 2\log x$ related via an SL(2) transformation. This can be used to describe the left particle, we also used that $\tilde{\gamma} = \gamma - e^{-\ell}/f$ and implicitly assumed that the last term in (A.5) has a $\partial_\gamma$ acting on a wavefunction with an $e^\gamma$ dependence, $e^{-\ell} \to e^{-\ell}\partial_\gamma$. We can then express $L_a$ in terms of $\tilde{f}$.

$$L_-^l = \tilde{f}^2\partial_{\tilde{f}} - 2\tilde{f}\partial_{\tilde{\ell}} - e^{-\tilde{\ell}}, \quad L_0^l = \tilde{f}\partial_{\tilde{f}} - \partial_{\tilde{\ell}}, \quad L_+^l = \partial_{\tilde{f}}. \tag{A.6}$$

We then make a gauge choice where we set $\tilde{f}_l = 0$ and $\tilde{f}_l' = 1$. In addition, we make the choice $f_r = 0$, but no condition on $f_r'$. The renormalized distance is then[18]

$$e^d \sim \frac{(t_1 - t_2)^2}{z_1 z_2} \quad \longrightarrow \quad e^{-\ell} = \frac{f_l' f_r'}{(f_l - f_r)^2} = \frac{\tilde{f}_l' f_r'}{(1 - \tilde{f}_l f_r)^2} \to f_r', \tag{A.7}$$

where $d = \ell - 2\log\epsilon$, with $\epsilon$ as discussed after (A.1).

In addition, we have that the $L_-^l$ expression becomes

$$L_-^l = -1, \tag{A.8}$$

and the gauge constraint $L_-^l + L_-^r = 0$ implies that $\pi = 1$. Then the Lagrangian then becomes (in Euclidean time)

$$I = \int \frac{1}{4}\dot{\ell}^2 + e^{-\ell}, \tag{A.9}$$

which is what we used in (2). We can view the other constraints as determining the $\tilde{\ell}$ and $\tilde{f}$ dependence away from the point $\tilde{\ell} = \tilde{f} = 0$.

# B  Away from the Schwarzian limit for the bosonic case

The description of the boundary gravitational mode in terms of an SL(2)/U(1) quotient was discussed in [10, 11, 39]. Here we just mention a few points about the form of the problem away from the Schwarzian limit.

Let us first review the case of $H_2$ space. Then we can write $g \in$ SL(2) as

$$g = e^{i\phi\sigma_2/2} e^{\sigma_3\rho/2} e^{i\chi\sigma_2/2}, \tag{B.1}$$

Then the left invariant forms can be defined as

$$g^{-1}\,\mathrm{d}g = \frac{1}{2}w_a\sigma^a = \frac{1}{2}\left[e^{-i\chi\sigma_2/2}e^{-\sigma_3\rho/2}i\sigma_2 e^{\sigma_3\rho/2}e^{i\chi\sigma_2/2}\,\mathrm{d}\phi + e^{-i\chi\sigma_2/2}\sigma_3 e^{i\chi\sigma_2/2}\,\mathrm{d}\rho + i\sigma_2\,\mathrm{d}\chi\right]$$

$$= \frac{1}{2}\left[i\sigma_2(\mathrm{d}\chi + \cosh\rho\,\mathrm{d}\phi) + e^{-i\chi\sigma_2/2}(\mathrm{d}\rho\,\sigma_3 - \sinh\rho\,\sigma_1\mathrm{d}\phi)e^{i\chi\sigma_2/2}\right]. \tag{B.2}$$

We quotient under the action $g \to g e^{i\alpha\sigma_2}$. This breaks the right SL(2) symmetry to U(1). This shifts the $\chi$ coordinate, $\chi \to \chi+$constant. The left SL(2) symmetry remains and will be the isometries of $H_2$. This gives the usual left invariant forms

$$w_2 = i(\mathrm{d}\chi + \cosh\rho\,\mathrm{d}\phi), \quad w_1 = \mathrm{d}\rho\sin\chi - \sinh\rho\,\mathrm{d}\phi\cos\chi, \quad w_3 = \mathrm{d}\rho\cos\chi + \sinh\rho\,\mathrm{d}\phi\sin\chi. \tag{B.3}$$

The metric in hyperbolic space is simply

$$w_1^2 + w_3^2 \propto \mathrm{d}\rho^2 + \sinh^2\rho\,\mathrm{d}\phi^2, \tag{B.4}$$

which is invariant under the qoutient group, as expected. Because of this reason we do not need to calculate the full $\chi$ dependence of the $w_i$ to get the metric (B.4). We can then write an invariant action of the form

$$S = \int \mathrm{d}t\frac{1}{2}(w_1^2 + w_3^2) + \int \mathrm{d}t q i(w_2 - iA) = \int \mathrm{d}t\frac{1}{2}(\dot{\rho}^2 + \sinh^2\rho\,\dot{\phi}^2) - q\int \cosh\rho\,\dot{\phi}, \tag{B.5}$$

---

[18]This comes from the formula $\cosh d = -Y_1.Y_2 = \frac{(t_1-t_2)^2 + z_1^2 + z_2^2}{2z_1 z_2}$ and we have rescaled $z \to \epsilon z$.

where we have added a term involving $w_2 - A$ where $A$ is the gauge field. We choose the gauge $\chi = 0$ and integrate out $A$. In the end, a term like $iq(w_2 - A)$ give us the magnetic-like coupling for the particle moving on $H_2$.

This action, (B.5), is also equivalent to starting with the full naive $G \times G$ invariant action and gauging the quotient group. The charge of the magnetic field coupling is imaginary. The reasonable coupling to the magnetic field would be one where $q$ is imaginary. However in our problem q is real [10, 11]. We can take the large $q$ limit, rescaling also $\rho$ as $e^\rho = qe^{\check\rho}$. Then the first term combines with the magnetic field coupling to get

$$\frac{1}{2}q^2(\frac{1}{2}e^{\check\rho}\dot\phi - 1)^2 - \frac{1}{2}q^2\,. \tag{B.6}$$

The last term is a constant that we can neglect. The first term imposes that $e^{-\check\rho} = \dot\phi/2$. We can then now substitute this back into the remaining terms of (B.5) to obtain

$$\int \mathrm{d}t \frac{1}{2}\big[\dot{\check\rho}^2 - \dot\phi^2\big] = \int \mathrm{d}t \frac{1}{2}\left[\left(\frac{\ddot\phi}{\dot\phi}\right)^2 - \dot\phi^2\right] = -\int \mathrm{d}t\{\tan\frac{\phi}{2}, t\}\,, \tag{B.7}$$

where we have neglected some total derivatives. So we see that we get the usual Schwarzian action.

The wavefunctions can be viewed as representations where we diagonalize the right factor. In other words, we can consider

$$\Psi_{j,m,q}(\rho, \phi) = \langle jm|g(\rho, \phi)|j, q\rangle\,, \tag{B.8}$$

where we have removed the right most factor in $g$ (B.1). For more details see [10, 11].

## B.1 Generators

It is useful to write the generators of $SL(2)$. These are obtained by acting with an SL(2) generator on $g$ and then writing the answer as $g(x + \delta x)$ where $x = (\phi, \rho, \chi)$. In this way we get

$$\begin{aligned}
i\sigma_2/2 &\;\to\; \partial_\phi\,, \\
\sigma_3/2 &\;\to\; -\frac{\sin\phi}{\tanh\rho}\partial_\phi + \frac{\sin\phi}{\sinh\rho}\partial_\chi + \cos\phi\,\partial_\rho\,, \\
\sigma_1/2 &\;\to\; -\frac{\cos\phi}{\tanh\rho}\partial_\phi + \frac{\cos\phi}{\sinh\rho}\partial_\chi - \sin\phi\,\partial_\rho\,.
\end{aligned} \tag{B.9}$$

The quotient is obtained by setting $\partial_\chi = q$. But now we could construct the Casimir which gives

$$C_2 \sim \partial_\rho^2 + \frac{1}{\tanh\rho}\partial_\rho + \frac{1}{\sinh^2\rho}(\partial_\phi - \cosh\rho\,\partial_\chi)^2 - \partial_\chi^2\,. \tag{B.10}$$

We can then identify this with the energy. This is the problem away from the Schwarzian limit [10, 11], we can then further take the large $q$ limit to obtain the Schwarzian limit.

## C  $\mathcal{N} = 1$ correlators

In this section we discuss some aspects about the $\mathcal{N} = 1$ super-Schwarzian theory. This case was analyzed in depth in [13] to whom we refer for further details (our conventions are slightly different). Here we will work out the boundary particle propagator.

We now have the supergroup $OSp(1|2)$ whose algebra is

$$[L_m, L_n] = (m-n)L_{m+n},$$
$$\{G_r, G_s\} = 2L_{r+s},$$
$$[L_m, G_r] = \left(\frac{m}{2} - r\right)G_{m+r}. \tag{C.1}$$

We can introduce coordinates $x, \rho, \gamma, \theta_\pm$ associated to each of the five generators. Writing the group element as

$$g = e^{-xL_-} e^{\theta_- G_-} e^{\rho L_0} e^{\theta_+ G_+} e^{\gamma L_+}, \tag{C.2}$$

we get the following expressions for the left generators

$$\mathcal{L}_- = -\partial_x,$$
$$\mathcal{L}_0 = -x\partial_x + \partial_\rho - \frac{1}{2}\theta_-\partial_{\theta_-},$$
$$\mathcal{L}_+ = e^{-\rho}\partial_\gamma - x^2\partial_x + x\left(2\partial_\rho - \theta_-\partial_{\theta_-}\right) - e^{-\rho/2}\theta_- D_+,$$
$$\mathcal{G}_+ = e^{-\rho/2}D_+ + x(\partial_{\theta_-} - \theta_-\partial_x) + 2\theta_-\partial_\rho,$$
$$\mathcal{G}_- = \partial_{\theta_-} - \theta_-\partial_x, \quad \text{with} \quad D_+ = \partial_{\theta_+} + \theta_+\partial_\gamma. \tag{C.3}$$

We also have the right generators

$$\mathcal{L}_+^R = \partial_\gamma, \quad \mathcal{G}_+^R = \partial_{\theta_+} - \theta_+\partial_\gamma, \quad \{\mathcal{G}_+^R, \mathcal{G}_+^R\} = -2\mathcal{L}_+^R, \tag{C.4}$$

and the Casimir

$$\mathcal{C} = L_0^2 - \frac{1}{2}(L_+L_- + L_-L_+) - \frac{1}{4}[G_-, G_+],$$
$$\mathcal{C} = \partial_\rho^2 + \frac{1}{2}\partial_\rho + e^{-\rho}\partial_x\partial_\gamma - \frac{1}{2}e^{-\rho/2}D_-D_+, \quad D_- = \partial_{\theta_-} + \theta_-\partial_x. \tag{C.5}$$

The spacetime supercharge is

$$Q = (\partial_\rho + \frac{1}{4})D_+ - e^{-\rho/2}\partial_\gamma D_-, \tag{C.6}$$

which obeys $Q^2 = \partial_\gamma(\mathcal{C} + \frac{1}{16})$. We can define the Hamiltonian to be $H = -(\mathcal{C} + \frac{1}{16})$.

An interesting fact about the OSp(1|2) algebra is that there is a superCasimir or sCasimir element $\mathcal{Q} = G^-G^+ - G^+G^- + 1/8$ which is Grassmann-even, commutes with all the bosonic generators, but anti-commutes with all the fermionic elements. There is a simple relation between the right sCasimir and the supercharge:

$$Q = 4\mathcal{Q}^R\mathcal{G}_+^R. \tag{C.7}$$

Note that this definition of $Q$ is manifestly left-invariant. In addition this notation makes it clear that $Q$ anti-commutes with $\mathcal{G}_+^R$.

## C.1 Invariants

Here we record the form of invariants of two coordinates

$$w = e^{\rho_1/2 + \rho_2/2}(x_1 - x_2 - \theta_{1-}\theta_{2-}), \tag{C.8}$$

$$\eta_1 = \theta_{1+} - \frac{e^{\rho_2/2}(\theta_{1-} - \theta_{2-})}{w}, \tag{C.9}$$

$$\eta_2 = \theta_{2+} - \frac{e^{\rho_1/2}(\theta_{1-} - \theta_{2-})}{w}, \tag{C.10}$$

$$\Phi = \gamma_1 - \gamma_2 + \frac{e^{(\rho_1-\rho_2)/2} + e^{-(\rho_1-\rho_2)/2}}{w} + \frac{(\theta_{1-} - \theta_{2-})(-e^{\rho_2/2}\theta_{1+} + e^{\rho_1/2}\theta_{2+})}{w}. \tag{C.11}$$

We can write the supercharge in terms of invariants

$$Q_1 = \frac{1}{2}(w\partial_w + \frac{1}{2})(\partial_{\eta_1} + \eta_1\partial_\Phi) + \frac{1}{w}\partial_\Phi(\partial_{\eta_2} - \eta_2\partial_\Phi). \tag{C.12}$$

We can also write the right supersymmetries in terms of invariants

$$\mathcal{G}^R_{+1} = \partial_{\eta_1} - \eta_1\partial_\Phi, \quad \mathcal{G}^R_{+2} = \partial_{\eta_2} + \eta_2\partial_\Phi, \tag{C.13}$$

which anticommute with the supercharge. Note that we can interpret these as extra fermions which do not appear in the Liouville description, which can be interpreted as the terms in parenthesis in (C.12). We could remove this extra degree of freedom by imposing a condition of the form

$$(\mathcal{G}^R_1 + \mathcal{G}^R_2)P = 0. \tag{C.14}$$

This condition removes the extra unwanted fermionic degrees of freedom.[19] Note that the operator in the left hand side squares to zero when acting on the propagator that is annihilated by $\mathcal{L}^R_{1+} + \mathcal{L}^R_{2+}$, due to

$$\mathcal{L}^R_1 P = -\mathcal{L}^R_2 P = -\mathfrak{q}P. \tag{C.15}$$

Note that we can "improve" the invariants by defining

$$\tilde{\Phi} = \Phi - \eta_1\eta_2, \quad \tilde{\eta} = \eta_1 - \eta_2. \tag{C.16}$$

Together with $w$, we have three invariants are annihilated by $\mathcal{G}^R_{+1} + \mathcal{G}^R_{+2}$.

## C.2 Propagator

Here we will compute the propagator for arbitrary times. In writing the propagator, it is convenient to think in terms of a worldline superspace propagator

$$P = \langle 1|e^{\kappa_1 Q}e^{-(u_1-u_2)H}e^{-\kappa_2 Q}|2\rangle = e^{-(u_{12}-\kappa_1\kappa_2)H}e^{(\kappa_1-\kappa_2)Q}, \quad \text{with} \quad Q^2 = H, \tag{C.17}$$

where $\kappa_i$ are the superspace coordinates of the worldline theory. We define the covariant derivatives $D_\kappa = \partial_\kappa + \kappa\partial_u$. Note that the propagator can depend only on the combinations $\kappa_1 - \kappa_2$ and $U = u_1 - u_2 - \kappa_1\kappa_2$. We can view this as a consequence of worldline supersymmetry.[20] In addition we see that we have that $Q_1 P = D_{\kappa_1} P$. When we compose the propagator as in $P(1,2)P(2,3)$ we do not integrate over $\kappa_2$ (as we do not integrate over $u_2$), we expect that the result should be independent of it.

We now make an ansatz for the propagator which is a parity even function of the invariants of the form

$$\hat{P} = e^{-\mathfrak{q}\tilde{\Phi}}[A + (\kappa_1 - \kappa_2)\tilde{\eta}C] = e^{-\mathfrak{q}\Phi}[(1 + \mathfrak{q}\eta_1\eta_2)A + (\kappa_1 - \kappa_2)(\eta_1 - \eta_2)C]. \tag{C.18}$$

This ansatz is manifestly invariant under $\mathcal{G}^R_{+1} + \mathcal{G}^R_{+2}$ as well as all the left generators.

We now impose the equation

$$Q_1\hat{P} = \sqrt{\mathfrak{q}}D_{\kappa_1}\hat{P}, \tag{C.19}$$

---

[19]Imposing $\mathcal{G}^R_{+1} + \mathcal{G}^R_{+2} = 0$ is equivalent to imposing that $\mathcal{G}^R$ commutes with the propagator $e^{-uH}$. We could also consider $\mathcal{G}^R_{+1} - \mathcal{G}^R_{+2} = 0$, which would impose that $\mathcal{G}^R$ anti-commutes with the propagator. This would be incorrect since the propagator is bosonic.

[20]These combinations are annihilated by $\tilde{Q}_1 + \tilde{Q}_2$ with $\tilde{Q} = \partial_\kappa - \kappa\partial_u$ being the combination that anticommutes with $D_\kappa$.

where the factor of $\sqrt{\mathfrak{q}}$ arises due to the normalization of $Q_1$ in (C.12).[21] This results in the conditions

$$Q_1(1+\mathfrak{q}\eta_1\eta_2)A = \sqrt{\mathfrak{q}}(\eta_1-\eta_2)C, \quad Q_1(\eta_1-\eta_2)C = -\sqrt{\mathfrak{q}}\partial_U(1+\mathfrak{q}\eta_1\eta_2)A = \sqrt{\mathfrak{q}}E(1+\mathfrak{q}\eta_1\eta_2)A,$$
(C.20)

where we assumed that we are in the subsector of energy $E$. Using the form of the supercharge (C.12), we find that (C.20) implies

$$C = \frac{\sqrt{\mathfrak{q}}}{2}\left[yA'(y)+(y-\frac{1}{2})A(y)\right], \quad y = \frac{2\mathfrak{q}}{w},$$
(C.21)

and an equation for $A$ which is solved by

$$A(y) = y\left[K_{\frac{1}{2}+i2s}(y)+K_{\frac{1}{2}-i2s}(y)\right] = \sqrt{y}\Psi_+(y), \quad y = \frac{2\mathfrak{q}}{w}, \quad E = s^2,$$
(C.22)

where we also defined the function $\Psi_+$ which is related to a super-Liouville wavefunction as we will review below. In summary, the final form of the propagator for fixed energy is (C.18) with $A$ in (C.22) and $C_1$ in (C.21). In addition, we need to integrate over energy to find the fixed $u$ propagator. In principle, that measure factor is determined by demanding the composition law of the propagators. In practice, we can determine it by comparing to the Liouville results in [17].

The integral over each point involves the measure

$$\int d\mu = \int dx\, d\rho\, e^{\rho/2}\, d\theta_-\, d\theta_+,$$
(C.23)

where the factor of $e^{\rho/2}$ is fixed by scaling. Note that $\theta_+$ is neutral under scaling (C.3).

General correlators are given by

$$\langle O_1(u_1,\kappa_1)\cdots O(u_n,\kappa_n)\rangle = \int \frac{\prod_i d\mu_i}{\text{vol}(OSp(1|2))}\hat{P}(1,2)\hat{P}(2,3)\cdots\hat{P}(n,1)$$
$$\times \prod_i e^{-\Delta_i\rho_i}\langle O(x_1,\theta_{1-})\cdots O(x_n,\theta_{n-})\rangle,$$
(C.24)

where the last factor include the expected form of the correlator of bulk fields when we take them near the boundary.

## C.3 Recovering the Liouville two point functions

As a check of this result, we recover the two point functions that were computed in [17] using the Liouville method. In the Liouville method one starts for a superliouville quantum mechanics which is the dimensional reduction of a theory in $1+1$ dimensions with $(1,1)$ supersymmetry. The variables are the wormhole length, $\ell$, and two fermionic partners arising from the action of the supersymmetry acting on the left and the right sides. We will not give the explicit form of the Lagrangian, which can be found in section 6 of [40]. We mention that we have two wavefunctions whose bosonic components have the form

$$\Psi_{s,\pm}(\ell) = e^{-\ell/4}\left[K_{\frac{1}{2}+2is}(2e^{-\ell/2})\pm K_{\frac{1}{2}-2is}(2e^{-\ell/2})\right].$$
(C.25)

And the correlation functions have the form

$$\langle E'|e^{-\Delta\ell}|E\rangle \propto \int d\ell\,\Psi_{s',+}(\ell)e^{-\Delta\ell}\Psi_{s,+}(\ell), \quad E = s^2.$$
(C.26)

---

[21]As discussed after (C.6) we get $Q^2 = \mathfrak{q}H$.

Now we will recover this from our propagator and (C.24) for $n = 2$. First, we choose the gauge conditions

$$x_1 = 1, \quad x_2 = 0, \quad \rho_2 = 0, \quad \theta_{1-} = \theta_{2-} = 0, \tag{C.27}$$

which implies that

$$w = e^{\frac{\ell}{2}}, \quad \ell = \rho_1, \quad \eta_1 = \theta_{1+}, \quad \eta_2 = \theta_{2+}, \tag{C.28}$$

whose Jacobian is trivial. We also set $\kappa_1 = \kappa_2 = 0$ for simplicity. Then the two point function becomes

$$\int d\ell \, e^{\ell/2} \, d\theta_{1+} \, d\theta_{2+} e^{-\ell\Delta} \hat{P}(1,2)\hat{P}(2,1). \tag{C.29}$$

The phase factor $\Phi$ cancels in this expression. We also set $\mathfrak{q} = 1$. After doing the $\theta_+$ integrals we get something proportional to

$$\int d\ell \, e^{\ell/2} \, d\theta_{1+} \, d\theta_{2+} e^{-\ell\Delta} A_{s'}(y)A_s(y), \quad y = 2e^{-\ell/2}, \tag{C.30}$$

which agrees with (C.26) after we use (C.22). Notice that the extra factor of $\sqrt{y}$ in (C.22) cancels agains the extra measure factor in (C.30).

Of course, the full two point function in time also involves and integral over the energies with the appropriate $\rho(E)\rho(E')e^{-u'E'-uE}$ factors.

# D  Invariants in terms of the $\theta_+$, $\bar{\theta}_+$ coordinates

In this appendix we discuss the expression of the invariants in terms of the superspace introduced in (101) (102). We find

$$\begin{aligned}
w &= e^{(\rho_1+\rho_2)/2}\left[x_1 - x_2 - \theta_{1-}\bar{\theta}_{2-} - \bar{\theta}_{1-}\theta_{2-}\right], \\
\Sigma &= e^{i(a_1-a_2)}\left(1 - \frac{(\theta_{1-}-\theta_{2-})(\bar{\theta}_{1-}-\bar{\theta}_{2-})}{(x_1-x_2)}\right), \\
\hat{\Phi} &= \gamma_1 - \gamma_2 + \frac{e^{(\rho_1-\rho_2)/2} + e^{-(\rho_1-\rho_2)/2}}{w} + (e^{\rho_2/2}\bar{\theta}_{1+} - e^{\rho_1/2}\bar{\theta}_{2+})\frac{(\theta_{1-}-\theta_{2-})}{w} + \\
&\quad + (e^{\rho_2/2}\theta_{1+} - e^{\rho_1/2}\theta_{2+})\frac{(\bar{\theta}_{1-}-\bar{\theta}_{2-})}{w}, \\
f_1 &= e^{ia_1}\left(\theta_{1+} - e^{\rho_2/2}\frac{(\theta_{1-}-\theta_{2-})}{w}\right), \\
\bar{f}_1 &= e^{-ia_1}\left(\bar{\theta}_{1+} - e^{\rho_2/2}\frac{(\bar{\theta}_{1-}-\bar{\theta}_{2-})}{w}\right), \\
f_2 &= e^{ia_2}\left(\theta_{2+} - e^{\rho_1/2}\frac{(\theta_{1-}-\theta_{2-})}{w}\right), \\
\bar{f}_2 &= e^{-ia_2}\left(\bar{\theta}_{2+} - e^{\rho_1/2}\frac{(\bar{\theta}_{1-}-\bar{\theta}_{2-})}{w}\right).
\end{aligned} \tag{D.1}$$

The first two are identical to the ones in (121) (122). Note that now we have more fermionic invariants because we have more fermionic variables. However, we also have additional constraints, since we want to demand also invariance under the right generators as in (110). To do so, we again introduce the Grassmann numbers $\chi_1, \chi_2$. These are two additional left-invariant quantities. So we have in total 4 more invariants, but also 4 more constraints coming from (110), $(\mathcal{G}_+^R + \hat{G}_+^R)_{1,2} = (\bar{\mathcal{G}}_+^R + \hat{\bar{G}}_+^R)_{1,2} = 0$.

The equation (115) now says that the two independent fermionic invariants are

$$\eta_1 = \chi_1 + \bar{f}_1, \quad \eta_2 = \chi_2 + \bar{f}_2. \tag{D.2}$$

Similarly, (115) implies that we should define a new phase factor

$$\Phi = \hat{\Phi} + f_1\bar{f}_1 + 2f_1\chi_1 - f_2\bar{f}_2 - 2f_2\chi_2, \tag{D.3}$$

where we use the dependence on $\theta_{+i}$ to constrain it and we wrote it in terms of the invariants we already had.

If we now use the gauge choice $\theta_{+i} = \bar{\theta}_{+i} = 0$, then we see that the invariants (D.2) reduce to (124) and $\Phi$ becomes the $\Phi$ in (123).

# E Computing the matrix elements

In this appendix, we compute matrix elements of various operators in the super-Liouville quantum mechanics.

## E.1 BPS operators

First we consider a matrix element of the state $|F^+\rangle$. The integral involved is the same as for the case with no supersymmetry [17] and we get

$$\langle F^+_{s',j'}|e^{-\Delta(\ell+2ia)}|F^+_{s,j}\rangle = \frac{\sqrt{E'E}}{\pi}\frac{\Gamma(\Delta \pm is \pm is')}{\Gamma(2\Delta)}\delta_{j',j-2\Delta}. \tag{E.1}$$

Now, in order to compute the other matrix elements we use that the other wavefunctions are given by

$$|H_{s,j}\rangle = \frac{1}{\sqrt{E}}\bar{Q}_l|F^+_{s,j}\rangle, \quad |L_{s,j}\rangle = \frac{1}{\sqrt{E}}i\bar{Q}_r|F^+_{s,j}\rangle. \tag{E.2}$$

In addition, we can use commutation relations

$$[\bar{Q}_r, e^{-\Delta(\ell+2ia)}] = [Q_l, e^{-\Delta(\ell+2ia)}] = 0. \tag{E.3}$$

This then leads to a simple evaluation of the matrix elements among $H$ or among $L$. In those cases we commute either the $\bar{Q}_r$ or the $Q_l$. For example, for

$$\begin{aligned}
\langle H_{s',j'}|e^{-\Delta(\ell+2ia)}|H_{s,j}\rangle &= \frac{1}{\pi\sqrt{EE'}}\langle F^+_{s',j'}|Q_l e^{-\Delta(\ell+2ia)}\bar{Q}_l|F^+_{s,j}\rangle \\
&= \frac{1}{\pi\sqrt{EE'}}\langle F^+_{s',j'}|e^{-\Delta(\ell+2ia)}Q_l\bar{Q}_l|F^+_{s,j}\rangle \\
&= \frac{1}{\pi\sqrt{EE'}}E\langle F^+_{s',j'}|e^{-\Delta(\ell+2ia)}|F^+_{s,j}\rangle \\
&= \frac{E}{\pi}\frac{\Gamma(\Delta \pm is \pm is')}{\Gamma(2\Delta)}\delta_{j',j-1}.
\end{aligned} \tag{E.4}$$

We can do the same for the $|L\rangle$ states and we get a similar answer, except that now we move the $\bar{Q}_r$ to the left side so we get a factor of $E'$ rather than $E$. In this way we can also see that

$$\langle H_{s',j'}|e^{-\Delta(\ell+2ia)}|L_{s,j}\rangle = 0, \tag{E.5}$$

by commuting the $Q_l$ from the left side to the right side For the case

$$
\begin{aligned}
\langle L_{s',j'}|e^{-\Delta(\ell+2ia)}|H_{s,j}\rangle &= -i\frac{1}{\pi\sqrt{EE'}}\langle F^+_{s',j'}|Q_r e^{-\Delta(\ell+2ia)}\bar{Q}_l|F^+_{s,j}\rangle \\
&= -i\frac{1}{\pi\sqrt{EE'}}\langle F^+_{s',j'}|\{[Q_r,e^{-\Delta(\ell+2ia)}],\bar{Q}_l\}|F^+_{s,j}\rangle \\
&= -i\frac{1}{\pi\sqrt{EE'}}\langle F^+_{s',j'}|(2i\Delta e^{-\ell/2+ia}+4\Delta^2\psi_r\bar{\psi}_l)e^{-\Delta(\ell+2ia)}|F^+_{s,j}\rangle \\
&= \frac{2\Delta}{\pi\sqrt{EE'}}\langle F^+_{s',j'}|e^{-(\Delta+\frac{1}{2})\ell-ia(2\Delta-1))}|F^+_{s,j}\rangle \\
&= \frac{2\Delta}{\pi}\frac{\Gamma(\Delta+\frac{1}{2}\pm is\pm is')}{\Gamma(2(\Delta+\frac{1}{2}))}\delta_{j',j-2\Delta+1}\,,
\end{aligned}
\tag{E.6}
$$

where we used that the operators annihilate the appropriate state to write it just as some commutators and anticommutators of the operator in question.

When we want to consider the zero energy states $Z_j$, then it is convenient to obtain these from limits of $|H\rangle$ as $s \to \frac{i}{2}(j-1/2)$. And it is convenient to start with the matrix elements in (E.6) which do not involve explicit factors of $E$ that are going to zero. So, starting for $\langle L|e^{-\Delta(\ell+2ia)}|H\rangle$ and taking the limit, we get the matrix element $\langle L|e^{-\Delta(\ell+2ia)}|Z\rangle$. Similarly, we could start with $\langle H|e^{-\Delta(\ell+2ia)}|H\rangle$ and take the limit on the $H$ on the left, setting $s'=\pm\frac{i}{2}(j-\frac{1}{2})$ to obtain the $\langle Z|e^{-\Delta(\ell+2ia)}|H\rangle$ matrix element. Further talking a limit of this one we get the $\langle Z|e^{-\Delta(\ell+2ia)}|Z\rangle$ matrix element.

## E.2 Neutral operators

The process for the neutral operator works similarly. We use the fact that each time we use commutator relations, we have two choices for which way to move an operator. We can do the calculation by averaging over these two choices. The basic matrix element needed is

$$
\langle F^+_{s',j'}|e^{-\Delta\ell}|F^+_{s,j}\rangle = \frac{\sqrt{E'E}}{\pi}\frac{\Gamma(\Delta\pm is\pm is')}{\Gamma(2\Delta)}\delta_{j,j'}\,.
\tag{E.7}
$$

It is useful to keep this property in mind:

$$
\{A,BC\} = B\{A,C\}+[A,B]C\,.
\tag{E.8}
$$

Using this relation, we can derive

$$
\{Q_l,[e^{-\Delta\ell},\bar{Q}_l]\}+\{[Q_l,e^{-\Delta\ell}],\bar{Q}_l\} = i\Delta e^{-\Delta\ell}\left(-i\partial_a+\Delta[\psi_l,\bar{\psi}_l]\right),
\tag{E.9}
$$

$$
\{Q_r,[e^{-\Delta\ell},\bar{Q}_r]\}+\{[Q_r,e^{-\Delta\ell}],\bar{Q}_r\} = i\Delta e^{-\Delta\ell}\left(i\partial_a+\Delta[\psi_r,\bar{\psi}_r]\right),
\tag{E.10}
$$

$$
\{Q_l,[e^{-\Delta\ell},\bar{Q}_r]\} = -i\Delta e^{-\ell\Delta}\left(e^{-\frac{\ell}{2}-ia}-i\Delta\psi_l\bar{\psi}_r\right).
\tag{E.11}
$$

We first compute $\langle H_{s',j'}|e^{-\Delta\ell}|H_{s,j}\rangle$:

$$
2\langle H_{s',j'}|e^{-\Delta\ell}|H_{s,j}\rangle = \frac{2}{\pi\sqrt{EE'}}\langle F^+_{s',j'}|Q_l e^{-\Delta\ell}\bar{Q}_l|F^+_{s,j}\rangle
$$

$$
= \frac{E+E'}{\pi\sqrt{EE'}}\langle F^+_{s',j'}|e^{-\Delta\ell}|F^+_{s,j}\rangle + \frac{1}{\pi\sqrt{EE'}}\langle F^+_{s',j'}|\{Q_l,[e^{-\Delta\ell},\bar{Q}_l]\}+\{[Q_l,e^{-\Delta\ell}],\bar{Q}_l\}|F^+_{s,j}\rangle\,.
\tag{E.12}
$$

Finally, since $\partial_a|F^+_{s,j}\rangle = i(j-\frac{1}{2})$, we obtain:

$$
\langle H_{s',j'}|e^{-\Delta\ell}|H_{s,j}\rangle = \frac{1}{2\pi}(E+E'+\Delta(\Delta+j-\frac{1}{2})\frac{\Gamma(\Delta\pm is\pm is')}{\Gamma(2\Delta)}\delta_{j,j'}\,.
\tag{E.13}
$$

The matrix elements $\langle H_{s',j'}|e^{-\Delta\ell}|Z_j\rangle$, $\langle Z_{j'}|e^{-\Delta\ell}|H_{s,j}\rangle$, $\langle Z_{j'}|e^{-\Delta\ell}|Z_j\rangle$ follow from (E.13) by taking the limit $s \to \frac{i}{2}(j - \frac{1}{2})$ for the appropriate state. Computing $\langle L_{s',j'}|e^{-\Delta\ell}|L_{s,j}\rangle$ using (E.10) we obtain

$$\langle L_{s',j'}|e^{-\Delta\ell}|L_{s,j}\rangle = \frac{1}{2\pi}\left(E + E' + \Delta(\Delta - j + \frac{1}{2})\right)\frac{\Gamma(\Delta \pm is \pm is')}{\Gamma(2\Delta)}\delta_{j,j'}. \qquad \text{(E.14)}$$

Finally, to compute the mixed matrix elements, we use (E.11) and the fact that $F^+$ state is annihilated by $Q_r, Q_l$:

$$\begin{aligned}
\langle H_{s',j'}|e^{-\Delta\ell}|L_{s,j}\rangle &= \frac{i}{\pi\sqrt{EE'}}\langle F^+_{s',j'}|Q_l e^{-\Delta\ell}\bar{Q}_r|F^+_{s,j}\rangle \\
&= -\frac{i}{\pi\sqrt{EE'}}\langle F^+_{s',j'}|i\Delta e^{-\ell\Delta}\left(e^{-\frac{\ell}{2}-ia} - i\Delta\psi_l\bar{\psi}_r\right)|F^+_{s,j}\rangle \\
&= \frac{\Delta}{\pi\sqrt{EE'}}\langle F^+_{s',j'}|e^{-\ell(\Delta+\frac{1}{2})-ia}|F^+_{s,j}\rangle \\
&= \frac{\Delta}{\pi}\frac{\Gamma(\Delta + \frac{1}{2} \pm is \pm is')}{\Gamma(2\Delta + 1)}\delta_{j',j-1} = \frac{1}{\pi}\frac{\Gamma(\Delta + \frac{1}{2} \pm is \pm is')}{\Gamma(2\Delta)}\delta_{j',j-1}. \quad \text{(E.15)}
\end{aligned}$$

## F  Check of the composition law

In this appendix we check the composition law (138). We will do this in two steps. First we will check it for the particular case of $j = 0$. And then we will make a separate argument that is valid for general $j$.

In order to check it for the $j = 0$ case, we can first make the gauge choices

$$x_1 = 1, \quad x_3 = 0, \quad \rho_3 = 0, \quad a_3 = 0, \quad \theta_{1-} = \bar{\theta}_{1-} = \theta_{3-} = \bar{\theta}_{3-} = 0. \qquad \text{(F.1)}$$

The propagator for $j = 0$ was given in (129) (130). Then, after integrating over $a_2, \chi_2, \theta_{2-}, \bar{\theta}_2$ with Mathematica[22] we get the integral

$$\frac{1}{2\pi\hat{q}}\int dx_2 d\rho_2\, da_2\, d\theta_{2-}\, d\bar{\theta}_{2-}\, d\chi_2 P_0(1,2)P_0(2,3) = (\chi_1 e^{-ia_1/2} + \chi_3 e^{ia_1/2})z_1^{1/4}e^{-(\gamma_1-\gamma_3)}\frac{1}{\pi}I, \quad \text{(F.2)}$$

where $z_i = e^{-\rho_i}$ and we set $\mathfrak{q} = 1$ in this calculation (note that $z_3 = 1$ due to (F.1)). $I$ is the integral

$$\begin{aligned}
I = \int_0^1 \frac{dx_2}{\sqrt{x_2(1-x_2)}}\int_0^\infty d\sqrt{z_2}\, e^{-\frac{(\sqrt{z_1}+\sqrt{z_2})^2}{(1-x_2)} - \frac{(1+\sqrt{z_2})^2}{x_2}}\Bigg[\frac{(1+\sqrt{z_2})^2}{x_2^2} + \frac{(\sqrt{z_1}+\sqrt{z_2})^2}{(1-x_2)^2} \\
+ \frac{2(1+\sqrt{z_2})(\sqrt{z_1}+\sqrt{z_2})}{x_2(1-x_2)} - \frac{1}{2}\left(\frac{1}{x_2} + \frac{1}{1-x_2}\right)\Bigg].
\end{aligned} \qquad \text{(F.3)}$$

Note that this integrand is not positive definite. This means that, after integrating out the fermions, we do not get a positive measure for the shape of the curve. From now on we drop the subscript '2'. Note that the $z$ integral in (F.3) is a total derivative, so that the integral over

---

[22]We used the package Grassmann.n from Mathew Headrick.

$\zeta = \sqrt{z}$ can be done as

$$I = \int_0^1 \frac{dx}{\sqrt{x(1-x)}} \int_0^\infty d\zeta e^\varphi \frac{1}{4}\left[(\partial_\zeta \varphi)^2 + \partial_\zeta^2 \varphi\right] = \int_0^1 \frac{dx}{\sqrt{x(1-x)}} \frac{1}{4} e^{\varphi(0)}(-\partial_\zeta \varphi(0))$$

$$= \frac{1}{2} \int_0^1 \frac{dx}{\sqrt{x(1-x)}} e^{-\frac{z_1}{1-x} - \frac{1}{x}} \left(\frac{\sqrt{z_1}}{(1-x)} + \frac{1}{x}\right),$$

$$\text{with} \quad \varphi(\zeta) \equiv -(\sqrt{z_1} + \zeta)^2/(1-x) - (1+\zeta)^2/x. \tag{F.4}$$

This final $x$ integral is done via a change of variables

$$y = \frac{x}{1-x}, \quad x = \frac{y}{1+y}, \quad 1-x = \frac{1}{1+y}, \tag{F.5}$$

which is the SL(2) transformation that maps one to infinity. Now the range of $y$ is $y \in [0, \infty]$. After this change of variables the above integral (F.4) becomes

$$I = \frac{1}{2} e^{-z_1 - 1} \int_0^\infty \frac{dy}{\sqrt{y}} \left(\frac{1}{y} + \sqrt{z_1}\right) e^{-z_1 y - 1/y} = \sqrt{\pi} e^{-(\sqrt{z_1}+1)^2}. \tag{F.6}$$

We see that we get the expected form of the propagator, $P(1,3)$ in the gauge (F.1).

Now we turn to the problem of checking the general $j$ case. It turns out that it is useful to derive an independent result which is also interesting.

## F.1 Zero energy wavefunctions of the boundary particle

Here we look at the zero energy wavefunctions for the $\mathcal{N} = 2$ problem. We solve for the functions now of one boundary particle that satisfy

$$Q\Psi = \bar{Q}\Psi = 0. \tag{F.7}$$

These wavefunctions are functions of $\gamma, \rho, x, a, \theta_-, \bar{\theta}_-, \chi$. We work the reduced formalism of section 4.3.2. These functions are not invariant under the left $OSp(2|2)$ symmetries, but we do impose the right gauge symmetries. We will require that they transform in a definite way under $\mathcal{J}^R$. Since $Q, \bar{Q}$ has $OSp(2|2)$ symmetry, the zero energy states must form a representation of this symmetry group.

We can furthermore diagonalize the left generator $\mathcal{L}_-$ to obtain

$$\Psi_{\mathsf{q},k,j} = e^{-\mathsf{q}\gamma + ikx + ija}\left[e^{-ia/2}F(\rho, \theta_-, \bar{\theta}_-) + e^{ia/2}\chi \tilde{F}(\rho, \theta_-, \bar{\theta}_-)\right]. \tag{F.8}$$

Note that in our notation $j - \frac{1}{2}$ corresponds to the right $R$ charge of the wavefunction. Note that on these functions $\mathcal{G}_-$ and $\bar{\mathcal{G}}_-$ act essentially as complex fermion. Of course, these commute with $Q$. So it is convenient to split the solutions into two possibilities obeying

$$\bar{\mathcal{G}}_-\Psi_+ = 0, \quad \mathcal{G}_-\Psi_- = 0. \tag{F.9}$$

Of course, due to the anticommutation relations we get that $\mathcal{G}_-\Psi_+ \propto \Psi_-$ and $\bar{\mathcal{G}}_-\Psi_- \propto \Psi_+$. Solving these equations we find

$$\Psi_+ = e^{-\mathsf{q}\gamma + ikx + ija} e^{-\rho/2}\left[e^{-ia/2}(1 - ik\theta_-\bar{\theta}_-)K_{\frac{1}{2}+j}(v) - 2e^{ia/2}\sqrt{ik\mathsf{q}}\chi\theta_-K_{\frac{1}{2}-j}(v)\right],$$

$$\Psi_- = e^{-\mathsf{q}\gamma + ikx + ija} e^{-\rho/2}\left[-e^{-ia/2}\frac{\bar{\theta}_-\sqrt{ik\mathsf{q}}}{\mathsf{q}}K_{\frac{1}{2}+j}(v) + e^{ia/2}\chi(1 + ik\theta_-\bar{\theta}_-)K_{\frac{1}{2}-j}(v)\right],$$

$$\text{with} \quad v \equiv 2e^{-\rho/2}\sqrt{ik\mathsf{q}}, \tag{F.10}$$

where each is defined up to a normalization. Note that the wavefunction with opposite q and $k$ can be obained from these by simply taking $k \to -k$ and $q \to -q$, which leaves $v$ invariant. Also note that $\Psi_+$ is Grassmann-even and $\Psi_-$ Grassmann-odd. The fact that there is a unique solution implies that there is a single zero energy solution for each $j$ and $k$. In writing these wavefunctions we have picked the solution of the equations that is normalizable when $\rho \to -\infty$. These wavefunctions are localized in the radial $\rho$ direction.

One of the lessons of this analysis is that, for each $j = \mathcal{J}^R$, there is a single irreducible representation in the zero-energy representation.

So, when we Fourier transform the propagator, we expect to find an answer which involves these functions. One question is: what is the inner product among these functions? The inner product can be computed with the measure in (131) involves

$$\int d\rho \, d\theta_- \, d\bar{\theta}_- \, d\chi \, \Psi_{q,k,j,+} \Psi_{-q,-k,-j,-} \propto -2ik\frac{1}{\cos \pi j}, \tag{F.11}$$

$$\int d\rho \, d\theta_- \, d\bar{\theta}_- \, d\chi \, \Psi_{q,k,j,-} \Psi_{-q,-k,-j,+} \propto 2ik\frac{1}{\cos \pi j}. \tag{F.12}$$

These wavefunctions were computed in Euclidean space, so they are appropriate for the description of the Fourier transform of the propagator. It is also interesting to discuss the wavefunction in Lorentzian signature. In that case we simply change $x \to -it$, $k \to -i\omega$, so that the time dependence is now $e^{-i\omega t}$. Then the variable $v$ becomes $v = 2e^{-\rho/2}\sqrt{\omega q}$. These are normalizable wavefunctions.

Note also that if we set $\theta_- = \bar{\theta}_- = 0$ the functions become

$$\Psi_{q,k,j,+} = e^{-q\gamma+ikx+ija}e^{-\rho/2}\left[e^{-ia/2}K_{\frac{1}{2}+j}(v)\right],$$
$$\Psi_{q,k,j,-} = e^{-q\gamma+ikx+ija}e^{-\rho/2}\left[e^{ia/2}\chi K_{\frac{1}{2}-j}(v)\right],$$
$$\text{with} \quad v \equiv 2e^{-\rho/2}\sqrt{ikq}, \tag{F.13}$$

where the dependence on $\chi$ appears in only one of the terms.

## F.2 Checking the composition law for general $j$

The propagator (129) has two terms, one involves $\chi_1$ and the other $\chi_2$. Therefore we expect that the propagator has the structure

$$\hat{P} \sim \int dk \left[\Psi_{q,k,j,+}(1)\Psi_{-q,-k,-j,-}(2) \pm \Psi_{q,k,j,-}(1)\Psi_{-q,-k,-j,+}(2)\right], \tag{F.14}$$

where we have not worked out the precise signs and factors. The Fourier transform of the functions appearing in the propagator can be computed at zero $\theta_-$ $\bar{\theta}_-$.

Note that [11]

$$I = \int_0^\infty dx \, e^{-ikx} \frac{q\sqrt{z_1 z_2}}{x} e^{-q\frac{z_1+z_2}{x}} K_\nu\left(\frac{2q\sqrt{z_1 z_2}}{x}\right) = 2q\sqrt{z_1 z_2}K_\nu\left(2\sqrt{ikq z_1}\right)K_\nu\left(2\sqrt{ikq z_2}\right). \tag{F.15}$$

But we see that if we want to reproduce the above wavefunctions would would need to write them as

$$I = \frac{2}{ik}\left[\sqrt{ikq z_1}K_\nu\left(2\sqrt{ikq z_1}\right)\right]\left[\sqrt{ikq z_2}K_\nu\left(2\sqrt{ikq z_2}\right)\right]. \tag{F.16}$$

The extra factor of $k$ seems consistent with the extra $k$ in the inner product, in the sense that it would cancel the extra $k$ in the inner products (F.11), (F.12).

The simple $j$ dependence in the inner products (F.11) (F.12) is also saying that the composition law works for any $j$ as is works for $j = 0$, once we put the extra factor of $\cos \pi j$ in the propagator as in (134). So the check of the composition law for general $j$ is now complete. Namely, we first checked it for $j = 0$ in position space. Then for general $j$ we go to Fourier space and we use the wavefunctions discussed above. We do not have to work out the precise signs and factors of $k$ because we already know that they work out for $j = 0$. So, all we need to check is the $j$ dependence of the inner products.

## G  Extra zero modes for small $N$

Here we discuss the zero modes of the operator $\psi_1 \bar{\psi}_2$, projected into the zero eigenstate sector.

For even $N$, we find that the number of zero modes is

$$\text{number of zero modes} = D(N, j, \hat{q}) - \binom{N-2}{N/2 + j - 1} = D(N, j, \hat{q}) - \binom{N-2}{\frac{1}{2}(N-2) + j}. \quad \text{(G.1)}$$

Here $D(N, j, \hat{q})$ is the number of ground states with charge $j$. For $j = 0, \hat{q} = 3, D = 2 \times 3^{N/2-1}$. This formula is valid if the above quantity is positive. If the above quantity is negative, there are no zero modes. Note that $D$ grows more slowly than $\binom{N}{N/2}$, so at large $N$ there are no degeneracies. To obtain this formula, consider starting with a ground state and acting with $\bar{\psi}$. This gives a state with charge $j - 1$. In general, this state is not a ground state, so we count the number of UV states with charge $j - 1$, which would be $\binom{N}{N/2+j-1}$. If this Hilbert space is smaller than the Hilbert space of ground states, we will get some degeneracies. So the naive number of zero modes would be the difference in the dimensions:

$$\text{naive number of zero modes} = D_{IR} - D_{UV} = D(N, j, \hat{q}) - \binom{N}{N/2 + j - 1}. \quad \text{(G.2)}$$

This counting is not quite right because $\bar{\psi}_2$ even as a UV operator has a kernel, e.g., states where a $\bar{\psi}_2 |0\rangle$. So when counting the UV dimension, we should only count states that are in the image of $\bar{\psi}_2$ and are not in the kernel of $\psi_1$. This constraint reduces the effective number of spins from $N \to N - 2$ in the second term of (G.2), leading to (G.1).

We can obtain a formula for the number of ground states by fourier transforming the refined index:

$$D(N, j, \hat{q}) = \frac{1}{\hat{q}} \sum_{l=0}^{(\hat{q}-3)/2} 2\cos\left(\frac{\pi j(1 + 2l)}{\hat{q}}\right) \left[2\cos\frac{\pi}{2q}(1 + 2l)\right]^N. \quad \text{(G.3)}$$

We numerically tested (G.1) for several small values of $N$ and $\hat{q}$.

## H  Schwarzian coefficient at large $q$

In the large $\hat{q}$ $\mathcal{N} = 1$ SYK model, the Schwarzian coefficient is

$$C_{\mathcal{N}=1} = \frac{N}{16\hat{q}^2 \mathcal{J}}. \quad \text{(H.1)}$$

This can be read off from the thermodynamics computed in [12]. In the $\mathcal{N} = 2$ model, the equations of motion are the complexified version of the equations of motion of the $\mathcal{N} = 1$

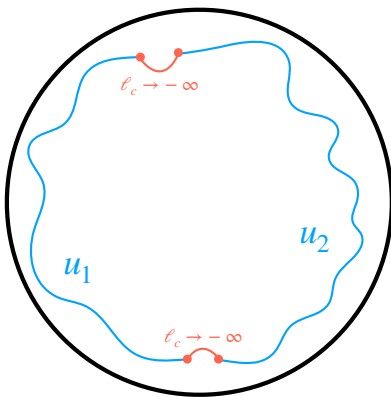

Figure 19: We can view the disk partition function with total length $u_1 + u_2 = \beta$ as an amplitude in the 2-sided Liouville theory $\langle \ell_c | e^{-\beta H} | \ell_c \rangle$, where the proper length of the geodesic on the top and bottom are going to zero. This implies that the renormalized length $\ell_c \to -\infty$. In this limit, the red curves disappear completely.

model. This implies that the Schwarzian coefficient for $\mathcal{N} = 2$ is simply twice that of the $\mathcal{N} = 1$ model (for any choice of $\hat{q}$).

$$C_{\mathcal{N}=2} = \frac{N}{8\hat{q}^2 \mathcal{J}} . \tag{H.2}$$

If this is the case, extrapolating to $q = 3$, we get a prediction $\alpha_s \approx N/(108J)$, with $\mathcal{J} = 3J/2$. This is close to the answer [30] obtained by numerically solving the large $N$ equations $\alpha_s = 0.00842N/J$.

# I Computation of the disk partition function from the 2-sided Liouville quantum mechanics

One question is whether we can reproduce the disk density of states using the Liouville quantum mechanics. This is not obvious because the density of states is a 1-sided quantity, whereas the Hilbert space of Liouville quantum mechanics is associated to the 2-sided theory. The idea is to compute the partition function of the thermal circle with total length $\beta$. We can arbitrarily divide the circle into a left side and right side. The two sides meet at two points, with the condition that the total length of the circle is $\beta$, see Figure (19). Then the proposal is that

$$Z(\beta) \propto \lim_{\ell_c \to -\infty} \langle \ell_c | e^{-\beta H_{\text{Liouville}}} | \ell_c \rangle , \tag{I.1}$$

where $|\ell_c\rangle$ is a position eigenstate. We are imposing that the renormalized length goes to $-\infty$ at two points in the Euclidean past and future. (This is because the length is going to zero). Now we can write this in terms of the energy eigenfunctions

$$Z(\beta) \propto \int \mathrm{d}s |\psi_s(\ell_c)|^2 e^{-\beta s^2} . \tag{I.2}$$

Here $\psi_s(\ell)$ is an energy eigenstate with energy $s^2$ with the scattering normalization that is natural to the 2-sided problem (see, e.g., equation 3.4 of [26]):

$$\psi_s(\ell_c) = \frac{2^{1-2is}}{\Gamma(-2is)} K_{2is}(4e^{-\ell_c/2}) . \tag{I.3}$$

The flat measure $ds$ is natural if we think of $s$ as the momentum of the incoming wave. Now notice that at large $\ell$, the Bessel function decays to zero, but in a way that is independent of the energy, e.g., $K_\alpha(z) \sim e^{-z}\sqrt{\pi/(2z)}$. Hence we get $|\psi_s(\ell)|^2 \propto s \sinh(2\pi s)$. This gives the expected density of states $\rho(E) \propto \sinh\left(2\pi\sqrt{E}\right)$. The overall normalization $\sim e^{S_0}$ is not determined in this method.

This works similarly in the $\mathcal{N} = 2$ Liouville case, where the eigenfunctions are also Bessel functions. In this case the boundary condition should be $|\ell_c, a = 0\rangle$. (Since the length of the segment is going to zero, the holonomy between the two sides should also go to zero if the gauge field is continuous.) In addition, we choose

$$(\psi_l + i\psi_r)|\ell_c, a = 0\rangle = (\bar{\psi}_l + i\bar{\psi}_r)|\ell_c, a = 0\rangle = 0 \,. \tag{I.4}$$

This is the natural condition for fermions in the infinite temperature TFD. Notice that as $\ell_c \to \infty$, we have that the second term in the supercharges (8) dominates (assuming that we regulate the position eigenstates by a small amount of smearing independent of $\ell_c$):

$$\begin{aligned} Q_r &= e^{-\ell_c/2}\psi_l \,, & \bar{Q}_r &= e^{-\ell_c/2}\bar{\psi}_l \,, \\ Q_l &= -e^{-\ell_c/2 - ia}\psi_r \,, & \bar{Q}_l &= -e^{-\ell_c/2}\bar{\psi}_r \,. \end{aligned} \tag{I.5}$$

Therefore this state satisfies

$$(Q_r - iQ_\ell)|\ell_c, a = 0\rangle = (\bar{Q}_r - i\bar{Q}_\ell)|\ell_c, a = 0\rangle = 0 \,. \tag{I.6}$$

Since the supercharges commute with the Hamiltonian, this must be true for the $|\text{TFD}\rangle$ state at finite $\beta$:

$$|\text{TFD}\rangle = e^{-\beta H/2}|\ell_c, a = 0\rangle \,. \tag{I.7}$$

This provides a derivation of the equation

$$(Q_r - iQ_\ell)|\text{TFD}\rangle = (\bar{Q}_r - i\bar{Q}_\ell)|\text{TFD}\rangle = 0 \,. \tag{I.8}$$

Note that this method also generalizes to the disk with a finite chemical potential, by inserting $e^{-\mu Q}$, which can be achieved by acting with $e^{-\mu J}$ in the 2-sided theory. In this way, we can read off the density of states for each charge sector $j$.

The following comment is an aside. Note that we could imagine evaluating $\langle \ell_c|e^{-\beta H_{\text{Liouville}}}|\ell_c\rangle$ via saddle point. The saddle point solutions are

$$e^{-\ell} = \sin^{-2}(\pi u/\beta)\,, \quad a = 2\pi n\hat{q}(u/\beta)\,. \tag{I.9}$$

Since the partition function is 1-loop exact, this implies that the saddle-point approximation in the Liouville theory is 1-loop exact in the limit of large $\ell_c$. It would be interesting to explain the 1-loop exactness in the context of Liouville quantum mechanics.

## J Simplifying the "phase" factors

In this subsection we will consider the bosonic case and simplify the expression of the phase factors that appears in the chain of propagators dressing a diagram. These involve the combination

$$\varphi_{12} + \varphi_{23} + \varphi_{34} + \cdots \varphi_{n1}\,, \quad \text{with} \quad \varphi_{ij} = \frac{e^{-\rho_i} + e^{-\rho_j}}{x_i - x_j}\,, \tag{J.1}$$

see (94). This does not involves the variables $\gamma_i$ so we expect that they could be expressed in terms of the distance variables. In fact, it is relatively easy to do that. First we note that a given $\rho_i$ appears in just two terms $\varphi_{i-1,i}$ and $\varphi_{i,i+1}$ and these two combine into

$$e^{-\rho_i} \frac{x_{i+1} - x_{i-1}}{(x_i - x_{i-1})(x_{i+1} - x_i)} = \exp\left(-\frac{\ell_{i,i+1}}{2} - \frac{\ell_{i,i-1}}{2} + \frac{\ell_{i-1,i+1}}{2}\right), \qquad \text{(J.2)}$$

where the total argument of the exponent in the right hand side is always negative due to the triangle inequality for distances. Some care is needed for the last term involving $\varphi_{n1}$, but in the end the formula is also valid for that case.

So the conclusion is that the final phase factor terms have the form

$$\prod_i e^{-\varphi_{i,i+1}} = \exp\left(-\sum_i e^{-\frac{\ell_{i,i+1}}{2} - \frac{\ell_{i,i-1}}{2} + \frac{\ell_{i-1,i+1}}{2}}\right). \qquad \text{(J.3)}$$

In particular, only the lengths between nearest neighbors and next-to-nearest neighbors appear. For the supersymmetric case, we expect a similar simplification.

## K    Relative entropy

An interesting question is whether there is any notion of the entanglement wedge of a subregion in the extremal regime, where the boundary mode is highly quantum. Here a subregion could just be one side of the 2-sided wormhole. To address this question, we compute the relative entropy of different density matrices $\rho, \sigma$ which are obtained by tracing out one side of the wormhole. By inserting different types of matter in the middle of the wormhole, one could diagnose our ability to do 1-sided bulk reconstruction. To compute the relative entropy, we can use the replica trick:

$$S(\rho \mid \sigma) = -\partial_n \operatorname{tr}\left(\rho \sigma^{n-1} - \rho^n\right)\Big|_{n=1}. \qquad \text{(K.1)}$$

If we take $\rho$ to be the maximally mixed state (e.g. the empty wormhole) and $\sigma$ to be 1-sided density matrix of a wormhole with $q$ identical light particles with $\Delta = 0$, we get (compare with (167)):

$$\lim_{n\to 1} \frac{1}{1-n}\left((2q(n-1)-1)!! - 1\right) = \psi^{(0)}\left(\frac{1}{2}\right) - \log(2) = q \times (1.270\dots), \qquad \text{(K.2)}$$

where the analytic continuation of the double factorial is as in (167).

For $q$ heavy particles $\Delta \gg 1$, we also get a relative entropy that grows linearly with $q$:

$$-\partial_n \left[\frac{(2q(n-1))!}{(q(n-1)+1)!(q(n-1))!} - 1\right]\Bigg|_{n=1} = q. \qquad \text{(K.3)}$$

In the above calculations we are asking whether we can distinguish the empty wormhole from the case for a wormhole with matter. Another question more closely related to the standard bulk reconstruction question is whether we can distinguish a wormhole with a qubit in the state $|\uparrow\rangle$ from a wormhole with a qubit in the state $|\downarrow\rangle$. To model this, we can consider 2 species of particles, both with the same dimension. For $\Delta = 0$, we get

$$\lim_{n\to 1} \frac{1}{1-n}\left((2(n-1)-1)!! - (2n-1)!!\right) = 2. \qquad \text{(K.4)}$$

This says that we can recover some partial information about the state of the probe qubit. Of course, we cannot recover the state exactly or else we would violate cloning.

An interesting setup for future work is to consider with $K$ particle insertions, all separated by infinite Euclidean time evolution. The state of $K-1$ of the $K$ particles are fixed, but one of the $K$ particles is treated as the probe qubit, the state of which we would like to reconstruct. In this setup, it would be interesting to study the relative entropy as a function of which of the $K$ particles we choose to have an unknown state. The semiclassical picture would suggest that if the probe qubit is to the left of the midpoint of the wormhole, the corresponding relative entropy of the left boundary would be large, whereas if the probe qubit is to the right of the midpoint, the relative entropy should be small. It would be interesting to do this computation at large $\Delta \gg 1$; for $\Delta = 0$ we note that we just have Wick contractions so there is no difference in which order we put the particles.

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
