# Peer review of "Looking at supersymmetric black holes for a very long time"

_SciPost Physics, doi:SciPost Phys. 14, 128 (2023)_

## Round 2 · Referee Report · Anonymous (Referee 1) · 2022-9-19

Report

The paper extensively studies N=2 super-Schwarzian theory. The authors explain its relation to supersymmetric black holes and SYK theory. The paper has a number of very interesting results and I highly recommend it for publication. One of them is the fact that black hole throat length stays finite even in the extremal limit. The other one is the emergence of bulk time. This model has an exponentially large ground state degeneracy. The authors find that one can compute expectation values in this BPS sector using bulk AdS_2 despite the fact that the boundary Hamiltonian is zero, hence there is no boundary time.

Despite above praises, there is still room for criticism: 1. Higher-dimensional supersymmetric black holes have Aretakis instability. Is it related to non-decaying correlators? 2. The fate of the bulk dilaton field is not addressed.
3. Sections 5.1, 5.2 are hard to read. The paper is already pretty long, so answering these questions is optional.

On more technical side, I noticed a few typos: 1. As far as I understand |j|<=1/2 below eq. (22) should be strict inequality 2. There is a number of "the the" in the paper

---

## Round 2 · Referee Report · Anonymous (Referee 2) · 2022-11-16

Report

This paper studies correlation functions of local operators in supersymmetric black holes. In the context of $N=2$ supersymmetric theories, the authors calculate the $n$-point function of operators inserted at the boundary of the near-horizon AdS$_2$. In particular, they obtain an explicit formula for the two-point function (in two different ways). They find that the two-point function is independent of time in the large-time limit. They also discuss the non-supersymmetric case where there is a similar universality at large time (but with different details).

This gives a way to produce an interesting set of non-trivial observables for the AdS$_2$ (and therefore CFT$_1$) theory.

Further, the authors give an interpretation of their results in terms of supersymmetric wormholes connecting the two sides of Lorentzian AdS$_2$ (empty for $n=2$ and wormholes with particles for $n>2$). The authors also study similar observables in the supersymmetric SYK model and show consistency.

In my opinion, these are beautiful results and should definitely be published as a paper in SciPost.

Here are my comments/questions, mainly about the presentation (some of these refer to the companion paper).

  1. What is the definition of $\Delta$? In terms of presentation, I think it will help a non-expert reader to be explicit about this (say at the level of a working definition) at the beginning of the paper. Indeed, there are useful comments about this in Section 5.5 , but it is a bit hidden and only appears after Looking at the paper for a very long time.

In particular, $\Delta$ is sometimes referred to as the conformal dimension -- which conformal group is being used here? Does it need an embedding into a UV CFT? (If so how do we discuss BHs in asymptotically flat space?) Is it the emergent boundary symmetry of the near-BPS BH? Or is it defined in terms of the AdS$_2$ bulk as the near-boundary behavior of a field (or perhaps through the relation to the mass of the field)? Or simply as the conjugate to $\ell$? Perhaps some of these definitions are equivalent, but it will be helpful if the authors comment on this.

Further, if there is a consistent embedding into an UV CFT (as in this paper), is $\Delta$ related to the UV theory?

  1. Consider the wormholes with operators inserted. The operators decrease the length as well as the entanglement entropy. Are these two phenomena related? What happens if the insertion is a unitary transformation on one of the boundaries -- in this case we expect that the entanglement does not change. What happens to the length formula in the paper? Is the constant value expected to vanish? (Btw, in many places in the companion paper this length is called $o(1)$ -- I suppose this is a typo and it should be $O(1)$ = constant?)

  2. In Section 5.4, it says that $x$ and $y$ are not the boundary time and temperature. However, is there a geometric interpretation at all of $x$ and $y$? Or should we think of it only as a generator of the microcanonical coefficients in Equation (171)? (The latter clearly have a bulk interpretation.)

  3. When we say that the correlators respect time-reparameterization symmetry, do we implicitly assume that the reparameterization is monotonic (or are folds allowed)? Where is this assumption made? I ask because of the dependence on order of the insertions.

---

## Editorial Decision

published